# N2FXm, a method for joint nuclear and cytoplasmic volume measurements, unravels the osmo-mechanical regulation of nuclear volume in mammalian cells

Fabrizio A. Pennacchio[1,2], Alessandro Poli[1], Francesca Michela Pramotton [3], Stefania Lavore[1], Ilaria Rancati[1], Mario Cinquanta [1], Daan Vorselen [4], Elisabetta Prina[1], Orso Maria Romano[1], Aldo Ferrari[3], Matthieu Piel [5,6], Marco Cosentino Lagomarsino [1,7] & Paolo Maiuri [1,8] ✉

In eukaryotes, cytoplasmic and nuclear volumes are tightly regulated to ensure proper cell homeostasis. However, current methods to measure cytoplasmic and nuclear volumes, including confocal 3D reconstruction, have limitations, such as relying on two-dimensional projections or poor vertical resolution. Here, to overcome these limitations, we describe a method, N2FXm, to jointly measure cytoplasmic and nuclear volumes in single cultured adhering human cells, in real time, and across cell cycles. We find that this method accurately provides joint size over dynamic measurements and at different time resolutions. Moreover, by combining several experimental perturbations and analyzing a mathematical model including osmotic effects and tension, we show that N2FXm can give relevant insights on how mechanical forces exerted by the cytoskeleton on the nuclear envelope can affect the growth of nucleus volume by biasing nuclear import. Our method, by allowing for accurate joint nuclear and cytoplasmic volume dynamic measurements at different time resolutions, highlights the non-constancy of the nucleus/cytoplasm ratio along the cell cycle.

In all living systems, size control of cells and intracellular organelles is essential for the optimal regulation of several biological functions[1–3]. In multicellular organisms, cell and organelle size is a characteristic feature of a given cell type, and both cell-to-cell and mean variations are often associated with pathological conditions such as cancer or aging[4,5].

The size of the nucleus, the largest cellular organelle, generally scales linearly with cell size. In yeast, nuclear to cell volumetric ratio (karyoplasmic ratio) is roughly constant along the entire cell-cycle[6–8]. For mammalian cells, the commonly accepted model considers cytoplasm and nucleus in mechanical equilibrium where the dominant forces are due to osmotic pressure, and nuclear envelope tension counteracting the growing tendency of the nucleus[9]. At the steady state, indeed, nucleus/cytoplasm osmotic imbalance is the result of the combination of active and passive transport mechanisms of ions and

[1]IFOM ETS—The AIRC Institute of Molecular Oncology, Via Adamello 16, 20139 Milan, Italy. [2]Laboratory of Applied Mechanobiology, Department of Health Sciences and Technology, ETH Zurich, Vladimir-Prelog-Weg 4, 8093 Zurich, Switzerland. [3]Laboratory of Thermodynamics in Emerging Technologies, Department of Mechanical and Process Engineering, ETH Zurich, Sonneggstrasse 3, Zurich CH-8092, Switzerland. [4]Department of Biology and Howard Hughes Medical Institute, University of Washington, Seattle, WA 98105, USA. [5]Institut Curie, PSL Research University, CNRS, UMR 144, F-75005 Paris, France. [6]Institut Pierre-Gilles de Gennes, PSL Research University, F-75005 Paris, France. [7]Dipartimento di Fisica, Università degli Studi di Milano, and I.N.F.N., Via Celoria 16, 20133 Milan, Italy. [8]Dipartimento di Medicina Molecolare e Biotecnologie Mediche, Università degli Studi di Napoli Federico II, Via S. Pansini 5, 80131 Naples, Italy. ✉e-mail: paolo.maiuri@unina.it

molecules through nuclear pores. Specifically, active RanGAP-mediated nucleocytoplasmic transport leads to different transport rates for different proteins in and out of the nucleus, resulting in distinct concentrations of these proteins in the two cellular compartments. Consequently, variations in active transport rates may contribute to osmotic pressure differences[10,11]. However, the detailed mechanisms underlying the dynamic of nucleus-cytoplasm volume coupling are still mostly unclear, partially due to technical limitations[12]. Indeed, the most used method to dynamically measure nuclear volume in living adherent mammalian cells is to perform 3D volume reconstruction of time-lapse confocal images. This technique is typically slow and with low throughput and, more importantly, it requires continuous illumination of cells, which finally affects cell cycle progression.

Here, we developed a method, nuclear fluorescence exclusion microscopy (N2FXm), to overcome the common limitations of confocal 3D reconstruction in measuring nuclear volume in living adherent mammalian cells. N2FXm allows joint high-throughput volumetric measurement of cytoplasm and nucleus. Thanks to this method, we highlighted the role of cytoplasmic mechanical forces in the regulation of nuclear volume. We finally propose the hypothesis that nuclear volume regulation leverages the previously reported mechano-mediated tuning of nucleo-cytoplasmic transport, and we support this hypothesis with a mathematical model.

## Results

### N2FXm, a new method to simultaneously measure cytoplasmic and nuclear volume in living cells

To measure jointly and in real time both nuclear and cytoplasmic volumes of living cells we developed a technique which we named

nuclear fluorescence exclusion microscopy (N2FXm). This method was conceived as an evolution of fluorescence exclusion microscopy (FXm), an imaging-based technique that allows measuring cellular volumes using low-magnification objectives[13–15]. In standard FXm, cells are injected in a microfluidic chamber of known height filled with a fluorescent dye, in our case high molecular weight RED-dextran, which is not internalized by cells. For any object in the chamber, the corresponding drop in dextran fluorescence is linearly proportional to its volume. Consequently, dextran fluorescence can be scaled linearly between two points of known height: zero, where the chamber is empty, and the maximum, the known chamber height, in correspondence of the pillars sustaining the chamber roof. This calibration allows to define the optical thickness of cells of arbitrary shape in the chamber and, accordingly, cell volume can be precisely computed by integrating it over the segmented cell area (see Methods, Fig. S1a−c). N2FXm extends the FXm technique to nuclear volume measurements. Cell nuclei were negatively stained with the ectopic expression of a green fluorescent protein coupled with a nuclear export signal (GFP-NES), which localizes in the entire cytoplasm except the nucleus, which is also marked with H2B-BFP (see Methods, Fig. 1a, b and S1c−e). In this doubly-labeled system, and in the presence of the extracellular dye, a second calibration is introduced and the cytoplasmic GFP fluorescent signal is scaled according to cell optical thickness. The two calibration steps generate calibrated images for both cell and cytoplasm, obtained from dextran and GFP fluorescent signals, respectively. Since the latter is scaled according to the former, intensity profiles from the two images differ only in correspondence of the nucleus, where the difference of the two is proportional to the nuclear height (Fig. S1c). Finally, to measure nuclear volume, the calibrated GFP signal is integrated within the segmented nuclear region, subtracting the resulting

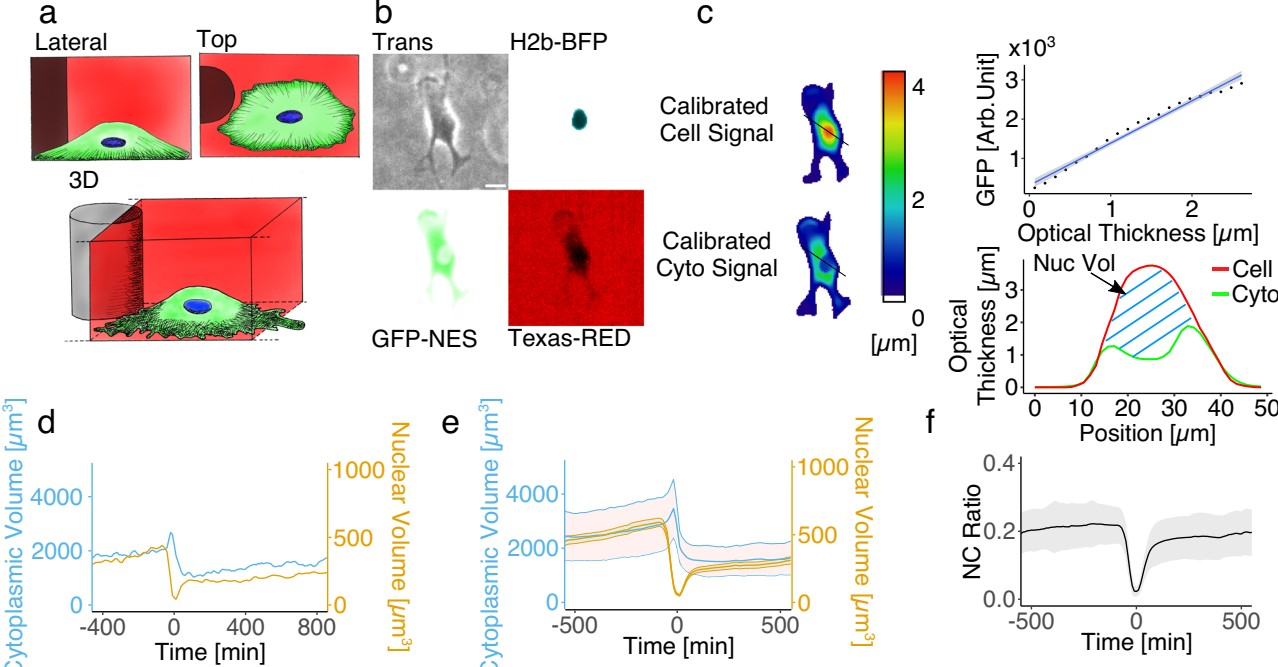

**Fig. 1 | N2FXm development. a** Schematic representation of the cell in the acquisition chamber: bottom, 3D rendering; top left lateral view; top right top view. (**b**) Widefield images of a RPE1 cell in the acquisition chamber: transmission light (top left), H2b-BFP (top right), GFP-NES (bottom left) and Texas red (bottom right). Scale bar is 20 μm. **c** Fluorescence calibration and nuclear volume calculation. On the left: calibrated color-coded images of the cell (top) and cytoplasm (bottom) heights. On the top right: example of experimental calibration curve for the GFP-NES signal in function of the corresponding optical thickness (linear regression

with 95% fit confidence interval). On the bottom right cell and cytoplasm calibrated signal profiles along black line in calibrated color-coded images in (**c**). **d** Nuclear (blue) and cytoplasmic (yellow) volumes trajectories for a single RPE1 cell. Mean (±SE) nuclear and cytoplasmic volumes (**e**) and mean (±SE) NC ratio (**f**) trajectories for RPE1 cell population (*n* = 99). All single curves are aligned to cytokinesis = 0 min. Source data are provided as a Source Data file. "*n*" represents the number of cells examined over at least three independent experiments.

value from the cell volume evaluated over the same domain (see Methods, Fig. S1d, e and Supplementary Movie 1). Cytoplasmic volume is then obtained by subtracting the nuclear volume from the entire cell volume (Figs. 1d and S1d, e). The nucleus/cytoplasm (NC) ratio is here defined and as such afterward referred, as the ratio between nuclear and cytoplasmic volumes (Fig. 1).

This described two-step calibration procedure is performed independently for each cell and at each time point, to correct for both cell-to-cell variations and time fluctuations of GFP-NES expression. It is important to notice that a possible inhomogeneity in the distribution of GFP-NES in the cytoplasm could perturb the linear dependency between the GFP-NES intensity of cytoplasmic pixels and their corresponding optical thickness. However, this effect appears minimal in our data, as the distribution of the $R^2$ of the calibration linear fits is strongly biased toward 1 (median 0.93, Fig. S1f). To experimentally test our method, we used N2FXm to measure the volumes of spherical polymeric μ-particles (DAAM particles[16], see Methods) internalized by RPE1 cells. N2FXm measurements were in good agreement with the volumes estimated with a geometrical measurement (see Methods and Fig. S1h–j, linear fit coefficient: slope = 0.94, Pearson correlation coefficient = 0.89, $R^2 = 0.78$). Moreover, the distribution of the volume ratio estimated by N2FXm and geometrical measurements was compatible with a normal distribution centred on 1, excluding possible systematic errors in our procedure (Fig. S1i). Additionally, we independently measured the nuclear volume distribution of RPE1 cells with both N2FXm and 3D confocal reconstruction (see Methods). The difference between the two distributions was not statistically significant (Fig. S1g). Finally, we assessed if the ectopic expression of GFP-NES or H2B-BFP perturbed cell or nuclear dimensions and we did not find any meaningful differences (Fig. S1k–l). To summarize, the main advantages of N2FXm are: 1) both cell and nuclear volumes are obtained from a single plane illumination, avoiding 3D reconstruction; 2) the technique works with small magnification and low numerical aperture objectives, enabling the simultaneous record of many cells in the same field of view; 3) the imaging needs low illumination, reducing phototoxicity and making the technique suitable for long-term experiments (see Supplementary Movie 1). Nevertheless, our method can measure the volume of the nucleus only when this compartment is properly defined. In the few frames between nuclear envelope breakdown and nucleus sealing after mitosis (certainly before post-mitotic nuclear expansion, the fast nuclear volume increase at mitosis exit[17]), it can be argued that the nucleus is not an isolated compartment anymore. During this time interval, GFP-NES diffuses inside the nucleus, but our method measures a residual volume that is still not accessible to it.

## N2FXm measurement of cytoplasmic and nuclear volumes across cell division

To show the potential of the method, we employed N2FXm to explore the nucleus-cytoplasm volumetric coupling in non-synchronized proliferating cells from 5 different epithelial cell lines, normal and transformed: RPE1, MCF10A, MCF7, MCF10 DCIS.COM (afterwards and in figures mentioned for simplicity as DCIS.com) and MCF10-CA, (Fig. 1d–f, S1m). Differently from earlier reports for yeast and from what hypothesized for decades per mammalian cells[6,7], cytoplasm and nucleus volume mean temporal trajectories clearly show that NC ratio is not constant over the cell cycle (Fig. 1f and Fig. S1m).

To gain a deeper understanding of the homeostatic nucleus-cytoplasm coupling we then exploited N2FXm analyzing single-cell dynamic measurements of cytoplasmic and nuclear volumes. Initially, we considered three specific time-points across cell division: *I*, the instant preceding nuclear envelope breakdown (NEB); *II*, the onset of cellular roundup at division; and *III*, the instant successive to the post-mitotic nuclear expansion (PME), (for point identification methods, see "Curve generation and point detection" section in supporting information, Fig. 2a, b and Fig. S2a). We found that NEB systematically

precedes the onset of cellular roundup by ~10–20 min (Fig. 2c). However, the temporal resolution of our experiments (10 min) was too small to precisely distinguish these two events. Moreover, we found a strong positive correlation between both nuclear and cytoplasmic volumes at NEB and PME, the end of a cell cycle and the start of the following one, respectively (Fig. S2b). These results are in line with previous findings and reinforce the idea that "size-memory" mechanisms act to preserve both cytoplasm and nuclear dimensions of the cell population[15]. Interestingly, while on average cytoplasmic volume almost perfectly halves across cell division[15], the nucleus decreases on average by ~2.5 times (Fig. 2d). NC volume ratio is not constant between NEB and PME and it increases by a minimum of ~20% for all the analyzed cell lines (Fig. 2e). These observations suggested nontrivial homeostatic coupling mechanisms between cytoplasm and nucleus.

Then, to test the relationship between temporal evolution of cytoplasm and nuclear volume growth we computed for both the volumetric specific growth rates, $\frac{1}{V} \cdot \frac{dV}{dt}$, at fixed size (Fig. 2f, g and Supplementary Fig. S2c). This quantitative analysis[18] discriminates between different average growth laws. Generally, an average exponential growth of the volume, $V(t) = V_0 \cdot e^{(\alpha \cdot t)}$, implies a constant specific growth rate, $\frac{1}{V} \cdot \frac{dV}{dt} = \alpha$. Differently, a size-independent constant growth, $V(t) = V_0 + K \cdot t$, implies a decreasing specific growth rate, $\frac{1}{V} \cdot \frac{dV}{dt} = \frac{K}{V}$, as a function of the volume.

Overall, the analysis of our preliminary results suggests that, for all the cell lines considered, the nucleus-cytoplasm volumetric coupling could not be simply defined by a pure osmotic equilibrium, which would lead, instead, to a constant value of the NC ratio[19,20].

## Cytoskeletal forces impact nuclear volume

To further explore the biophysical factors regulating the nucleus-cytoplasm volumetric coupling, we performed a set of experiments on shorter time scales, to osmotically or mechanically perturb RPE1 cells. These stimuli are known to impact both cell and nuclear volume[21–24]. We coupled these experiments with measurements of the cytoskeletal forces acting on the nuclear envelope (NE), using a FRET sensor based on the LINC complex protein Nesprin 1. This tension sensor spans from the inner NE, where it binds SUN domain-containing proteins, to the cytoplasm, where it binds to actin. Therefore, it is sensitive only to forces exerted by the cytoskeleton to the NE and does not sense tension parallel to the membrane[25]. We also used a set of simple biophysical models to rationalize the results (see SI Appendix)[19,20]. In the basic version of this model, NC volumes are set only by osmosis and surface tensions are negligible. As expected from a model considering only osmotic equilibrium and membrane tension (Eq. (2) in SI Appendix), hyperosmotic shock induced a substantial shrinkage of both nuclear and cytoplasmic volumes, leaving NC ratio and cytoskeletal forces acting on the nuclear envelope mostly unmodified (Fig. S3a, b). Our model also shows that the same behavior is expected for a non-negligible constant surface tension, with a small correction on the slope, but the expected nuclear volume changes due to external forces are small. Conversely, lowering cytoskeletal forces exerted on NE, as induced in suspended, latrunculin-treated or ROCK inhibitor Y-27632 treated cells (see supporting information), caused a substantial nuclear volume decrease, with only minor effects on the cytoplasmic volume (Fig. S3a, b). Clearly, these biophysical changes also lead to significant variations of the NC ratio (Fig. S3a) contrary to the expectations of a pure osmotic equilibrium (SI Appendix).

## Osmo-mechanical regulation of nuclear volume

Finally, we used the N2FXm at its full potential to measure the volumetric dynamic response at shorter timescales during three kinds of perturbations: hyperosmotic shock, cell detachment and cell spreading. These types of experiments were thoroughly investigated in previous works[22,26]. In particular, the work of Finan and coworkers has the

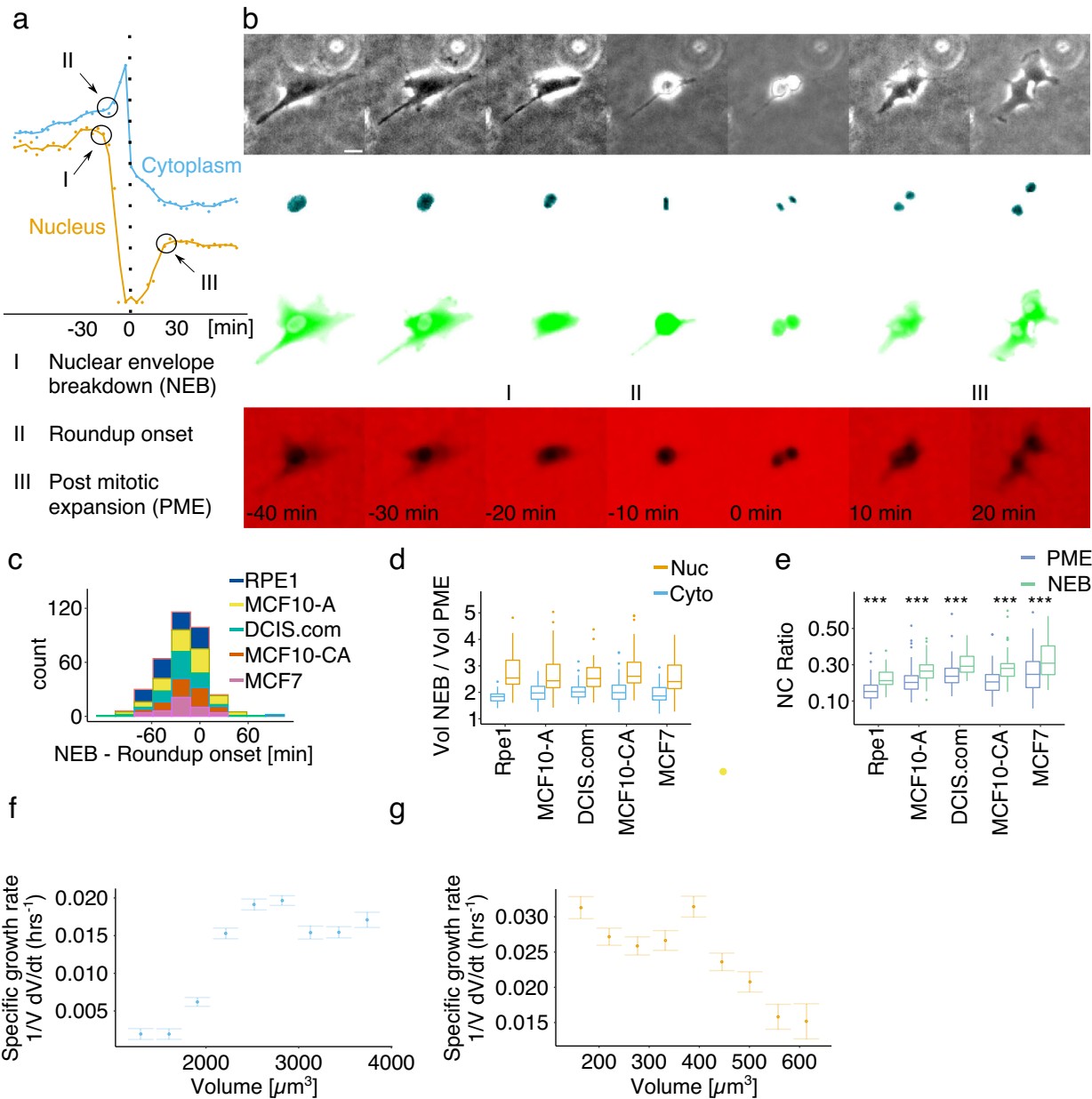

**Fig. 2 | Analysis of nucleus-cytoplasm volumetric coupling across cell division reveals general relations linking nucleus and cytoplasm volumes.**
**a** Experimental detection example of I (NEB), II (roundup onset) and III (PME).
**b** Representative experimental images of an RPE1 cell across division. From top to bottom: transmission light, H2b-BFP, GFP-NES and Texas red channels. Scale bar is 20 μm. **c** Cumulative distribution of delay between NEB and PME for all the 5 cell lines analyzed. **d** Boxplot representing the distribution of the volumetric ratio between NEB and PME for the nucleus and cytoplasm. **e** Boxplot of the NC ratios at NEB and PME. NC ratio increases by 32%, 22%, 20%, 26%, and 23% for the 5 cell lines,

respectively. *P* value of corresponding two tailed paired *t* test: 2.2·e$^{-16}$, 1.0·e$^{-10}$, 3.4·e$^{-8}$, 5.5·e$^{-9}$, 0.0001. Rpe1 cytoplasm (**f**) and nucleus (**g**) specific volume growth rate, defined as the binned average of $\frac{1}{V} \cdot \frac{dV}{dt}$ at fixed volume *V*, plotted as a function of volume (mean ± SE). In (**c**), (**d**) and (**e**): RPE1 *n* = 82, MCF10-A *n* = 88, DCIS.com *n* = 66, MCF10-CA *n* = 66, MCF7 *n* = 42. In (**f**) and (**g**): Rpe1 *n* = 99. In (**d**) and (**e**) boxplots, middle bars are medians, the rectangles span from the first to the third quartiles and the bars extent from ±1.5*IQR. Source data are provided as a Source Data file. "*n*" represents the number of cells examined over at least three independent experiments.

merit of showing that the volume increase of the nucleus upon hypo-osmotic shocks, saturates and does not follow a linear increase as the cell volume does. In our cells subjected to a hyperosmotic shock, both nuclear and cytoplasmic volumes simultaneously decreased without any detectable lag (at the temporal resolution of our experiments, 2 min), and with negligible effects on the NC ratio (Fig. 3a–d), once again confirming the osmotic model. Moreover, the force acting on NE, independently assessed, was slightly affected along hyperosmotic shocks (Fig. 3e). Instead, during cell detachment, both nuclear volume and force on NE decreased, with no considerable effects on

cytoplasmic volume (Fig. 3f–j). Equally, during spreading, nuclear volume and force on NE increased and the cytoplasmic volume decreased (Fig. 3k–o)[27]. Altogether, these results confirmed the osmotic coupling existing between the nucleus and cytoplasm, which coherently reacts to an osmotic perturbation (Fig. 4a). However, they also highlight that external mechanical forces exerted on the NE must be an important regulator of nuclear volume. Indeed, during cell detachment or spreading, or upon cytoskeleton perturbations, nuclear and cytoplasmic volumes are strongly decoupled, with nuclear volume variations coherent with the changes of forces exerted on the NE and

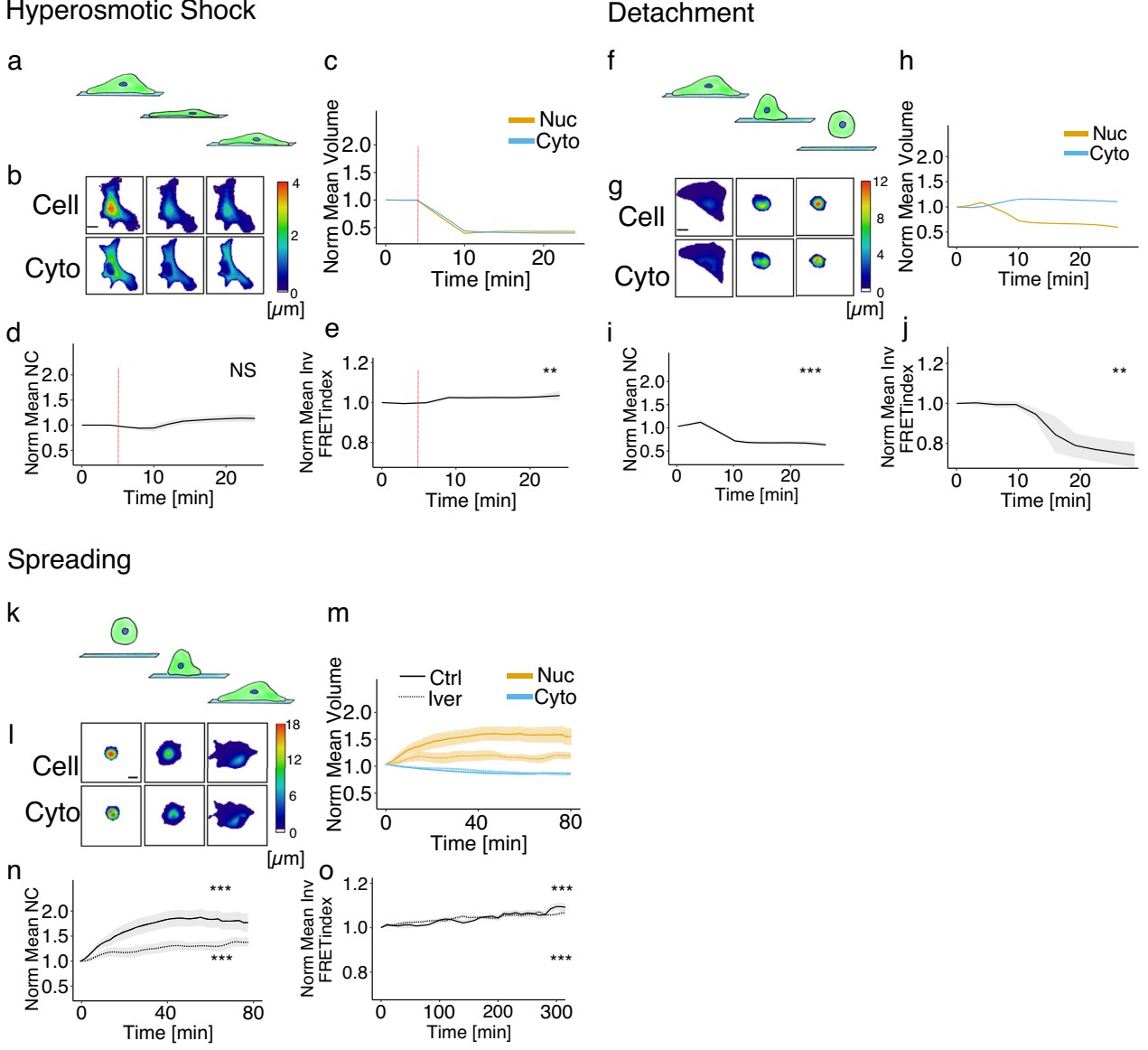

**Fig. 3 | Osmotic, mechanical, and active transport contribute to nucleus-cytoplasm volumetric coupling determination.** Dynamic perturbations: **a**–**e** hyperosmotic shock, **f**–**j** detachment and **k**–**o** spreading. Representative cell and cytoplasmic height evolution of a RPE1 cell during hyperosmotic shock (**b**), detachment (**g**) and spreading (**l**) experiments. Scale bar is 20 μm. Mean (±SE) normalized nuclear and cytoplasmic volume trajectories during hyperosmotic shock (n = 24) (**c**), detachment (n = 23) (**h**) and spreading (**m**). In (**c**), the red vertical line represents the osmotic shock time point. In (**m**), the continuous line is relative to untreated cells (n = 47) while the dashed line is to ivermectin ones (n = 22). Mean (±SE) normalized NC ratio trajectories during hyperosmotic shock (**d**), detachment (**i**) and spreading (**n**). In (**d**), the red vertical line represents the osmotic shock time

point. In (**n**), the dashed line is relative to ivermectin-treated cells, while the continuous line to untreated cells. Mean (±SE) normalized inverted FRET index value during hyperosmotic shock (n = 20) (**e**), detachment (n = 10) (**j**) and spreading (**o**). In (**e**), the red vertical line represents the osmotic shock time point. In (**o**), the dashed line is relative to ivermectin-treated cells (n = 10), while the continuous line to untreated cells (n = 14). Difference in normalized volumetric NC ratio was tested with a two-tailed paired t-test, p values: 0.08 (**d**), 4.05·e⁻⁸ (**i**), 3.14·e⁻⁶ (N, ctrl), 0.0005 (N, iver). Difference in normalized iFRET index was tested with a two-tailed paired t test, p values: 0.0016 (**e**), 0.0012 (**j**), 3.76·e⁻⁵ (**o**, ctrl), 6.31·e⁻⁷ (**o**, iver). Source data are provided as a Source Data file. "n" represents the number of cells examined over at least three independent experiments.

not with the changes of cytoplasmic volume (Fig. 4a). When perturbations are applied to the cell, including changes in the cytoskeletal forces that act on the nucleus, transport rates can be modified, with—in general—different changes for the import and export rates. Thus, while it is true that the naïve osmotic effect of changes in cytoskeletal force are not significant in our experiments, there is an important indirect effect of those forces on the transport rates that determine the protein concentrations in the nucleus and cytoplasm, and hence the osmotic pressure difference and NC ratio. This was a major point of previous studies[7,19,20]. Based on the model and these previous results, we reasoned that the behavior under spreading and detachment was very

unlikely to be a direct effect, as, independently of the direction, compressive or stretching, the force required to change nuclear size would be abnormally large (see SI Appendix)[28]. Instead, recent findings suggest that NE tension, possibly controlling nuclear pore size[29,30], could alter nuclear import rate of small proteins[31,32], potentially affecting osmosis[9].

We explored a model variant where, in addition to the osmotic equilibrium, we added a term reflecting the contributions of the mechano-mediated regulation of nucleo-cytoplasmic transport (Figs. 4b and S3e, and SI Appendix). This model, calibrated with data from ref. 20, suggests that moderate tension changes may affect

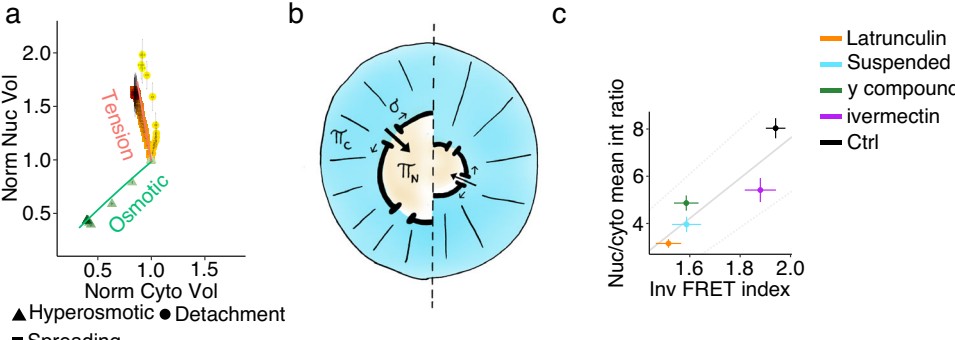

**Fig. 4 | Modeling of the tension-biased regulation of nucleus-cytoplasmic volumetric coupling. a** Scatter plot representing different nucleus-cytoplasm volumetric coupling relationships (variations of nuclear volume as a function of cytoplasmic volume, normalized to resting values) in hyperosmotic shock (green, $n = 13$), detachment (yellow, $n = 20$) and spreading (orange, untreated cells, $n = 40$) experiments (mean ± SE). Here, time progression is coded with a color gradient (from light to dark). The green and orange solid lines are predictions from the theoretical model with parameter values: 20% fraction NLS/NES proteins (in the absence of applied tension), and an external tension-transport bias of $4 \cdot 10^3$ m/N (estimated from ref. 31). The orange line is a prediction for a variation of the tension caused by external forces $\sigma_{ext}$ from 0 to $10^{-4}$ N/m. Reference cytoplasmic volume (for the model) $V_C = 2000 \, \mu m^3$, reference nuclear volume $V_N = 354 \, \mu m^3$. **b** Sketch of the theoretical model. Here, $\pi_c$ and $\pi_n$ represent the oncotic pressure of the cytoplasm and the nucleus, respectively, while sigma is the NE tension **(c)** Scatter plot of the mean value of nucleus to cytoplasm ratio of the GFP-NLS mean intensity in function of the inverted FRET index (mean ± SE) in the different condition considered. Number of cells analyzed per conditions (GFP-NLS ratio/iFRET index): Ctrl 58/156, ivermectin 44/85, y compound 44/119, suspended 39/91, latrunculin 45/129. Source data are provided as a Source Data file. "$n$" represents the number of cells examined over at least three independent experiments.

nuclear size through this effect (SI Appendix and Fig. 4a, b and S3e). Hence, we tested this hypothesis experimentally. We first quantified nuclear import efficiency as a function of the force acting on NE, using as proxy the nucleus-cytoplasm fluorescence intensity ratio in cells expressing GFP-NLS protein. We found that perturbations decreasing the extent of forces exerted on NE always decrease nuclear import efficiency (Fig. 4c). We then assessed volume dynamics in spreading cells treated with ivermectin, an inhibitor of the α/β importins transport pathway[33], at a concentration affecting nuclear import but unable to perturb non-cytoplasmic or nuclear volume, NC ratio or forces acting on the NE (Figs. 4c, S3c, d). During spreading, despite as in untreated cells forces on NE increases and cytoplasm volume decreases, importin impairment led to a significantly slower nuclear volume increase (Fig. 3m–o). Hence, we conclude that the observed force-related volumetric decoupling was mainly linked to a modulation of the nuclear import. This mechanism can tune the nuclear macro-molecular content and thereby controls nuclear volume "amplifying" the sensed forces[9].

## Discussion

In conclusion, the N2FXm method, overcoming common limitations on simultaneous dynamic measurements of cell and nuclear volume in living cells, can be used to decipher fundamental mechanisms of nuclear-cytoplasmic volume coupling. While more statistics might be needed to consolidate these results, our preliminary measurements on five different cell lines (Figs. 2f, g and S2-c) suggest that the volume of single cells grows, on average, at least exponentially (confirming previous reports[15,18,34] and with similar cell-cycle trends as reported in ref. 18), while the nucleus of single cells grows on average clearly at a different pace, sub-exponentially, and, to a first approximation, close to linearly. Whatever the detailed trends, our findings suggest that in cultured single mammalian cells, cytoplasm and nucleus grow, on average, differently. This phenomenon is intriguing but remains unexplained and indeed cell-cycle stage dependency of cellular and nuclear growth law is still to be explored[18]. While our current measurements focus on short time scales and offer coherent explanations for the observed trends, distinct explanations are likely necessary for longer time scales[12]. Rollin et al.[35] recently proposed a mechanism involving counterion release by chromatin folding that may partially account for this trend during G1 phase. More widely, nuclear envelope

stress could have two possible independent effects, which can act on different timescales: one on nuclear import (and osmotic equilibrium), which we focus on in our study, which makes nuclear volume increase when the envelope is more stretched compared to a reference state; and one, that has a direct mechanical effect, and balances the difference of osmotic pressure between the nucleus and cytoplasm, by which increasing tension leads to a decrease of nuclear volume for a given osmotic balance at steady state. In our study, due to cell spreading, we can hypothesize that we are in a regime of low effective constitutive tension and the import effect dominates. A comprehensive understanding of how the combination of these complex opposite effects determine nucleus volume growth during the cell cycle would require further experiments and a clearer picture of the underlying mechanisms. Despite these wider considerations, results suggest that nuclear size homeostasis may be related to a sophisticated mechano-sensing mechanism biasing nuclear transport. Indeed, forces exerted by the cytoskeleton, affecting NE tension, regulate nuclear-cytoplasmic transport, eventually affecting osmosis[31]. Accordingly, specific transcription factors such as YAP, key regulator of organ growth and regeneration as well as of mechanotransduction, could also be affected by this mechanism[32], establishing intriguing links between biophysical and regulatory pathways to be explored in the future. Finally, nuclear volume regulation, controlling the concentration of nuclear components and translocation of transcription factors, impacts chromatin state and nuclear organization, broadly affecting cell activity.

## Methods
### Cell culture
All culture media were supplemented with 1% L-glutamine and 1% penicillin-streptomycin. hTert RPE1 ATCC cells were cultured in DMEM-F12 supplemented with 10% FBS. MCF10-A ATCC and MCF10 DCIS.COM (in the text and figure named simply as DCIS.com) cells (Barts Cancer Institute, Queen Mary University of 924 London, UK) were cultured in DMEM-F12 supplemented with 5% horse serum, 10 μg/ml human insulin, 0.5 mg/ml hydrocortisone and 20 ng/ml epidermal growth factor (EGF). For MCF10-A cells, we used the same medium of DCIS.com, further supplemented with 100 ng/ml of cholera toxin. MCF10-CA cells (Barts Cancer Institute, Queen Mary University of 924 London, UK) were cultured in DMEM-F12 supplemented with 5% horse

serum, 1.05 mM calcium chloride and 10 mM HEPES. MCF7 NCI cells were cultured in RPMI 1640 medium. All the cell lines were grown at 37 °C and in humified atmosphere with 5% $CO_2$.

## Cell transfection, lentiviral production, and cell transduction

Lentiviral particles were produced as described here[36]. Briefly, $3 \times 10^6$ HEK293T cells were transfected with pCDH lentiviral plasmids encoding H2B-BFP or 2X_NES-GFP (nuclear exportation signal), or MiniNesprin1_FRET sensor, together with psPax2 and pMD2.G vectors with a ratio of 4 μg: 2 μg: 1 μg using Polyethylenimine (PEI) as transfection reagent. 24 h later the medium was removed, and virus collection performed after 24/48/72 h. Viral aliquots were pulled together and filtered with 0.45 μm filters. $2 \times 10^5$ cells were transduced with 1 ml of fresh virus supplemented with Polybrene (8 μg/ml) through 20' of centrifugation at 2000 rpm, then seeded in six-well plates.

Cell transfection of hTERT_RPE1 cells was performed using the Neon Transfection System (Thermo Fischer Scientific) following the manufacturer's protocol. Expression of nuclear GFP was obtained through a plasmid encoding an NLS (nuclear localization signal) tagged eGFP (Addgene #67652).

## N2FXm chip fabrication

The N2FXm chip is a microfluidic device composed by two "acquisition" chambers connected by "loading" channels (Fig. S1a). Acquisition chambers are 18 μm in height and used to image cells for volume measurements. Reservoir channels are higher (~150 μm) and ensure a continuous nutrient delivery to cells in the acquisition chambers (Fig. S1a). Microstructured silicon mold was used to create the N2FXm chip. The template wafers were fabricated at the cleanroom facilities of the Binning and Rohrer Nanotechnology Center (BRNC) using a standard two step-photolithography process[37]. First, the pillars in the central chamber were etched until a final depth of 18 μm. Finally, the lateral channels were added by crosslinking a 150 μm thick layer of SU-8 resist. The resulting silicon wafers were passivated as follow: soaking for 30 min in a silanization solution I (Sigma-Aldrich, USA), then 10 min in hexane (Sigma-Aldrich, USA), 10 min in 1-octanol (Sigma-Aldrich, USA), and finally rinsed with acetone and deionized water. (Silicon master fabrication). The microfluidic chip was then fabricated in PDMS (Sylgard 184) through standard replica molding. Briefly, the PDMS precursor was mixed with the crosslinker (10:1) and poured on the silicon mold, degassed for 1 h in a vacuum bell and then cured for 3 h at 90 °C. Once demolded, 1.5 mm diameter punches were made in correspondence with the inlet and outlet of the chip. The chip was then treated for 1 min with oxygen plasma and irreversibly bonded to a 35 mm bottom-glass petri dish (Mattek). To reinforce the bonding treatment, the bonded device was heated for 30 min at 40 °C. To prevent medium leakages, two PDMS cubes (height ~3 mm) were punched (2 mm diameter) and bonded in correspondence of the inlet and outlet. The day before the experiment, the chip was retreated for 1 min with oxygen plasma, filled with fibronectin at the appropriate concentration (10 μg/ml for RPE1 cells, 20 μg/ml for all the other cell lines) and incubated at RT for 1 h. The chip was then rinsed with PBS, filled with the appropriate medium and incubated overnight at 37 °C. The day of the experiment, cells were injected into the chip at ~$300 \times 10^3$ cells/ml. The petri dish was humified (filled with medium) to prevent medium evaporation from the chambers during the experiment. For all the experiments excepting spreading, after 3–6 h from cell injection (the time necessary to ensure a good adhesion of a specific cell line), the culture medium was replaced with a medium containing 70 kDa Texas Red Dextran at 1 mg/ml (Thermofisher Scientific) and, after additional 3 h, the microfluidic system was imaged at the microscope. In the case of spreading experiments, cells were instead directly resuspended in the Texas Red Dextran-containing medium, injected in the chip and imaged ~15 min later.

## N2FXm for nuclear volume measurement

Cell and nuclear volumes were measured imaging GFP-NES and H2B-BFP positive cells in the N2FXm chip filled with a fluorescent dye (Dextran Texas red 70 kDa Neutral, Invitrogen) not internalized by cells (see "Chip fabrication" section, Fig. S1a). We acquired images in four channels: transmitted light, red (dextran), green (GFP-NES) and blue (nuclear staining). Images were analyzed using ImageJ. First, dextran field illumination homogeneity was corrected dividing the original image by a correction/normalization image, which was obtained by dividing the smoothed original image (or the maximal projection or multiple frames, in case of acquisitions longer than one time point) for its mode. Here, the smoothing was obtained by applying a median filter with a radius value ranging from 35 to 50, which corresponded to the minimum value needed to eliminate the contribute of the pillars (i.e., that appeared as black circles in the image) from the mode calculation of the dextran channel. Then, as in Zlotek-Zlotkiewicz et al. a drop of dextran intensity in correspondence of cells was used to calculate cell volume[13]. Any object in the chamber, occupying a specific space, excludes the red fluorescent dextran from this volume. Dextran fluorescence intensity can be scaled between two points of known height: zero, where the chamber is empty and the maximum, the known chamber height (18 μm), in correspondence of the pillars sustaining the chamber roof (as schematically shown in Fig. S1b). This calibration allowed defining the optical thickness of cells in the chamber at each pixel and was performed independently for all fields of view and all time points. Although optical thickness at each pixel can differ from cell height, its integration over a slightly enlarged cell area, here automatically defined from the GFP-NES fluorescence signal, well quantify the cell volume (Fig. S1c–e)[13].

For measuring nuclear volumes, a second calibration was introduced. Cell nuclei were negatively stained with the ectopic expression of a green fluorescent protein coupled with a nuclear export signal (GFP-NES), which localizes in the entire cytoplasm except the nucleus, also marked with H2B-BFP (see "Cell transfection, lentiviral production and cell transduction" section). Since we assume that the intensity of the GFP signal is linearly proportional to the volume of the corresponding portion of cytoplasm, it can as well be calibrated according to the one of calibrated dextran. The GFP calibration was then performed by fitting the conditional average of GFP intensity in the cytoplasm (all cell excluding the nucleus) at fixed optical thickness (Fig. S1c–e). Cytoplasm and nucleus were automatically defined by combining the GFP and BFP signals (Fig. S1e). Linear fitting was carried out in R using the "robustbase" package [http://robustbase.r-forge.r-project.org/]. Robustbase: Basic Robust Statistics. R package version 0.93–9. Importantly, to correct for possible signal fluctuation or variation of GFP expression, GFP calibration was performed at each timepoint and for each analyzed cell. In order to strengthen the calibration, we also introduced the possibility to perform and mediate it over a variable number of timepoints (nzz). Volume trajectories relative to asynchronous cells along the cell cycle (Figs. 1d–f and S1m) were obtained by setting nzz=3 (see Supplementary Movie 1). Spreading, detachment and hyperosmotic shock dynamic experiments were analyzed with nzz = 1.

Finally, to measure nuclear volume, the calibrated GFP signal was integrated within the nuclear region, and the resulting value subtracted from cell volume was evaluated over the same domain (see Methods and Fig. S1e). Cytoplasmic volume was calculated by subtracting the nuclear volume from the entire cell volume and the NC ratio as the ratio between nuclear and cytoplasmic volumes. All the experimental data were pulled from at least three independent experiments.

## Live imaging experiments

All live imaging experiments were performed at 37 °C and 5% $CO_2$ atmosphere. FXm experiments were performed using a LEICA

widefield DMI8 inverted system equipped with a HC PL Fluotar 10x NA = 0.32 (Leica, #506522) objective. The excitation source was an LED illumination. Images were acquired with a sCMOS Andor Neo 5.5 camera.

FRET experiments and 3D reconstruction were both performed with an inverted LEICA confocal SP8 equipped with a HC PL APO 40x, NA = 1,30 OIL immersion (Leica, #506358) or a HC PL APO 63×, NA = 1,40 OIL immersion (Leica, #506350) objective, respectively.

GFP-NLS nuclear-cytoplasmic localization was obtained with an inverted spinning disk (CSU-X1 Perkin Elmer Ultraview) Nikon Ellipse Ti, equipped with Nikon PLAN Fluor 40 ×1.3 OIL objective and a Hamamatsu EM-CCD C9100 camera. All the experimental data were pulled from at least three independent experiments.

## DAAM particle measurement and N2FXm precision

Poly-acrylamide-co-acrylic acid particles (DAAM particles) were used for evaluating the precision of our technique. In particular, if internalized by the cell, DAAM particles are excluded by the cytoplasmic staining (while increasing the total cell volume) and their volume could be then measured using our technique. DAAM particles were preferred to glass beads because the refractive index (RI) of the latter (RI > 1.47) was too different from that of the water (RI ~ 1.33) and generated optical artifacts, while DAAM RI ranged between 1.33 and 1.34[16]. DAAM particles were produced as previously described with a mean diameter of 3.5 μm[16]. DAAM particles were resuspended at $50 \times 10^6$ beads/ml in the RPE1 cells medium and injected in the acquisition chamber containing adhered RPE1 cells. After 4 h from particle injection, culture medium was replaced with Texas red dextran (1 mg/ml) containing medium. To estimate the error, volumes of the internalized particles were quantified with our technique (e.g., measured volume) and compared with the ones obtained through a geometrical calculation (e.g., expected volume) (Fig. S1g–i). Here, because of the particle spherical shape, expected volumes could be calculated measuring particle diameter (averaged between two measurements). This procedure was performed manually and not by an automatic measurement.

Moreover, we compared nuclear volume distribution measured with the N2FXm with the one obtained by means of confocal 3D reconstruction. 3D reconstruction was performed by imaging RPE1 cells in the same culture condition used for the N2FXm experiments (see "chip microfabrication" section). Cells were then acquired along z with a step size 0.3 μm. The stacks were then analyzed with the 3D object counter plugin of ImageJ. From this analysis, we did not find statistical differences between the volumes obtained with the two methods (Fig. S1f). All the experimental data were pulled from at least three independent experiments.

## Drugs, adhesion and osmotic perturbations experiments

All the drug experiments were performed for both volume (N2FXm) and nuclear tension (FRET) measurements. For N2FXm, drugs were added at the proper concentration to the specific medium (containing the fluorescent dextran) and injected in the device containing adherent cells (as described in the "Chip fabrication" section). In GFP-NLS localization experiments, all the condition were explored except for hyperosmotic shock.

For perturbing cytoskeleton and cell mechanics, latrunculin-A (LatA, Sigma Aldrich) and Y-27632 (Y27, Sigma Aldrich) were added to the culture medium at 1 μM and 500 μM, respectively. Cells were then imaged after 1 h (for LatA) or 12 h (for Y27). For hyper-osmotic shock experiments, D-sucrose (Carlo Erba reagents, Dasit Group) was resuspended in pure water at 500 mM and, to perform dynamic measurements, added to cells during imaging. Detachment experiments were performed by adding EDTA to a serum-free culture medium at (0.05%). As in the case of osmotic shock experiments, EDTA containing medium was added during imaging for obtaining dynamic measurement.

Spreading experiments were conducted by imaging cells ~15 min after seeding. Here, untreated (control) cells and cells pre-treated with ivermectin (Sigma Aldrich, 10 μM for 24 h) were used.

In N2FXm dynamic experiments (i.e., hyper-osmotic shock, detachment and spreading), cells were acquired each 2 min (see Supplementary Movie 1). In FRET dynamic measurements cells were acquired each 3 min during osmotic shock and detachment, while during spreading each 10 min. The lower acquisition rates used for spreading FRET experiments were justified by the need to perform z-stack acquisitions (z step size = 1 μm) for accounting cell movements along the z axis. All the experimental data were pulled from at least three independent experiments.

## Curve generation and point detection

All the figures and statistical analysis were performed in R. Packages used were: "robustbase", "ggplot2", "gridExtra", "rcarbon", "ggsignif", "rcarbon", "rsvg".

All the plotted dynamic datasets were smoothed using the "runMean" function ("fill" smoothing option on 3 points). In all the smoothed datasets, eventual NA was removed using the "na_interpolation" (linear option) function.

To generate mean volume and ratio trajectories of non-synchronized growing cells (Fig. 1e, f and S1m), single curves were first temporally aligned and then mediated over time. Here, cytokinesis was used as reference time point (represented as the time "0 min"), which was manually identified for each cell analyzed from the acquired images.

Nuclear envelope breakdown (NEB), postmitotic expansion (PME) and roundup onset points relative to single-cell trajectories were automatically detected (Figs. 2a and S2a). The detection was performed as follows:

**NEB.** This point was detected by combining the first and the second differential of smoothed nuclear volume data (runMean function, "fill" smoothing option on 3 points). In particular, the NEB was defined 1 point before the last positive value of the first differential, which was considered in a 10 points wide window preceding the maximum value of the second differential before division. The second differential was analyzed in a 15 points window before division.

**PME.** Five points successive to the minimum value of the second differential of smoothed nuclear volume data (runMean function, "fill" smoothing option on 15 points or on the maximum length of the nuclear trajectory after division, if the latter was lower than 15), considering a 15 points wide window after division.

**Roundup onset.** 2 points before the maximum value of the second differential of the smoothed cytoplasmic volume data (runMean function, "fill" smoothing option on 15 points or on the maximum length of the nuclear trajectory before division, if the latter was lower than 15), considering a 15 points wide window before division.

All the detected points were manually checked.

To generate the normalized dynamic mean curves relative to volumes, NC ratio and inverted FRET index (Fig. 3c–e, h–j, m–o) during spreading, detachment and osmotic shock experiments, single curves were first smoothed by means of the runMean function ("linear" smoothing option on 3 points). Successively, data relative to each curve were divided by their first value to be normalized. Finally, all the curves were mediated over time. All the experimental data were pulled from at least three independent experiments.

## Growth speed analysis

Cytoplasm and nuclear growth modes were obtained by analyzing the specific volume growth rate ($1/V \cdot dV/dt$) in function of the volume (V). Here, because of the already reported high variability of single-cell

trajectories[18], growth modes were evaluated considering the population averages, and in particular the trend of the conditional averages. In this analysis, the two limit cases are represented by the linear and the exponential growth modes, characterized by a constant/increasing or decreasing value of the specific growth rate in function of the volume, respectively.

First, single cell volumetric data (i.e., nucleus and cytoplasm volumes used for obtaining the mean volumes trajectories of Figs. 1e and S11) were smoothed with a central average algorithm considering a sliding window of 20 frames. To calculate the specific growth rate ($1/V \cdot dV/dt$), a linear fit of single-cell volume curves as a function of time were performed on sliding windows of 12 frames. To get the average population behavior, growth rates were mediated within 9 volumetric intervals defined to cover the entire volumetric dynamic range. All the experimental data were pulled from at least three independent experiments.

## Nuclear envelope tension analysis and synthesis of a novel cpst-FRET sensor

The extent of cytoplasmic forces acting on the Nuclear Envelope (NE) was measured exploiting a novel circularly permutated stFRET (cpstFRET)[21] sensor. The sensor has been built using Nesprin 1 protein as backbone and named Mini-Nesprin 1. A detailed explanation about how this construct was created is reported here[25]. Briefly, N-terminus Nesprin 1 (1521 bp, aa 1- 507) and C-terminus (1422 bp, aa 8325-8797) coding sequences containing Nesprin 1 Calponin (CH-CH) and KASH domains respectively were synthesized exploiting IDT gBlocks Gene Fragments Technology. A circularly permutated stFRET (cpstFRET) probe was inserted between the two blocks. Final coding sequence of Mini Nesprin1 cpstFRET sensor was obtained through molecular cloning and inserted into a pCDH lentiviral plasmid. Image acquisition was performed using a Leica TCS SP8 confocal microscope with microscope with Argon light laser as excitation source tuned at 458 nm and HC PL APO CS2 ×40/1.30 oil-immersion objective. Emission signals were captured at 490–510 nm and 515–535 nm, and inverted FRET index calculated as their ratio. Data were analyzed using in-house ImageJ macro.

## GFP-NLS localization analysis

Nucleoplasmic transport was evaluated by quantifying the nucleus on cytoplasm mean intensity ratio of GFP-NLS protein. All the images were analyzed with ImageJ. Here, higher ratios indicate higher active molecular nuclear import. All the experimental data were pulled from at least three independent experiments.

## Reporting summary

Further information on research design is available in the Nature Portfolio Reporting Summary linked to this article.

# Data availability

All data are available in the main text or the Supplementary Information file, or from the corresponding author upon request. Source data are provided in this paper.

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

## Acknowledgements

The authors would like to thank IFOM Imaging Facility for help with performing experiments and the clean room facilities of the Binning and Rohrer Nanotechnology Center (BRNC) for contributing in mold fabrication. We would also like to thank Clotilde Cadart, Sylvain Monnier, Alessandro Marcello, Kristina Havas Cavalletti, Nils Gauthier, Ylli Doksani and Giorgio Scita for helpful comments and discussions. We acknowledge Gilda Nappo (gilda.nappo@gmail.com) for the illustrations. This project was supported by Italian Association for Cancer Research (AIRC): Investigator Grants (P.M. #24976 and M.C.L. #23258) and individual fellowships (F.A.P. #23966 and OMR #*22419*). A.P. was founded by Fondazione Umberto Veronesi Post-doctoral fellowships (#000359) and Short-EMBO Fellowship (#8386).

## Author contributions

FAP was responsible for the project. FAP, AP, and EP designed, carried out, and interpreted experiments. AP developed the miniNesprin1 FRET sensor. FMP, under the supervision of AF, participated in the design and produced the microfluidic measuring chamber. SL, under the supervision of IR, performed cellular biology experiments. MC and AP designed and cloned the plasmids. DV provided DAAM particles and helped design experiments. MP and MCL designed and interpreted experiments. MCL and OMR developed the mathematical model. FAP carried out images, data and statistical analysis and wrote the paper with PM and MCL. PM supervised the study.

## Competing interests

The authors declare no competing interests.
