## [Peer Review File · Nature Communications]

REVIEWER COMMENTS

Reviewer #1 (Remarks to the Author):

This paper experimentally explores the effects of various biochemical and mechanobiological perturbations of single cells on the ratio of their nuclear to cytoplasmic volumes. It is shown that changes in cytoskeletal forces exerted on the nucleus (via latrunculin or ROCK inhibition) changes the NC ratio in a manner that is not simply due to the direct effect of those forces on the osmotic pressure balance. In addition, it is shown that while the NC ratio in many cell types (under hyperosmotic conditions) is constant during most of interphase, this ratio varies considerably during cell division. Interestingly, it is found that right after cell division, the cytoplasmic volume increases exponentially, while the nuclear volume increases only linearly with time. The results will stimulate further experiments and theoretical models and should be published in Nature Communications, pending the following changes.

We believe that the authors overstate the impact of their results with respect to the concepts presented in Refs. 7, 16, and 17. In particular, the present manuscript uses the term “falsify” (once) “or “falsified” (twice) to relate their findings to the presumably naïve “osmotic” models in these references. This, besides being polemical, is not correct and does not reflect the subtlety of the ideas previously presented. While the NC volumetric ratio is indeed determined by osmotic pressure differences, the proteins involved are those whose active transport through the nuclear pores, results in differing concentrations in the cytoplasm and nucleus. The import rate and export rates can be different for systems with active transport, resulting in concentrations of proteins that are directly related to these rates. When various perturbations are applied to the cell, including changes in the cytoskeletal forces that act on the nucleus, those transport rates are modified, with – in general – different changes for the import and export rates. Thus while it is true that the naïve osmotic effect of changes in cytoskeletal force are not significant in the experiments described, there is an important indirect effect of those forces on the transport rates that determine the protein concentrations in the nucleus and cytoplasm, and hence the osmotic pressure difference and NC ratio. This was indeed a major point of these references which seems to have been missed in the present manuscript.

Nevertheless, the paper is interesting and presents several novel results about how the NC ratio changes during the cell cycle and under biochemical or mechanobiological changes, which justify publication in Nature Communications. However, before we can formally recommend publication, the interpretation of these changes with respect to such perturbations should be rewritten to accurately incorporate the previously discussed changes in osmotic pressures due to active transport and to qualitatively relate these transport changes to the perturbations applied in the present manuscript. In addition, we request the authors to address the following major and minor comments.

Major comments:

1. At the bottom of page 5, the authors write “Our model also shows that the same behavior is expected for a non-negligible constant surface tension...”. Compared to what scale is the surface tension non negligible? The work by Finan et al. (Annals of Biomedical Engineering, 2009) presents the notion of NE that is stretched compared to one that is relaxed. The authors should relate the findings of their model with respect to the work of Finan et al.
2. The authors conducted three types of experiments to modulate the forces that are exerted on the cell and are transduced into the nucleus: hyperosmotic shock, cell detachment, and cell spreading. These types of experiments were thoroughly investigated in previous works such as Finan et al. (mentioned above), Guo et al. (ref. 19 in the manuscript), and others. However, discussion that relates the results of the manuscript to these previous works is lacking. The authors should present their experimental results in a wider context and relate them to these previous works.
3. At the bottom of page 37, the authors relate the tension to external force by the equation $F_{ext} = 8\pi RN \sigma_{ext}$. The authors should explain this relation as its derivation is unclear. Perhaps a cartoon showing the force, tension, and the relation between them can make the derivation clearer.
4. The authors model the effect of force on nuclear import using an exponential relation. The author should discuss the underlying biophysical reasoning of this dependence or state it is just a fitting function.
5. In their theoretical model, the authors treat the external force as a scalar although by definition force is not a scalar. Furthermore, the authors assume that the nucleus is spherically symmetric, which is not true in general, when deformed by non-isotropic external forces (that may have shear components).
6. The authors state that they assume the forces that are exerted on the nucleus are compressive. In that case and when the forces are spherically symmetric, the contribution of the external forces to the tension should be negative. Is this the case? This should be clearly explicated since this affects the way the exponential relation between the surface tension and nucleocytoplasmic ratio is interpreted.

Minor comments:

1. Can there be forces that are exerted on the nucleus which will not be detected by stretching of the LINC complex (e.g. squashing of the nucleus)?
2. Some works (e.g. Jahed et al., Journal of Cell Science 2016) suggest that the cytoskeleton and LINC complex are indirectly involved in nucleocytoplasmic transport by modulating the Ran protein gradients. Is it possible that the nucleocytoplasmic transport in the latrunculin-treated or the ROCK inhibited cells is

affected via this pathway rather than mechanical “expansion” of the NPCs? Can the author comment on this possibility.

3. At the bottom of page 6 the authors write that the cytoskeletal forces are compressive. However, these can also be expansive if the nucleus is laterally stretched. Can the authors provide a reference to a work that distinguish between these two possible modes of nuclear deformation?

4. The authors mention that the estimate of the NE tension (10^{-6} N/m) is a lower bound due to fluctuations of active origin of the NE shape. Are there evidence for such active fluctuations? While this does not change the conclusion of the authors, it is worthwhile to give a reference for such active processes that facilitate NE shape fluctuations.

Reviewer #2 (Remarks to the Author):

Pennacchio et al extended the powerful fluorescence exclusion microscopy, FXm, to simultaneously measure cell volume and nuclear volume – a new method they coined “N2FXm”. This novel method is sound and allowed the authors to study the nuclear-to-cell volumetric (NC) ratio. First, they found that the NC ratio was not constant during the cell cycle, being particularly impacted by mitosis. Second, they showed that the cell and nucleus did not follow the same growth laws. Third, they discovered that the NC ratio could not be explained solely by an osmotic balance and that cytoskeletal forces exerted onto the nuclear envelope could bias nuclear import and nuclear osmotic pressure, thus also participating in setting the NC ratio.

The manuscript is clearly written, and the data are of high quality and interest, shading light on novel regulatory mechanisms of nuclear volume. I would greatly recommend publication should the following questions be addressed.

1/ The N2FXm method

I found the N2FXm method extremely powerful and relevant. Although most of my concerns are addressed in the methods or in the paper, I have one additional question and one recommendation for the authors:

i) Have the authors checked that there is no GFP-NES in the nucleus, and if the signal is homogeneous in the cytosol? Depending on the cell state/strength of the NES, there may still be some GFP in the nucleus, which would affect the measurement – especially if this bias is cell cycle-dependent, through for instance NE tension impacting nuclear export. Even if non-significant, there seems to be a small difference in the

nuclear volume measurement between the 3D reconstructed and the N2FXm in Fig. S1F. Moreover, this could also be important in the perturbation experiments, impacting the actual measurement.

ii) I believe one important point is that the GFP calibration is done at every time point and for each cell, thus limiting temporal fluctuations or cell-cell heterogeneity in GFP concentration. As such, it may be better to directly state it in the main text, and not only in the methods.

2/ Nuclear growth and NC ratio

I have a few questions regarding nuclear volume growth and subsequent NC ratio values, as well as their interpretations based on the model proposed by the authors:

i) The NC ratio seems rather constant in each cell line from -500 min to 0 min. First, can the authors provide a y-axis zoom before and after mitosis, to have a better visualization (Figs. S1L & 1F)? Second, it seems constant even though nuclear volume does not increase exponentially. Can the authors comment on this point?

ii) Can the authors speculate (and potentially expand the discussion on this point) as to why the nucleus would grow linearly in time? Per the authors' model, if it does not increase exponentially, this could for instance mean either that import/export is cell cycle-dependent such that nuclear osmotic pressure is not constant over the cell cycle. This could for instance be the case if NE tension changes during the cell cycle (see the question on tension in point 3/ below).

iii) Related to the previous point, it does seem that, after the spreading phase (phase III), there is no or little growth of the nuclear volume, followed by more rapid growth. This seems true for the cell lines in Fig. S1L. Has this been considered in the calculation of volumetric growth? Could it be that the binning hides different growth regimes of the nucleus during the cell cycle?

iv) Finally, can the authors speculate as to why the nucleus would decrease by roughly 2.5 times its volume at mitosis? This suggests that it grows "more than necessary", and that this could be linked with cell cycle-dependent growth regimes.

3/ Force-biased import and nuclear envelop tension

I find the conclusions of the authors that the force-biased import plays a role in regulating nuclear volume very appealing. I have a few questions regarding the experiments and the model:

i) Since the measurement of NE tension is not a standard method (and even though Ref. 22 is in press, it is still unpublished), maybe the authors should give more details about this method in the main text.

ii) Related to point 2/ above, have the authors checked how is NE tension regulated during the cell cycle? Could tension explain why nuclear volume increases linearly?

iii) Related to the hyperosmotic experiments: although at first I was satisfied with the result, I am now a bit lost. It is known that a hyperosmotic shock rapidly induces changes in the cytoskeleton (see for instance Thirone, 2009, PMC5047760). As such, there should be an effect on NE tension, and by the authors' model, an effect on the nucleus volume, but it does not seem to be the case. Can the authors comment on this point? Maybe the effect is only on the cortex and not translated to the NE envelop?

iv) Per the authors' model, it is my understanding that growing cells on substrata of different rigidities should impact the NC ratio. I did not find anything on this point in the literature. Do the authors know something about this?

v) Related to the mathematical model: I appreciated the model and its explanation/parametrization strategy. I had one question on the choice of the exponential dependence of the nuclear concentration of macromolecules with external stress: why this choice? Why not linear in the first place? Especially because it seems that it is linearized in the equation defining V_n at the bottom of p. 5.

Minor comments

i) Figures

- adding the legend (control vs. ivermectin) on Fig. S3D would help the reader
- which condition is ivermectin in Fig. 4C? in general, what are the conditions in this plot?
- maybe ivermectin could be its own subfigure on Fig. 4 to help the reader and avoid going back to Fig. 3
- Y-axis in Fig. S2B: unit should be $\mu\text{m}^3/\text{h}$ and not μm^3

ii) Although the manuscript is nicely written, I found several typos, in particular in the appendix. I may have missed some, but here is a list of those I found:

- in methods section "cell transfection", problem unit polybrene
- in methods section "N2FXm", GFP-nes instead of NES
- in methods section "DAAM particle measurement", typo on "technique"

- in appendix, “left” and “right” hand side are inverted I believe
- in appendix, “nucleus surface tension and e contribution” → a
- in appendix, page 2, $c_{ions_out} c_{out}$ instead of just c_{ions_out} (I believe)
- in appendix, section “estimates”, in order TO confirm, prOessure
- in appendix, section “force biased”, there is a verb lacking in first sentence
- p. 6 repetition “the the”

Reviewer #3 (Remarks to the Author):

This manuscript describes experiments and modeling of the regulation of nuclear size by osmotic forces and mechanical properties of the nuclear envelope in cultured mammalian cells. The study of the NC ratio in mammalian cells has been hampered by a lack of accurate volume measurements. This manuscript introduces an elegant new method for estimating nuclear/ cytoplasmic volume using a fluorescence exclusion-based approach, as well as a new FRET sensor for measuring nuclear envelope tension in cells. The work investigates NC ratio changes through the cell cycle and upon perturbations such as osmotic shock and cellular detachment from the substrate. The data support a mechano-osmotic model that implicates osmotic pressure and nuclear transport as critical components for nuclear size control.

In general, this work has potential to be an important advance in the field, but it is currently too preliminary for publication. There are many significant concerns described below. These include insufficient characterization of the new methods and a general tendency for over interpretation leading to inappropriate conclusions. They present some interesting initial observations without establishing sufficient context or additional insights. Some limited experimental data and careful rewriting will be needed to improve this manuscript.

Major comments

1. The title, summary and parts of the abstract are not appropriate as key statements are not sufficiently backed by the data in this paper. For example, the Summary statement is: "Cytoskeletal forces exerted on the nuclear envelope impact on nuclear volume through modulation of force-coupled nucleo-cytoplasmic transport. However, there is little in this paper on cytoskeletal forces or nuclear transport. The closest test is an effect with ivernectin. However this result does not directly show that nuclear

transport rates have changed. There is also no clear data on cytoskeletal forces on nuclear import; small effects with LatA are shown, but these may be highly pleiotropic and cannot be just assumed to be actin effects on the nuclear envelope. Their statement in the last paragraph "forces exerted by the cytoskeleton, affecting NE tension, impact on nuclear pores size (26–28)" have not been conclusively demonstrated in these cited papers and have not been proven here. While the authors certainly may hypothesize upon force-effects on nuclear transport and pores, these statements should not be the central conclusion of this paper. I think that the authors should address these concerns by judicious rewriting the wording of their conclusions, rather than being asked to perform the large number of experiments needed to support claims.

2. There are important concerns about the N2FXm method that need to be addressed. Additional data and more detailed descriptions of the calibration method are needed so that the accuracy of the method can be evaluated (see below). The authors statement that the GFP "localizes in the entire cytoplasm except the nucleus" is not true. For example, other organelles besides the nucleus may also exclude the cytoplasmic GFP; lysozymes, mitochondria, lipid droplets etc., might represent 2-10% of cellular volume. This issue and how it may affect NC ratio measurements using N2JXm certainly needs to be addressed. Can the authors determine if these organelle volumes significantly affect the NC ratio measurements? Another concern is whether NES-GFP is truly all excluded from the nucleoplasm. Can the authors include a control using high-resolution confocal sections to show that there is no significant nuclear signal? If there is some nuclear signal, could they determine how that would affect the accuracy measurements? They should also show whether the localization of the NES-GFP marker is affected by their perturbations. For instance, mechanical or drug perturbations could cause entry of the NES-GFP in the nucleus, which could potentially affect their measurements and conclusions.

3. There are also concerns about the FRET-based nuclear envelope force sensor. This is a new sensor developed by this group, and its development is described only in this work and in a preprint. However, neither manuscript presents sufficient background characterization of this probe to demonstrate that it is acting as a force sensor at the nuclear envelope. Although the handful of data points are consistent, more quantitative and systematic controls would give more confidence that this sensor indeed measures force at the nuclear envelope.

4. One of the most surprising results is the linear growth of the nucleus vs. exponential growth of the cell volume. Other than pointing out the difference, this is not further investigated. More data are needed to confirm and develop this observation. One simple prediction is that NC ratios systematically fall as cells grow larger during interphase (however, this is not apparent in Figure 1F). Could this difference arise from some systematic error in the N2FXm measurements? The reader is left without any context on how to think about this linear growth rate, or whether it carries much significance.

5. Another result is the apparent loss of nuclear volume after mitosis. Has it been confirmed by other methods or in other papers? Could the results be explained by the behavior of the NES marker at these

cell cycle stages? If NC ratios are low after mitosis and then fall over interphase because of linear growth, then how does the nuclear volume ever catch up? How are NC ratios then maintained over multiple cell cycles? In Figure S2A, there is an example where nuclear growth suddenly speeds up after 50 frames (what is the frame rate?). Is this transition typical (as it is not seen in the averaged data)? Does it correspond to a known cell cycle transition? In general, these data are not discussed sufficiently in context to what is currently known about these processes. If the authors hope to describe thoroughly the NC ratio changes over the cell cycle, these issues should be clarified. Overall, the measurements of nuclear volume just before and after mitoses in Figure 2 represent a weak point in the manuscript and could even be deleted (or placed in another manuscript).

Specific Comments

p.3 and Methods

More details of the calibration methods and evaluation of the accuracy of the approach are needed. First, can the authors provide an experimental version of Fig S1C (top right) to demonstrate linearity with the Texas red dextran marker they use?

Second, more detailed descriptions of how the NES-GFP intensity is calibrated with the cell volume data is needed. Can the authors show in a figure a real-life example (not just schematics)? For calibration, more details are needed on what cytoplasmic portions of the cell is used for comparison? How well do the red and green intensities inversely correlate; can the authors add plot showing correlation of green and red intensities per pixel? How is the noise in the intensity measurements dealt with for?

Inhomogeneity in the cytoplasm could have a large impact on the interpretation of these measurements. Can the authors test whether inhomogeneities are significant enough to affect the calibration and subsequent measurements.? For example if there many lysosomes in one part of the cytoplasm, is that portion of the cell not used for calibration?

L.7 p3 , L.6 p4 & L.27 p.5

Authors use the same nomenclature "NC ratio" for ratio of the nuclear to cell volume as well as nuclear to cytoplasmic volume. One definition should be used consistently. Also, the term "NC volume" might be reconsidered.

L.12 p.4

"We also compared nuclear volume distribution of RPE1 cells measured with both N2FXm and 3D confocal reconstruction (see materials and methods) obtaining similar results"

This wording implies that the same cells were imaged with the two methods; is this true? The bar graph appears to show a sizable difference, but the authors say that it is not significant. Can they provide the mean values and the standard error? Can the authors include an example of the confocal z stacks with their markers. (This might address whether there is any detectable NES-GFP in the nucleus).

L.23 p.4

“DCIS.com ”

Are the authors referring to the cell line “MCF10 DCIS.COM”? This should be clarified. Also, in the methods, the authors refer to DCIS.com medium; is this correct?

Figure1

Can Panel D and E be plotted with the same y-axis to facilitate comparison.

Figure 2A

As the nucleus breaks down during mitosis, it is confusing what is being measured with this assay when there is no intact nucleus. This should also be clarified in the text. Consider removing these values after NEB on the graph, or make them a lighter orange color to denote that it is not really the nuclear volume.

L.3 p5 & Figure 2C

“We found that NEB systematically precedes the onset of cellular roundup by ~10-20 min.”

This statement is not backed well by the corresponding graph. The time points in the graph of 20 min apart do not allow for an accurate assessment of timing, and three of the five strains have similar percentages at -20 and 0 time points. These data should be discussed in relationship to previously published data on this time.

L10. P.5

“This implies a non-constancy of the NC volume ratio between NEB and PME (Fig. 2F)”

Can the authors present a statistical test between the paired Nuc and Cyto ratio results presented in Figure E and the NC ratio at PME and NEB in panel F.

L12 p5

The results on growth rates need to be better described. Currently it is presented as the last graph in Figure 2, along the division data, and discussed in the same paragraph as the cell division, but describes

behavior on an entirely different magnitude time scale. This is confusing to the reader. At the very least, the data on growth should be discussed in separate paragraphs for division in the text, and ideally presented in separate figures. As mentioned above, more context to this result should be given.

L.17 p5

“Overall, these results indicate that in mammalian cells the nucleus-cytoplasm volumetric coupling could not be simply defined by a pure osmotic equilibrium, which would lead, instead, to a constant 20 value of the NC ratio (16, 17).”

There are certainly circumstances in which NC ratios can vary even if they only use osmotic mechanism. Reference 16 shows that if the nuclear growth rate speed is proportional to the cell volume then a pure osmotic theory can explain N/C ratio maintenance. Can the authors show the Vol growth speed for nuclei as a function of Norm volume for cells? Is it flat or linear? If it is not linear than the pure osmotic model cannot explain these results.

L.28 p5

“As expected from a purely osmotic model”

Can the authors clarify what they define as a "purely osmotic model". Can they refer to an equation? They cite this pure osmotic model with references 16, 17, but these papers use models that also take in account membrane tension.

L.30 p.5

“Our model also shows that the same behavior is expected for a non-negligible constant surface tension, with a small correction on the slope, but the expected nuclear volume changes due to external forces are [...]”

Can they refer to an equation?

Figure 3 H & I : The plots look very similar, even in the SD. Please confirm that the NC data shown are correct.

L.11 p.6

“was mostly unaffected along hyperosmotic shocks (Fig. 3E).”

Can the authors conduct a test for the Norm Mean Inv FRETindex at t=0 min and after the hypertonic shock t>10 minutes. It looks like there is a significant difference

Figure 3

Panel E,J,O. Can these be plotted with the same y axis to facilitate comparison?

Panel C,D,E. Can the authors label when the hyper osmotic shock occurs on the graph?

Figure 3 H & I. The plots look very similar, even in the SD. Please confirm that the NC data shown are correct.

L.19 p.6

“nuclear and cytoplasmic volumes are strongly decoupled, with nuclear volume variations coherent with the changes of forces exerted on the NE and not with the changes of cytoplasmic volume (Fig. 4A).”

It would be important to test if NES-GFP localization is altered during these experiments.

L.21 p.7

"specific transcription factor such as YAP, key regulator of organ growth and regeneration as well as of mechanotransduction, also are affected by this mechanism."

Please cite a reference for this statement.

References

Update Lemiere reference #16

Fix Deveri reference #17

Fix Zimmerli reference #26

Reviewer #1 (Remarks to the Author):

This paper experimentally explores the effects of various biochemical and mechanobiological perturbations of single cells on the ratio of their nuclear to cytoplasmic volumes. It is shown that changes in cytoskeletal forces exerted on the nucleus (via latrunculin or ROCK inhibition) changes the NC ratio in a manner that is not simply due to the direct effect of those forces on the osmotic pressure balance. In addition, it is shown that while the NC ratio in many cell types (under hyperosmotic conditions) is constant during most of interphase, this ratio varies considerably during cell division. Interestingly, it is found that right after cell division, the cytoplasmic volume increases exponentially, while the nuclear volume increases only linearly with time. The results will stimulate further experiments and theoretical models and should be published in Nature Communications, pending the following changes.

We are grateful for the positive remarks, and for the positive recommendation.

We believe that the authors overstate the impact of their results with respect to the concepts presented in Refs. 7, 16, and 17. In particular, the present manuscript uses the term “falsify” (once) “or “falsified” (twice) to relate their findings to the presumably naïve “osmotic” models in these references. This, besides being polemical, is not correct and does not reflect the subtlety of the ideas previously presented. While the NC volumetric ratio is indeed determined by osmotic pressure differences, the proteins involved are those whose active transport through the nuclear pores, results in differing concentrations in the cytoplasm and nucleus. The import rate and export rates can be different for systems with active transport, resulting in concentrations of proteins that are directly related to these rates. When various perturbations are applied to the cell, including changes in the cytoskeletal forces that act on the nucleus, those transport rates are modified, with – in general – different changes for the import and export rates. Thus while it is true that the naïve osmotic effect of changes in cytoskeletal force are not significant in the experiments described, there is an important indirect effect of those forces on the transport rates that determine the protein concentrations in the nucleus and cytoplasm, and hence the osmotic pressure difference and NC ratio. This was indeed a major point of these references which seems to have been missed in the present manuscript.

Nevertheless, the paper is interesting and presents several novel results about how the NC ratio changes during the cell cycle and under biochemical or mechanobiological changes, which justify publication in Nature Communications. However, before we can formally recommend publication, the interpretation of these changes with respect to such perturbations should be rewritten to accurately incorporate the previously discussed changes in osmotic pressures due to active transport and to qualitatively relate these transport changes to the perturbations applied in the present manuscript. In addition, we request the authors to address the following major and minor comments.

We apologize to the reviewer for our miscommunication. Our intent was not to go against previous models (which of course had different scopes and were not informed by our data), and we meant our statement referred exclusively to the ingredients of our own model, which is a naive version of the osmotic model, taking into account just few essential ingredients

and making some simplifications in order to keep a minimal number of parameters. We have rephrased the statements in order to avoid any misunderstanding, and also added a statement on the results of the cited refs. Specifically we added this sentence in the discussion section: “When perturbations are applied to the cell, including changes in the cytoskeletal forces that act on the nucleus, transport rates can be modified, with – in general – different changes for the import and export rates. Thus while it is true that the naïve osmotic effect of changes in cytoskeletal force are not significant in our experiments, there is an important indirect effect of those forces on the transport rates that determine the protein concentrations in the nucleus and cytoplasm, and hence the osmotic pressure difference and NC ratio. This was a major point of previous studies [REFS 7-16-17].”

Major comments:

1. At the bottom of page 5, the authors write “Our model also shows that the same behavior is expected for a non-negligible constant surface tension...”. Compared to what scale is the surface tension non negligible? The work by Finan et al. (Annals of Biomedical Engineering, 2009) presents the notion of NE that is stretched compared to one that is relaxed. The authors should relate the findings of their model with respect to the work of Finan et al.

We thank the reviewer for this comment and we refined the inaccurate statement in the current version of the manuscript. We have explored constitutive surface tensions in the range $1e^{-6}$ to $1e^{-2}$ N/m. The value around $2e^{-2}$ N/m found by Finan and coworkers by a fit of their model can be compared with the upper bound of the constitutive surface tension that we explored. Hence, our results should apply (see e.g. fig SM3 in the SI appendix). It is important to state that their richer model (based on the swelling of porous gels) contains other parameters, hence although the basic underlying physical picture of our simpler model is similar, it is possible that the values of the parameters do not map exactly between the two frameworks. Finally, we note that while we explore for simplicity our tension-biased transport model in a regime of negligible tension, this choice has no impact on our main statement, as the choice of a finite/large constitutive tension would make it equally difficult for osmotic and mechanical extensile forces to swell the nucleus.

2. The authors conducted three types of experiments to modulate the forces that are exerted on the cell and are transduced into the nucleus: hyperosmotic shock, cell detachment, and cell spreading. These types of experiments were thoroughly investigated in previous works such as Finan et al. (mentioned above), Guo et al. (ref. 19 in the manuscript), and others. However, discussion that relates the results of the manuscript to these previous works is lacking. The authors should present their experimental results in a wider context and relate them to these previous works.

The work of Finan and coworkers has the merit of showing that the volume increase of the nucleus upon hypo-osmotic shocks, saturates and does not follow a linear increase as the cell volume does (Ponder plot). The simplest interpretation of that is that part of the pressure is compensated by nuclear envelope tension, leading to less volume increase than expected when tension is negligible.

We could say that nuclear envelope stress can have two effects, which can act on different timescales: one on nuclear import, which we focus on in our study, which makes nuclear volume increase when the envelope is more stretched compared to a reference state, and one that has a direct mechanical effect, and balances the difference of osmotic pressure between the nucleus and cytoplasm, by which increasing tension leads to a decrease of nuclear volume for a given osmotic balance at steady state. In our study, due to cell spreading, we can hypothesize that we are in a regime of low effective constitutive tension, and the import effect dominates.

3. At the bottom of page 37, the authors relate the tension to external force by the equation $F_{ext} = 8\pi RN \sigma_{ext}$. The authors should explain this relation as its derivation is unclear. Perhaps a cartoon showing the force, tension, and the relation between them can make the derivation clearer.

We apologize for the missing elements in our argument. We have clarified the assumptions, and we have specified in the revised text that we intend this relation as a simple estimate relating a scenario of tension of external origin to the total magnitude of the radial forces that are needed to generate the equivalent mechanical pressure difference.

4. The authors model the effect of force on nuclear import using an exponential relation. The author should discuss the underlying biophysical reasoning of this dependence or state it is just a fitting function.

We agree that this assumption needs an explanation - which was pointed out also by reviewer 2. We have made it clear in the text that we have assumed an activated process. This assumption does not really affect the quantitative aspects, since other functional forms would be equivalent in the relevant range of values.

5. In their theoretical model, the authors treat the external force as a scalar although by definition force is not a scalar. Furthermore, the authors assume that the nucleus is spherically symmetric, which is not true in general, when deformed by non-isotropic external forces (that may have shear components).

We have clarified these aspects. We chose the spherically symmetric case to simplify the model from parameters that could not be fixed reliably based on our data. Given the assumption, the force field only has one relevant degree of freedom, hence the sloppy notation (now clarified). Additionally, we have clarified that we mean the model in this study more as a conceptual guide to understand the experiments than as a full-blown theoretical description. The purpose is to isolate and select scenarios, hence we systematically reverted to the simplest assumptions.

6. The authors state that they assume the forces that are exerted on the nucleus are compressive. In that case and when the forces are spherically symmetric, the contribution of the external forces to the tension should be negative. Is this the case? This should be clearly explicated since this affects the way the exponential relation between the surface tension and nucleocytoplasmic ratio is interpreted.

We have clarified this part in the revised SI text. In our notation the contribution of compressive external forces to the tension is positive (as they both contribute to forces pointing inwards). We analyzed both cases of positive and negative contribution of external forces to the tension, but given the previous experimental literature [e.g. <https://doi.org/10.1073%2Fpnas.0908686106>] we believe that the first scenario could be more likely.

Minor comments:

1. Can there be forces that are exerted on the nucleus which will not be detected by stretching of the LINC complex (e.g. squashing of the nucleus)?

This observation is correct. The FRET tension sensor we developed is designed on Nesprin1, one of the components of the LINC complex. It spans from the inner nuclear envelope, where it binds SUN domain containing proteins, to the cytoplasm, where it binds to actin. Therefore, it is sensitive only to forces exerted by the cytoskeleton to the NE. If external forces applied to the cell indirectly also affect the link between actin and SUN proteins at the NE, then the sensor can detect it. More detail about the FRET sensor can be found in Poli et al 2023 [<https://doi.org/10.1038/s41467-023-37064-0>]. Please consider also our response to point 3 below.

2. Some works (e.g. Jahed et al., Journal of Cell Science 2016) suggest that the cytoskeleton and LINC complex are indirectly involved in nucleocytoplasmic transport by modulating the Ran protein gradients. Is it possible that the nucleocytoplasmic transport in the latrunculin-treated or the ROCK inhibited cells is affected via this pathway rather than mechanical “expansion” of the NPCs? Can the author comment on this possibility.

We do not exclude that there may be additional/other mechanisms at play, as suggested by the reviewer. However, we have observed a strong correlation between nuclear envelope tension, as measured by our FRET sensor, and the mean intensity ratio of GFP-NLS between the nucleus and cytoplasm (used as a marker of protein nuclear incorporation efficiency) (Fig. 4C). This dependency is essential for our model, independently of whether this is due to nuclear pores expansion or to other possible mechanisms.

3. At the bottom of page 6 the authors write that the cytoskeletal forces are compressive. However, these can also be expansive if the nucleus is laterally stretched. Can the authors provide a reference to a work that distinguish between these two possible modes of nuclear deformation?

The reviewer’s observation is correct. It could be, in certain circumstances, that positive forces are exerted to the nucleus. With our model, indeed, we analyzed the effect on nuclear volume of positive as well as negative external forces (always with the assumption of a spherically symmetric nucleus). In both cases variations in nuclear volume induced by direct forces are negligible with respect to the ones arising from changes in osmolarity (less than 10% for σ_0 in the range 10^{-6} to 10^{-2} N/m).

We now modify that sentence in: “Based on the model, we reasoned that this was very unlikely to be a direct effect, as, independently of the direction, compressive or stretching, the force required to change nuclear size would be abnormally large (see SI appendix) [24, Kalukula et al, Nature Reviews Molecular Cell Biology, (2022), 583-602, 23(9), DOI: <https://doi.org/10.1038/s41580-022-00480-z>].”

4. The authors mention that the estimate of the NE tension (10^{-6} N/m) is a lower bound due to fluctuations of active origin of the NE shape. Are there evidence for such active fluctuations? While this does not change the conclusion of the authors, it is worthwhile to give a reference for such active processes that facilitate NE shape fluctuations.

We agree and we have added this statement: “The apparent tension of the nucleus, estimating σ_N or σ_0 is found to be around 10^{-6} N/m in nuclear shape fluctuations experiments (Chu et al., 2017; Introini et al., 2021).”

Reviewer #2 (Remarks to the Author):

Pennacchio et al extended the powerful fluorescence exclusion microscopy, FXm, to simultaneously measure cell volume and nuclear volume – a new method they coined “N2FXm”. This novel method is sound and allowed the authors to study the nuclear-to-cell volumetric (NC) ratio. First, they found that the NC ratio was not constant during the cell cycle, being particularly impacted by mitosis. Second, they showed that the cell and nucleus did not follow the same growth laws. Third, they discovered that the NC ratio could not be explained solely by an osmotic balance and that cytoskeletal forces exerted onto the nuclear envelope could bias nuclear import and nuclear osmotic pressure, thus also participating in setting the NC ratio.

The manuscript is clearly written, and the data are of high quality and interest, shading light on novel regulatory mechanisms of nuclear volume. I would greatly recommend publication should the following questions be addressed.

We thank the reviewer for the overall positive judgment of our work and for the enthusiastic recommendation.

1/ The N2FXm method

I found the N2FXm method extremely powerful and relevant. Although most of my concerns are addressed in the methods or in the paper, I have one additional question and one recommendation for the authors:

i) Have the authors checked that there is no GFP-NES in the nucleus, and if the signal is homogeneous in the cytosol? Depending on the cell state/strength of the NES, there may still be some GFP in the nucleus, which would affect the measurement – especially if this bias is cell cycle-dependent, through for instance NE tension impacting nuclear export. Even if non-significant, there seems to be a small difference in the nuclear volume measurement between the 3D reconstructed and the N2FXm in Fig. S1F. Moreover, this could also be important in the perturbation experiments, impacting the actual measurement.

We thank the referee for these relevant remarks, which were also raised by reviewer 3. GFP-NES background in the nucleus, indeed, could affect the precision of N2FXm. Similarly, dishomogeneity in the distribution of GFP-NES in the cytoplasm, could perturb the second calibration, then impacting on nuclear volume estimates. We now explicitly discuss these possible drawbacks in the revised version of our manuscript. Additionally, following the reviewer's advice, we evaluated the distribution of the GFP-NES signal in correspondence of the nucleus (Rebuttal-Fig1.A). The mean radial intensity profile is minimum at the center and increases toward the periphery. In low magnification widefield images, the signal background simply originated by the presence of GFP-NES in the nucleus is expected to be rather constant or proportional to nucleus depth, then maximum at the center and minimum at the periphery. The profile we measured (middle panel in Rebuttal-Fig1.A) is instead compatible with a signal coming from the cytoplasm and then integrated over the cytoplasmic volume surrounding the nucleus. In the third plot in Rebuttal-Fig1.A we show the expected theoretical radial profile for a spherical nucleus (of unitary radius) embedded in a labeled cytoplasm:

$$f(r) = \int_0^r (1 - \sqrt{1-x^2}) dx = r - \frac{1}{2} * \arcsin(r) - \frac{1}{2} * r * \sqrt{1-r^2}.$$

This is compatible with the experimental profile we measured.

Importantly, we cannot exclude that a small portion of the signal we record in correspondence to the nucleus comes from inside the nucleus. Indeed we recorded a background line on the radial profile that could have originated both by GFP-NES on top and bottom of the nucleus or by GFP-NES in the nucleus. However, when we evaluate in a confocal section (Rebuttal-Fig1.C) the proportion between cytoplasmic and nuclear GFP-NES signal, it becomes clear that the majority of the signal comes from the cytoplasm. In the mid-nucleus confocal sections, the mean intensity of GFP-NES in the cytoplasm is approximately 5 times higher than in the nucleus (Rebuttal-Fig1.C). Importantly, this ratio is not perturbed by drug treatments (Rebuttal-Fig1.C).

To allow the reviewers to judge directly if and how inhomogeneity in the distribution of GFP-NES signal in the cytoplasm could impact our measurements we plotted a representative example of a cumulative distribution of R^2 values, relative to more than 150 frames over 10 cells, for linear fits between GFP-NES cytoplasmic intensities and calibrated optical heights (Rebuttal-Fig1.B and ref to Figure 1.C). R^2 distribution is strongly biased towards 1, median = 0.93, sd = 0.14. Significant inhomogeneity in the distribution of GFP-NES signal, such as local accumulations or depauperation, would have perturbed the linear dependency between these two quantities and then the distribution of R^2 values.

All together these considerations suggest that GFP-NES background in the nucleus and as well as possible dishomogeneity in the distribution of GFP-NES in the cytoplasm and its impact on second calibration, are neglectable/minimal. This is probably due to the 'low' sensitivity and resolution of the imaging system settings as they are necessary for the N2FXm method to properly work, basically low numerical aperture and low magnification objective. Actually, it is important to notice that we challenged our method with two independent controls. First, we compared our nuclear volume measurements with 3D confocal reconstruction of nuclei, and we didn't record a significant shift in the distribution of nuclear volume estimates of the same cell population. Second, we measure in living cells the volume of artificial spherical objects (DAAMs particles). N2FXm measurements were in good agreement with theoretical volume computed by geometrical reconstruction. These two complementary validations well support the goodness of our method.

ii) I believe one important point is that the GFP calibration is done at every time point and for each cell, thus limiting temporal fluctuations or cell-cell heterogeneity in GFP concentration. As such, it may be better to directly state it in the main text, and not only in the methods.

We thank the reviewer for this comment and as suggested we now specified it in the main text.

2/ Nuclear growth and NC ratio

I have a few questions regarding nuclear volume growth and subsequent NC ratio values, as well as their interpretations based on the model proposed by the authors:

i) The NC ratio seems rather constant in each cell line from -500 min to 0 min. First, can the authors provide a y-axis zoom before and after mitosis, to have a better visualization (Figs. S1L & 1F)? Second, it seems constant even though nuclear volume does not increase exponentially. Can the authors comment on this point?

To enhance the difference in NC ratios we showed it at its maximum, between PME (post mitotic expansion, nucleus life start) and NEB (nuclear envelope breakdown, nucleus life end), please see plot in Fig2.F. NC ratios are not constant and typically increase by at least 20% along the nucleus lifetime. We added this quantification in the main text of the revised version of our manuscript. As suggested, we also added in Fig2.F a graphical representation of relative paired t-tests results.

ii) Can the authors speculate (and potentially expand the discussion on this point) as to why the nucleus would grow linearly in time? Per the authors' model, if it does not increase exponentially, this could for instance mean either that import/export is cell cycle-dependent such that nuclear osmotic pressure is not constant over the cell cycle. This could for instance be the case if NE tension changes during the cell cycle (see the question on tension in point 3/ below).

The average linear growth on cell-cycle time scale is an intriguing fact that we report experimentally, but whose explanation goes beyond our scopes. Most of our data focus on effects on time scales of minutes to less than an hour, and our explanations are fully coherent on these time scales. Cell-cycle time scales involve different processes and likely different explanations are necessary.

This said, we can reason on how the data relate to the prediction of different simple models.

Let us suppose first a scenario of perfect osmotic force balance. If the tension were always negligible, the volume of the nucleus would grow exponentially as the total volume. Our analyses suggest that the tension would have to be too large to have a relevant impact on nuclear size and explain the average linear growth.

If we consider our model coupling tension with transport, we can ask whether extrapolating it to cell-cycle time scales would support linear growth. In this model the observed tension with cell cycle would increase nuclear import, causing the nucleus to swell more than predicted by osmotic balance. Initially, it would grow faster than the cytoplasm, as it does in our data (please see also the reply to point 4 of Reviewer#3). However, without more specific information our model cannot be easily used to suggest that this growth would be linear, which will require a specific and gradual release of external tension.

Recently, Rollin and coworkers have proposed a mechanism that could partly explain this trend (<https://www.biorxiv.org/content/10.1101/2022.08.01.502021v1.full>). In this preprint the authors claim a role of counterion release by chromatin folding. According to their model, the NC ratio (formula 15) can be intermediate between an expression that is the ratio between nuclear proteins and cytoplasmic proteins (NC1, the prediction of pure osmotic force balance) and one (NC2, larger) that depends on DNA charge, which is constant during

G1 phase, while the number of proteins in the nucleus grows with time, so NC2 decreases with time. Hence, this prediction, valid only in G1, could generate the right qualitative trend.

More widely, nuclear envelope tension might have at least two effects (acting on different timescales): one on nuclear import, which we focus on here, which would make the volume increase when the envelope feels more tension, while one direct mechanical effect would lead to a decrease of nuclear volume for a given osmotic balance at steady state. For high enough nuclear envelope tension, the volume loss effect might dominate, while for low tension (as in our study, due to cell spreading), the import effect might dominate.

Altogether, these synergies of complex effects make it hard to predict the volume growth of the nucleus during the cell cycle: a complete model would be needed, plus many more experiments.

iii) Related to the previous point, it does seem that, after the spreading phase (phase III), there is no or little growth of the nuclear volume, followed by more rapid growth. This seems true for the cell lines in Fig. S1L. Has this been considered in the calculation of volumetric growth? Could it be that the binning hides different growth regimes of the nucleus during the cell cycle?

We agree with the reviewer that at faster time scales, there may be several intriguing cell-cycle dependent phenomena that we missed in this study. We hope to perform higher-resolution time-resolved studies in the near future to address this and other questions. In particular Venkova and coworkers (Venkova et al, eLife 2022: <https://doi.org/10.7554/eLife.72381>) have shown nontrivial osmotic behavior of the cytoplasm during the spreading phase, so the referee is correct that it is reasonable to expect interesting effects for nuclear volume during vs after post-mitotic cell spreading.

iv) Finally, can the authors speculate as to why the nucleus would decrease by roughly 2.5 times its volume at mitosis? This suggests that it grows “more than necessary”, and that this could be linked with cell cycle-dependent growth regimes.

In a pure dynamic osmotic coupling it is expected to follow the cell volume pace up to double. We speculate that tension coupling dynamic contributes to the extra accumulated nuclear volume. But other processes could also contribute to this unbalance.

3/ Force-biased import and nuclear envelope tension

I find the conclusions of the authors that the force-biased import plays a role in regulating nuclear volume very appealing. I have a few questions regarding the experiments and the model:

i) Since the measurement of NE tension is not a standard method (and even though Ref. 22 is in press, it is still unpublished), maybe the authors should give more details about this method in the main text.

We apologize for the lack of details relative to the NE tension FRET sensor we developed. The paper from Poli and al. is now accepted (<https://doi.org/10.1038/s41467-023-37064-0>).

In the reply to reviewers section of that paper there are also additional controls. We hope now all the information relative to that valuable tool is publicly available.

ii) Related to point 2/ above, have the authors checked how is NE tension regulated during the cell cycle? Could tension explain why nuclear volume increases linearly?

Nuclear envelope tension has already been reported to increase during the cell cycle [Introini et al., doi: <https://doi.org/10.1101/2021.11.25.469847>; Chu et al., doi: <https://doi.org/10.1073/pnas.170222611>; Lomakin et al., doi: [10.1126/science.aba2894](https://doi.org/10.1126/science.aba2894)]. Interestingly, also assuming that NE tension increases linearly with time, that is not sufficient to explain how nuclear volume grows. The relationship between NE tension and nuclear volume, as previously stated, is complex and it could be both positive and negative. Positive, since an increase in NE tension would facilitate diffusion of molecules into the nucleus, affecting the osmotic equilibrium between the nucleus and the cytoplasm. Negative, since NE membrane tension counterbalances nucleus inflation.

iii) Related to the hyperosmotic experiments: although at first I was satisfied with the result, I am now a bit lost. It is known that a hyperosmotic shock rapidly induces changes in the cytoskeleton (see for instance Thirone, 2009, PMC5047760). As such, there should be an effect on NE tension, and by the authors' model, an effect on the nucleus volume, but it does not seem to be the case. Can the authors comment on this point? Maybe the effect is only on the cortex and not translated to the NE envelop?

This is an excellent point and indeed, as the reviewer, we were puzzled by that result. Interestingly, looking carefully at the movies, we noticed that upon hyperosmotic shock cells shrink immediately but mainly by lowering their height and not by changing their spreading area. We can speculate that this kind of compression response is much faster and reversible than restructuring of adhesion and contraction of the cell base. This seems to imply that disassembly of focal adhesions might happen later as a slower response/adaptation.

iv) Per the authors' model, it is my understanding that growing cells on substrata of different rigidities should impact the NC ratio. I did not find anything on this point in the literature. Do the authors know something about this?

We agree with the reviewer, our model indeed predicts that substrate rigidity, impacting on cell contractility and NE tension, would affect NC ratio. While the situation is of course way more complex than our model can describe, this is a very interesting point that we would like to address more in a future study. Interestingly, recent work from Pundel, Blowes and Connelly (<https://doi.org/10.1002/adv.202105545>), shows that extracellular physical cues affect nuclear and nucleolar volumes. Their results may be related to our model.

v) Related to the mathematical model: I appreciated the model and its explanation/parametrization strategy. I had one question on the choice of the exponential dependence of the nuclear concentration of macromolecules with external stress: why this choice? Why not linear in the first place? Especially because it seems that it is linearized in the equation defining V_n at the bottom of p. 5.

We thank the reviewer for the positive remarks on the model. We chose the exponential dependence under the hypothesis that biased transport due to the stress felt by the nuclear pores could be an activated process. However, as the reviewer correctly points out, this choice is equivalent to a linear-response assumption, and of course the data do not allow us to formulate any claim regarding this assumption. In the revised text we provided an explanation for our assumption and explicitly stated that to our scopes the assumption of linear response would be equivalent.

Minor comments

i) Figures

- adding the legend (control vs. ivermectin) on Fig. S3D would help the reader
- which condition is ivermectin in Fig. 4C? in general, what are the conditions in this plot?
- maybe ivermectin could be its own subfigure on Fig. 4 to help the reader and avoid going back to Fig. 3
- Y-axis in Fig. S2B: unit should be $\mu\text{m}^3/\text{h}$ and not μm^3

We apologize for the lack of clarity and mistakes. We improved the plot and amended the relative legend.

ii) Although the manuscript is nicely written, I found several typos, in particular in the appendix. I may have missed some, but here is a list of those I found:

- in methods section “cell transfection”, problem unit polybrene
- in methods section “N2FXm”, GFP-nes instead of NES
- in methods section “DAAM particle measurement”, typo on “technique”
- in appendix, “left” and “right” hand side are inverted I believe
- in appendix, “nucleus surface tension and e contribution” → a
- in appendix, page 2, $c^{\text{ions_out}}$ c_{out} instead of just $c^{\text{ions_out}}$ (I believe)
- in appendix, section “estimates”, in order TO confirm, prOessure
- in appendix, section “force biased”, there is a verb lacking in first sentence
- p. 6 repetition “the the”

We apologize for these mistakes and we thank the reviewer for spotting them.

Reviewer #3 (Remarks to the Author):

This manuscript describes experiments and modeling of the regulation of nuclear size by osmotic forces and mechanical properties of the nuclear envelope in cultured mammalian cells. The study of the NC ratio in mammalian cells has been hampered by a lack of accurate volume measurements. This manuscript introduces an elegant new method for estimating nuclear/ cytoplasmic volume using a fluorescence exclusion-based approach, as well as a new FRET sensor for measuring nuclear envelope tension in cells. The work investigates NC ratio changes through the cell cycle and upon perturbations such as osmotic shock and cellular detachment from the substrate. The data support a mechano-osmotic model that implicates osmotic pressure and nuclear transport as critical components for nuclear size control.

In general, this work has potential to be an important advance in the field, but it is currently too preliminary for publication. There are many significant concerns described below. These include insufficient characterization of the new methods and a general tendency for over interpretation leading to inappropriate conclusions. They present some interesting initial observations without establishing sufficient context or additional insights. Some limited experimental data and careful rewriting will be needed to improve this manuscript.

We thank the reviewer for seeing potential in our work.

Major comments

1. The title, summary and parts of the abstract are not appropriate as key statements are not sufficiently backed by the data in this paper. For example, the Summary statement is: "Cytoskeletal forces exerted on the nuclear envelope impact on nuclear volume through modulation of force-coupled nucleo-cytoplasmic transport. However, there is little in this paper on cytoskeletal forces or nuclear transport. The closest test is an effect with ivermectin. However this result does not directly show that nuclear transport rates have changed. There is also no clear data on cytoskeletal forces on nuclear import; small effects with LatA are shown, but these may be highly pleiotropic and cannot be just assumed to be actin effects on the nuclear envelope. Their statement in the last paragraph "forces exerted by the cytoskeleton, affecting NE tension, impact on nuclear pores size (26–28)" have not been conclusively demonstrated in these cited papers and have not been proven here. While the authors certainly may hypothesize upon force-effects on nuclear transport and pores, these statements should not be the central conclusion of this paper. I think that the authors should address these concerns by judicious rewriting the wording of their conclusions, rather than being asked to perform the large number of experiments needed to support claims.

We agree with the reviewer that it is not the scope of our work to prove that forces exerted by the cytoskeleton affect nuclear transport by impacting on nuclear pores size. The phenomenon of tension-biased transport that we leverage to explain our data appears to be well supported by the literature. Indeed, the idea of tension-biased transport was introduced, supported and quantified in a seminal paper from the Roca-Cusachs lab (Elosegui-Artola et al., Cell 2017, ref. 27). The Roca-Cusachs lab recently published an elegant and systematic report on how cytoskeletal forces affect nuclear cytoplasmic transport (Andreu et al., NCB 2022, ref. 25). The idea that this process could be coupled to pore size was supported by two later independent structural studies, Zimmerli et al. (Science 2021, ref.26 ) and Schuller

et al. (Nature 2021, ref. 28), both showing that nuclear pores size depends on the cell's tensional state. However, this is not essential for us. In our study, Fig4.C, shows a clear correlation between NE tension and GFP.nls NC ratio, here used as a marker of nuclear cytoplasmic transport efficiency. As the reviewer highlights, based on our data it is reasonable to conclude that cytoplasmic forces, affecting NE tension, impact on nuclear cytoplasmic transport and nuclear volume. We completely agree that our data does not add anything on the question of nuclear pores size changes. We rephrased our conclusions accordingly in the revised manuscript.

2. There are important concerns about the N2FXm method that need to be addressed. Additional data and more detailed descriptions of the calibration method are needed so that the accuracy of the method can be evaluated (see below). The authors statement that the GFP "localizes in the entire cytoplasm except the nucleus" is not true. For example, other organelles besides the nucleus may also exclude the cytoplasmic GFP; lysozymes, mitochondria, lipid droplets etc., might represent 2-10% of cellular volume. This issue and how it may affect NC ratio measurements using N2JXm certainly needs to be addressed. Can the authors determine if these organelle volumes significantly affect the NC ratio measurements? Another concern is whether NES-GFP is truly all excluded from the nucleoplasm. Can the authors include a control using high-resolution confocal sections to show that there is no significant nuclear signal? If there is some nuclear signal, could they determine how that would affect the accuracy measurements? They should also show whether the localization of the NES-GFP marker is affected by their perturbations. For instance, mechanical or drug perturbations could cause entry of the NES-GFP in the nucleus, which could potentially affect their measurements and conclusions.

We thank the referee for these relevant remarks, which were also raised by reviewer 2. GFP-NES background in the nucleus, indeed, could affect the precision of N2FXm. Similarly, dishomogeneity in the distribution of GFP-NES in the cytoplasm, could perturb the second calibration, then impacting on nuclear volume estimates. We now explicitly discuss these possible drawbacks in the revised version of our manuscript. Additionally, following the reviewer's advice, we evaluated the distribution of the GFP-NES signal in correspondence of the nucleus (Rebuttal-Fig1.A). The mean radial intensity profile is minimum at the center and increases toward the periphery. In low magnification widefield images, the signal background simply originated by the presence of GFP-NES in the nucleus is expected to be rather constant or proportional to nucleus depth, then maximum at the center and minimum at the periphery. The profile we measured (middle panel in Rebuttal-Fig1.A) is instead compatible with a signal coming from the cytoplasm and then integrated over the cytoplasmic volume surrounding the nucleus. In the third plot in Rebuttal-Fig1.A we show the expected theoretical radial profile for a spherical nucleus (of unitary radius) embedded in a labeled cytoplasm:

$$f(r) = \int_0^r (1 - \sqrt{1-x^2}) dx = r - \frac{1}{2} * \arcsin(r) - \frac{1}{2} * r * \sqrt{1-r^2}.$$

This is compatible with the experimental profile we measured.

Importantly, we cannot exclude that a small portion of the signal we record in correspondence to the nucleus comes from inside the nucleus. Indeed we recorded a

background line on the radial profile that could have originated both by GFP-NES on top and bottom of the nucleus or by GFP-NES in the nucleus. However, when we evaluate in a confocal section (Rebuttal-Fig1.C) the proportion between cytoplasmic and nuclear GFP-NES signal, it becomes clear that the majority of the signal comes from the cytoplasm. In the mid-nucleus confocal sections, the mean intensity of GFP-NES in the cytoplasm is approximately 5 times higher than in the nucleus (Rebuttal-Fig1.C). Importantly, this ratio is not perturbed by drug treatments (Rebuttal-Fig1.C).

To allow the reviewers to judge directly if and how inhomogeneity in the distribution of GFP-NES signal in the cytoplasm could impact our measurements we plotted a representative example of a cumulative distribution of R^2 values, relative to more than 150 frames over 10 cells, for linear fits between GFP-NES cytoplasmic intensities and calibrated optical heights (Rebuttal-Fig1.B and ref to Figure 1.C). R^2 distribution is strongly biased towards 1, median = 0.93, sd = 0.14. Significant inhomogeneity in the distribution of GFP-NES signal, such as local accumulations or depauperation, would have perturbed the linear dependency between these two quantities and then the distribution of R^2 values.

All together these considerations suggest that GFP-NES background in the nucleus and as well as possible dishomogeneity in the distribution of GFP-NES in the cytoplasm and its impact on second calibration, are neglectable/minimal. This is probably due to the 'low' sensitivity and resolution of the imaging system settings as they are necessary for the N2FXm method to properly work, basically low numerical aperture and low magnification objective. Actually, it is important to notice that we challenged our method with two independent controls. First, we compared our nuclear volume measurements with 3D confocal reconstruction of nuclei, and we didn't record a significant shift in the distribution of nuclear volume estimates of the same cell population. Second, we measure in living cells the volume of artificial spherical objects (DAAMs particles). N2FXm measurements were in good agreement with theoretical volume computed by geometrical reconstruction. These two complementary validations well support the goodness of our method.

3. There are also concerns about the FRET-based nuclear envelope force sensor. This is a new sensor developed by this group, and its development is described only in this work and in a preprint. However, neither manuscript presents sufficient background characterization of this probe to demonstrate that it is acting as a force sensor at the nuclear envelope. Although the handful of data points are consistent, more quantitative and systematic controls would give more confidence that this sensor indeed measures force at the nuclear envelope.

We apologize for the lack of details relative to the NE tension FRET sensor we developed. The paper from Poli and al. is now accepted (DOI: 10.1038/s41467-023-37064-0). In the reply to reviewers section of that paper there are also additional controls. We hope now all the information relative to that valuable tool is publicly available.

4. One of the most surprising results is the linear growth of the nucleus vs. exponential growth of the cell volume. Other than pointing out the difference, this is not further investigated. More data are needed to confirm and develop this observation. One simple prediction is that NC ratios systematically fall as cells grow larger during interphase (however, this is not apparent in Figure 1F). Could this difference arise from some

systematic error in the N2FXm measurements? The reader is left without any context on how to think about this linear growth rate, or whether it carries much significance.

We agree with the reviewer: the observation that cell and nucleus volumes don't grow at the same pace is probably the most relevant result of our study. Most of our data focus on effects on time scales of minutes to less than an hour, and our explanations are fully coherent on these time scales. Cell-cycle time scales involve different processes and likely different explanations are necessary. Please see also reply to point ii of Reviewer#2. The fact that nuclear volume growth speed is constant (and independent of nuclear size) doesn't necessarily mean it is always slower than the cytoplasmic one. Indeed, while the cytoplasm doubles in volume during the cell cycle, the nucleus grows approximately 2.5 times (Fig2.E). To better clarify this point, we simulated time dependent volume growth curves for cyto and nucleus, and relative NC ratio (Rebuttal-Fig1.D). The cyto volume is assumed to grow exponentially while the nucleus linearly. The parameters used for the simulations, volume growth speed and minimal volume, are the ones obtained from the fit of experimental averages volume growth speed for RPE1 cell (Fig2.G). As the reviewer pointed out, at long time points, the NC ratio will collapse. However, for a time interval in the scale of a cell cycle and with the parameters obtained fitting the experimental data, the NC ratio is initially growing and lately starts to decay. For time points over the average span of a cell cycle, the NC ratio will certainly tend to zero. It would be for sure interesting to test our model in cells going to senescence and investigate the NC ratio dynamics in those conditions. This is however behind the scope of this work.

Moreover, we evaluated the time averaged values of cytoplasm and nucleus normalized volume growth speeds for all cell lines analyzed (Rebuttal-Fig1.D). Normalized average nucleus volume growth speeds are comparable, and typically higher (with the exception of the MCF7 cell line), than the one of the cytoplasm. This explains why the NC ratio doesn't systematically fall in the time scale of a cell cycle.

As suggested, in the new version of our manuscript we further discuss the linear versus exponential volume growth speed.

5. Another result is the apparent loss of nuclear volume after mitosis. Has it been confirmed by other methods or in other papers? Could the results be explained by the behavior of the NES marker at these cell cycle stages? If NC ratios are low after mitosis and then fall over interphase because of linear growth, then how does the nuclear volume ever catch up? How are NC ratios then maintained over multiple cell cycles? In Figure S2A, there is an example where nuclear growth suddenly speeds up after 50 frames (what is the frame rate?). Is this transition typical (as it is not seen in the averaged data)? Does it correspond to a known cell cycle transition? In general, these data are not discussed sufficiently in context to what is currently known about these processes. If the authors hope to describe thoroughly the NC ratio changes over the cell cycle, these issues should be clarified. Overall, the measurements of nuclear volume just before and after mitoses in Figure 2 represent a weak point in the manuscript and could even be deleted (or placed in another manuscript).

It is certainly true that our method can measure the volume of the nucleus only when this compartment is properly defined. In the few frames between nuclear envelope breakdown (NEB) and nucleus sealing after mitosis, centrally preceding nucleus postmitotic expansion (PME), the nucleus is not an isolated compartment anymore and can't be defined. During

this time interval, GFP-NES diffuses inside the nucleus and our method measures a residual volume that is still not accessible to it. We expressly clarify this point in the updated method session of our manuscript. For this reason, when we want to compare nuclear volume measurements before and after mitosis we compare them at NEB and PME, time points when the nucleus is still integer and when it is surely integer again, respectively. We apologize for the lack of explanation, nuclear post mitotic expansion (PME), the fast nuclear volume increase after mitosis, has been previously described, please see as example: Gerlich et al., NCB 2001; Baarlink et al., NCB 2017 and Krippner et al., EMBO Reports 2020. We now introduce PME with a relevant reference in the revised version of our manuscript.

Regarding how the nucleus can “catch up” its volume after mitosis, please see the reply to point 4.

The single-cell plot in Fig.S2A was intended to show the quality of the original data. Analyzing at the single-cell level the relationship between specific cell cycle stage or transition with nuclear and cellular volume growth is beyond the scope, and the reach, of this work. This goal can be potentially achieved with our method but will require a considerable amount of work in future studies.

Specific Comments

p.3 and Methods

More details of the calibration methods and evaluation of the accuracy of the approach are needed. First, can the authors provide an experimental version of Fig S1C (top right) to demonstrate linearity with the Texas red dextran marker they use?

We apologize for the lack of clarity, the plot in Fig1.C is indeed experimental. As previously mentioned, to better evaluate the robustness of fits along different frames and cells now in Rebuttal-Fig1.B we also show the cumulative distribution of R^2 relative to linear fits of GFP-NES intensity in function of cytoplasm optical heights measured with standard FXm. Please notice that the calibration is performed for each cell and at each time point.

Second, more detailed descriptions of how the NES-GFP intensity is calibrated with the cell volume data is needed. Can the authors show in a figure a real-life example (not just schematics)?

Please see the reply to the previous point. A real-life example was already provided in the original version of our manuscript.

For calibration, more details are needed on what cytoplasmic portions of the cell is used for comparison? How well do the red and green intensities inversely correlate; can the authors add plot showing correlation of green and red intensities per pixel? How is the noise in the intensity measurements dealt with for?

We apologize for the lack of clarity. For the GFP-NES calibration we use the intensity of all cell pixels excluding the ones in the nuclear area, as defined by the H2B signal. A real example of that is in Fig1.C. The linear dependency is robust, as can be appreciated by the cumulative distribution of R^2 for several single time frame linear fits from different cells (Rebuttal-Fig1.B).

Inhomogeneity in the cytoplasm could have a large impact on the interpretation of these measurements. Can the authors test whether inhomogeneities are significant enough to affect the calibration and subsequent measurements.? For example if there are many lysosomes in one part of the cytoplasm, is that portion of the cell not used for calibration?

We agree with the reviewer, that is an important point. Please see the response to your second major point. It is true, essentially, accumulation as well as depauperation of GFP-NES in the cytoplasm, as could be generated by vesicles excluding or concentrating the fluorescent protein, could perturb the linear dependency between the GFP-NES and the calibrated optical heights. Distribution of R^2 in Rebuttal-Fig1.B shows that is not the case.

L.7 p3 , L.6 p4 & L.27 p.5

Authors use the same nomenclature "NC ratio" for ratio of the nuclear to cell volume as well as nuclear to cytoplasmic volume. One definition should be used consistently. Also, the term "NC volume" might be reconsidered.

We apologize for this mistake and we amended it in the updated version of our manuscript.

L.12 p.4

"We also compared nuclear volume distribution of RPE1 cells measured with both N2FXm and 3D confocal reconstruction (see materials and methods) obtaining similar results"

This wording implies that the same cells were imaged with the two methods; is this true? The bar graph appears to show a sizable difference, but the authors say that it is not significant. Can they provide the mean values and the standard error? Can the authors include an example of the confocal z stacks with their markers. (This might address whether there is any detectable NES-GFP in the nucleus).

We apologize for the lack of clarity. The referee is correct, we did not measure the same cells, we refer to the same cell population. Now we clarified that sentence as: "We independently measured nuclear volume distribution of RPE1 cells with both N2FXm and 3D confocal reconstruction (see materials and methods). The difference between the distributions is not statistically significant."

In boxplot in Fig. S1.F, the boxes extend from the first to the third quartiles. The middle line represents the median. The whiskers extend up or down to $1.5 \times \text{IQR}$, where IQR is the inter-quartile range.

L.23 p.4

"DCIS.com "

Are the authors referring to the cell line "MCF10 DCIS.COM"? This should be clarified. Also, in the methods, the authors refer to DCIS.com medium; is this correct?

We apologize for the non appropriate naming. In the main text and in the method section of our revised manuscript we now specify that in text, figure and legend we used the name "DCIS.com" as an abbreviation of "MCF10 DCIS.COM".

Figure1

Can Panel D and E be plotted with the same y-axis to facilitate comparison.

Yes, in the new version of figure 1 now the two plots have the same scale.

Figure 2A

As the nucleus breaks down during mitosis, it is confusing what is being measured with this assay when there is no intact nucleus. This should also be clarified in the text. Consider removing these values after NEB on the graph, or make them a lighter orange color to denote that it is not really the nuclear volume.

This is a very important point and as suggested we clarified it in the text. We agree with the reviewer, indeed at the envelope breakdown there is no nucleus anymore and GFP-NES diffuse in the nucleus. Interestingly, with our method we measure a residual volume that is still not accessible to it. We could only speculate that this is the space occupied by the chromatin.

L.3 p5 & Figure 2C

"We found that NEB systematically precedes the onset of cellular roundup by ~10-20 min."

This statement is not backed well by the corresponding graph. The time points in the graph of 20 min apart do not allow for an accurate assessment of timing, and three of the five strains have similar percentages at -20 and 0 time points. These data should be discussed in relationship to previously published data on this time.

We agree with the reviewer and indeed in the text we specified: "However, the temporal resolution of our experiments (10 min) was too small to precisely distinguish these two events."

L10. P.5

"This implies a non-constancy of the NC volume ratio between NEB and PME (Fig. 2F)"

Can the authors present a statistical test between the paired Nuc and Cyto ratio results presented in Figure E and the NC ratio at PME and NEB in panel F.

We thank the reviewer for this suggestion and we added test results in the updated version of Fig2.

L12 p5

The results on growth rates need to be better described. Currently it is presented as the last graph in Figure 2, along the division data, and discussed in the same paragraph as the cell division, but describes behavior on an entirely different magnitude time scale. This is confusing to the reader. At the very least, the data on growth should be discussed in separate paragraphs for division in the text, and ideally presented in separate figures. As mentioned above, more context to this result should be given.

As suggested we describe this result in a separate paragraph.

L.17 p5

"Overall, these results indicate that in mammalian cells the nucleus-cytoplasm volumetric coupling could not be simply defined by a pure osmotic equilibrium, which would lead, instead, to a constant value of the NC ratio (16, 17)."

There are certainly circumstances in which NC ratios can vary even if they only use osmotic mechanism. Reference 16 shows that if the nuclear growth rate speed is proportional to the cell volume then a pure osmotic theory can explain N/C ratio maintenance. Can the authors show the Vol growth speed for nuclei as a function of Norm volume for cells? Is it flat or linear? If it is not linear than the pure osmotic model cannot explain these results.

As suggested by the reviewer we plotted nuclear volume grow speed as a function of cell volume (Rebuttal-Fig1.E). Nuclear volume grow speed doesn't correlate with cyto volume.

L.28 p5

“As expected from a purely osmotic model”

Can the authors clarify what they define as a "purely osmotic model". Can they refer to an equation? They cite this pure osmotic model with references 16, 17, but these papers use models that also take in account membrane tension.

We apologize, we correct that sentence with:“As expected from a model considering only osmotic equilibrium and membrane tension (eq.2 in SI appendix)” .

L.30 p.5

“Our model also shows that the same behavior is expected for a non-negligible constant surface tension, with a small correction on the slope, but the expected nuclear volume changes due to external forces are [...]”

Can they refer to an equation?

We now refer to eq.6 in the SI Appendix.

Figure 3 H & I : The plots look very similar, even in the SD. Please confirm that the NC data shown are correct.

We thank the reviewer and we confirm that data plotted are correct.

L.11 p.6

“was mostly unaffected along hyperosmotic shocks (Fig. 3E).”

Can the authors conduct a test for the Norm Mean Inv FRETindex at t=0 min and after the hypertonic shock t>10 minutes. It looks like there is a significant difference

We thank the reviewer for noticing it and indeed there is a statistical difference. Now we added statistical test analysis to the plots, please see updated version of figure 3.

Figure 3

Panel E,J,O. Can these be plotted with the same y axis to facilitate comparison?

Panel C,D,E. Can the authors label when the hyper osmotic shock occurs on the graph?

Figure 3 H & I.The plots look very similar, even in the SD. Please confirm that the NC data shown are correct.

Thank you for these suggestions. Following the reviewer's advice we updated Fig.3.

L.19 p.6

“nuclear and cytoplasmic volumes are strongly decoupled, with nuclear volume variations coherent with the changes of forces exerted on the NE and not with the changes of cytoplasmic volume (Fig. 4A).”

It would be important to test if NES-GFP localization is altered during these experiments.

Please find in Rebuttal-Fig1.C single plane confocal images of GFP-NES upon drug treatments. Moreover, the mean intensity ratio between cytoplasm and nucleus is not perturbed by drugs treatments.

L.21 p.7

"specific transcription factor such as YAP, key regulator of organ growth and regeneration as well as of mechanotransduction, also are affected by this mechanism."

Please cite a reference for this statement.

We apologize for this omission. We now add also here citation to ref.27 as it should be.

References

Update Lemiere reference #16

Fix Deveri reference #17

Fix Zimmerli reference #26

Thank you, we updated these references in the new version of our manuscript.

Rebuttal-Fig1. A. Left, widefield representative image of RPE1 cell expressing GFP-NES; highlighted in white the nucleus area; scale bar $20 \mu\text{m}$. Middle, average experimental radial profile of GFP-NES mean intensity in the nucleus area. Right, theoretical prediction of radial

profile. B. Cumulative distribution over several frames and different cells of R^2 relative to linear fits of pixels GFP-NES intensity in the cytoplasm in function of the corresponding pixel optical height computed with traditional FXm. C. Left, single plane confocal representative image of RPE1 cell expressing GFP-NES; highlighted in white the line used for the relative intensity scan; scale bar $20 \mu\text{m}$. Middle, line intensity profile relative to the image on the left. Right, cytoplasm to nucleus ratio of GFP-NES mean intensity for ctrl, Y-compound and latrunculin treated cells with relative representative images, scale bars $20 \mu\text{m}$. D. Left, simulations of time dependent cytoplasm and nucleus volume growth curves. Parameters used for the simulation were extrapolated from the fit of experimental grow speed of RPE1 cells (main Fig2.G). Middle, nucleus to cytoplasm volume ratio as obtained from the simulation. Right, average normalized cytoplasm and nucleus volume grow speeds. E. Conditional average of nuclear grow speed in function of cytoplasm volume.

REVIEWER COMMENTS

Reviewer #1 (Remarks to the Author):

The authors have responded to the points we raised. We strongly suggest that they make the following final modifications related to these points before the paper is accepted.

1. Page 1 of remarks "We apologize...": No apology is needed, but the science should be explicated. I appreciate that that authors distinguish a "naive" tension model from one that accounts for active transport in the Discussion. However, I feel that since the active model (which also includes the "naive" model in the limit that the activity is set to zero) was previously published that it be contrasted with the model of the authors in the Introduction. I believe that this is a more accurate representation of the field and the contribution of the present paper that is a significant advance, and that we already said should, after revision, be published in Nat. Comm.

2. Major comments: Point 1: Similar to the spirit of the previous point, we think the work of Finan should be discussed in the Introduction. It does not detract from the very nice results of the present paper if the previous research is openly presented.

3. Consistent with the spirit of points 1 and 2, we suggest that the authors explicate in the main text, the point they so nicely state in the response to Major comment: point 3, "We could say that nuclear.."

4. We still do not understand why a compressive force on the nucleus (assuming, as do the authors, that it is transmitted mostly to the lamina) should result in an increase in the tension. Consider the opposite case of an extensile force -- for example, that the acto-myosin cytoskeleton is pulling on the nucleus: that pulling force would tend to expand the lamina and thus increase its tension. Thus, a compressive force on the nucleus might tend to buckle the lamina proteins and thus decrease the tension. The energy would then go into the buckling of the protein network, but the tension would be decreased; this is indeed the case for many biopolymers where stretching "costs" energy, but where compression results in buckling the network. We apologize if we are confused about this, but if the referee is confused, it might be that the readers will not follow the argument and the logic should be carefully explicated or modified as noted.

5. Minor points: point 4: We read what the authors added to the supplementary information and have a further comment. The fact that the fluctuations decrease may not be due to an decrease in the tension due to activity, but rather, to an increase in the "fluctuating forces" that act on the system due to activity, over and above the thermal fluctuating forces. In other words, the tension might remain the same, but since the fluctuations depend on the ratio of the tension and the magnitude of the fluctuating forces, the fluctuations might increase due to an effectively higher temperature, and not a lowered tension due to activity. The authors can see, for example,

<https://journals.aps.org/prl/abstract/10.1103/PhysRevLett.106.238103>

The authors should consider and explicitly note this possibility.

Reviewer #2 (Remarks to the Author):

I am mostly satisfied with the way the authors addressed my concerns. I believe that the novel methodology and its subsequent application in this study grant publication in Nature Communications. This methodology will certainly be applied by other research groups and stimulate investigations on nuclear volume regulation.

Reviewer #3 (Remarks to the Author):

In general, there are several major concerns of this work that have not been suitably addressed in this revision. Most of the concerns were discussed but not directly addressed. In fact, the additional data and information provided in this round further reveal the weaknesses in the data. Thus, there are still significant reservations about many aspects of this manuscript, including the validity of the N2FXm assay and the distinction of linear vs exponential growth rate. This work unfortunately is not ready for publication in any journal.

Major concerns

1. The title, summary and parts of the abstract are not appropriate as key statements are not sufficiently backed by the data in this paper.

-The authors revised the text slightly to remove the implications that this work shows direct effects on nuclear pore size. However, the current title is still inappropriate as it places emphasis on elements that have not been sufficiently tested in this work. As remarked in the previous round, the involvement of nuclear import (the conclusion of the title) is supported only by an effect of ivermectin, an inhibitor of α/β importin- based nuclear transport. Broader effects of actin inhibition are not considered. Thus, there is insufficient experimental data to make this the take-home message. To do so would require new experiments that directly document a change in nuclear import rate that is dependent on nesprin and its ability to bind to actin at the nuclear envelope.

2. There are important concerns about the N2FXm method that need to be addressed. Additional data and more detailed descriptions of the calibration method are needed so that the accuracy of the method can be evaluated (see below).

-The authors have not provided sufficient additional data to address the multiple concerns of this method. There is still little data regarding its accuracy.

-Issue of organelle volume in the cytoplasm was not addressed.

-Is GFP-NES really excluded from the nucleus? Although the data in rebuttal Figure 1 is used for the authors to claim that this is not an issue, the data in rebuttal Figure 1 AC clearly shows measurable non-zero intensity in the nucleus in the confocal imaging in the image and the graph. The modeling in rebuttal Figure 1A does not rule out GFP-NES in the nucleus, and the graph in Figure 1A shows a non-zero intensity at the base of the graph. As the authors say, the leakage of GFP-NES into the nucleus would lead to inaccurate measurement of nuclear volume. For instance, it might explain the unexplained discrepancy between the measurements using confocal 3D reconstructions and the N2FXm.

-Do GFP-NES levels in the nuclear region change during drug or mechanical perturbations? The new data in rebuttal Figure 1C show that the distribution of intensity values certainly change (at least the variability increases) suggesting that at least for a good portion of cells that this might be a real issue. Changes in GFP-NES in the nucleus would further raise serious questions into the validity of the measurements.

3. There are also concerns about the FRET-based nuclear envelope force sensor.

-The authors should clarify the description of the force sensor. This description could be similar to what they explained to reviewer 1: "The FRET tension sensor we developed is designed on Nesprin1, one of the components of the LINC complex. It spans from the inner nuclear envelope, where it binds SUN domain containing proteins, to the cytoplasm, where it binds to actin. Therefore, it is sensitive only to forces exerted by the actin cytoskeleton to the NE." Importantly, it should be clarified that this FRET probe most likely does not sense membrane tension (parallel to the membrane) but rather forces by actin directly attached to the Nesprin probe (perpendicular to the membrane).

4. One of the most surprising results is the linear growth of the nucleus vs. exponential growth of the cell volume. Other than pointing out the difference, this is not further investigated. More data are needed to confirm and develop this observation.

-New data on this point only further questions of the validity of this claim of linear growth. The graphs used to support linear growth of the nucleus (Figure 2G and rebuttal Figure 1E) are inconsistent with this conclusion. Rebuttal Figure 1E shows that the points for each cell type simply do not obviously fit a flat line (linear) or a line with positive slope (exponential). In Figure 2G, this deviation from a flat line is hidden by inappropriate choice of Y-axis values but is apparent by close examination (Note that this

graph shows the same points as Rebuttal Figure 1E). Hence, the data do not support their conclusion that nuclear growth is linear, which is considered a major take-home of this work.

Other issues

Figure 1D. The Y-axis was changed in response to reviewers. However, the axis change was very crudely done, which in effect visually changes the values of the graph.

The information on error bars, number of replicates, N must be provided for every experiment shown.

Many details in the methods seem to be lacking. For instance, what were the concentration and time of treatment of ivermectin? How were the osmotic shocks performed.

Reviewer #1 (Remarks to the Author):

The authors have responded to the points we raised. We strongly suggest that they make the following final modifications related to these points before the paper is accepted.

We appreciate the recognition of our work during the revision process. We would like to express our gratitude to the reviewer for her/his helpful suggestions, which have significantly contributed to the improvement of our work.

1. Page 1 of remarks "We apologize...": No apology is needed, but the science should be explicated. I appreciate that that authors distinguish a "naive" tension model from one that accounts for active transport in the Discussion. However, I feel that since the active model (which also includes the "naive" model in the limit that the activity is set to zero) was previously published that it be contrasted with the model of the authors in the Introduction. I believe that this is a more accurate representation of the field and the contribution of the present paper that is a significant advance, and that we already said should, after revision, be published in Nat. Comm.

As suggested, we extended the introduction of the revised version of our manuscript. Now we added this sentence:

"For mammalian cells, the commonly accepted model considers cytoplasm and nucleus in mechanical equilibrium where the dominant forces are due to osmotic pressure, and nuclear envelope tension counteracting the growing tendency of the nucleus (9). Interestingly, a previous study on pig chondrocytes has already observed a differential behavior of the nucleus with respect to the cytoplasm in response to an osmotic perturbation, suggesting a possible decoupling of the two compartments, and described it by a model of a spherical gel with a membrane stress term subjected to osmotic loading (10)."

2. Major comments: Point 1: Similar to the spirit of the previous point, we think the work of Finan should be discussed in the Introduction. It does not detract from the very nice results of the present paper if the previous research is openly presented.

Please see the reply to the previous point.

3. Consistent with the spirit of points 1 and 2, we suggest that the authors explicate in the main text, the point they so nicely state in the response to Major comment: point 3, "We could say that nuclear.."

We thanks the reviewer for the suggestion and we added in the discussion of our revised manuscript this paragraph:

"More widely, nuclear envelope stress could have two possible independent effects, which can act on different timescales: one on nuclear import (and osmotic equilibrium), which we focus on in our study, which makes nuclear volume increase when the envelope is more stretched compared to a reference state; and one, that has a direct mechanical effect, and balances the difference of osmotic pressure between the nucleus and cytoplasm, by which

increasing tension leads to a decrease of nuclear volume for a given osmotic balance at steady state. In our study, due to cell spreading, we can hypothesize that we are in a regime of low effective constitutive tension, and the import effect dominates. A comprehensive understanding of how the combination of these complex opposite effects determine nucleus volume growth during the cell cycle would require further experiments and a clearer picture of the underlying mechanisms.”

4. We still do not understand why a compressive force on the nucleus (assuming, as do the authors, that it is transmitted mostly to the lamina) should result in an increase in the tension. Consider the opposite case of an extensile force -- for example, that the acto-myosin cytoskeleton is pulling on the nucleus: that pulling force would tend to expand the lamina and thus increase its tension. Thus, a compressive force on the nucleus might tend to buckle the lamina proteins and thus decrease the tension. The energy would then go into the buckling of the protein network, but the tension would be decreased; this is indeed the case for many biopolymers where stretching "costs" energy, but where compression results in buckling the network. We apologize if we are confused about this, but if the referee is confused, it might be that the readers will not follow the argument and the logic should be carefully explicated or modified as noted.

We apologize for the lack of clarity in the first rebuttal letter. For compressive cytoskeletal forces acting on the nucleus we intend forces that squeezing the nucleus as a whole, like compressing it from the top, would then generate an overall increase of NE tension. We agree with the reviewer that pointy acting forces, pinching inward the NE, are expected to buckle it and generate a local decrease of tension. It is difficult to figure out if these kinds of local forces are sufficient to then generate global effects. For the scope of our work, it is important to highlight and reiterate that direct forces, both positive and negative, are unlikely to deform the nucleus, the size of which is instead mainly set by mechano-regulated osmosis.

5. Minor points: point 4: We read what the authors added to the supplementary information and have a further comment. The fact that the fluctuations decrease may not be due to a decrease in the tension due to activity, but rather, to an increase in the "fluctuating forces" that act on the system due to activity, over and above the thermal fluctuating forces. In other words, the tension might remain the same, but since the fluctuations depend on the ratio of the tension and the magnitude of the fluctuating forces, the fluctuations might increase due to an effectively higher temperature, and not a lowered tension due to activity. The authors can see, for example, <https://journals.aps.org/prl/abstract/10.1103/PhysRevLett.106.238103>

The authors should consider and explicitly note this possibility.

This is exactly what we meant (see e.g. the introduction of Introini et al. 2013, where this point is explained in detail). We apologize for the lack of clarity, we modified the relative paragraph of the Appendix as follows:

“The apparent tension of the nucleus, estimating σ_N or σ_0 is found to be around 10^{-6} N/m in nuclear shape fluctuations experiments (Chu et al., 2017; Introini et al., 2023). However, this value is an underestimation of the actual tension, since it is affected by excess flickering due to transient deformations of active origin, since fluctuating forces due to activity decrease the effective tension with respect to the actual mechanical tension (see Introini et al., 2023 for a

detailed explanation). We can assume that this value is a lower bound for the real mechanical tension σ_0 ."

Reviewer #2 (Remarks to the Author):

I am mostly satisfied with the way the authors addressed my concerns. I believe that the novel methodology and its subsequent application in this study grant publication in Nature Communications. This methodology will certainly be applied by other research groups and stimulate investigations on nuclear volume regulation.

We appreciate the recognition of our work during the revision process. We would like to express our gratitude to the reviewer for her/his helpful suggestions, which have significantly contributed to the improvement of our work. Additionally, we are pleased by the reviewer's enthusiasm for our project and her/his recognition of the potential impact it could have in the community.

Reviewer #3 (Remarks to the Author):

In general, there are several major concerns of this work that have not been suitably addressed in this revision. Most of the concerns were discussed but not directly addressed. In fact, the additional data and information provided in this round further reveal the weaknesses in the data. Thus, there are still significant reservations about many aspects of this manuscript, including the validity of the N2FXm assay and the distinction of linear vs exponential growth rate. This work unfortunately is not ready for publication in any journal.

We would like to express our gratitude to the reviewer for her/his helpful suggestions, which have significantly contributed to the improvement of our work.

We are sorry but we disagree with the reviewer. The new quantifications and data provided in the first rebuttal letter, are not affecting the conclusions of our work and moreover they confirmed the observed decoupling between cellular and nuclear volume growth dynamics.

Major concerns

1. The title, summary and parts of the abstract are not appropriate as key statements are not sufficiently backed by the data in this paper.

Following editor and reviewer suggestion we propose to change the working title of our manuscript in: "Differential osmo-mechanical regulation of nuclear and cytoplasmic volume in mammalian cells".

We also toned down the abstract.

-The authors revised the text slightly to remove the implications that this work shows direct effects on nuclear pore size. However, the current title is still inappropriate as places emphasis on elements that have not been sufficiently tested in this work. As remarked in the previous round, the involvement of nuclear import (the conclusion of the title) is supported only by an effect of ivermectin, an inhibitor of α/β importin- based nuclear transport. Broader effects of actin inhibition are not considered. Thus, there is insufficient experimental data to make this the take-home message. To do so would require new experiments that directly document a change in nuclear import rate that is dependent on nesprin and its ability to bind to actin at the nuclear envelope.

As stated in the original rebuttal letter, it is not the scope of our work to demonstrate the existence of a tension-biased nuclear transport. This mechanism has been conceived and quantitatively described in two seminal papers from Roca-Cusachs lab (Elosegui-Artola et al., Cell 2017 and Andreu et al., NCB 2022). Moreover, two structural papers from different labs also suggest that cyto-skeletal tension could stretch nuclear pores (Zimmerli et al., Science 2021 and Schuller et al., Nature 2021). Although of course it is possible that other processes are in place, this evidence is well in line with Roca-Cusachs model. In our work we rely on this well established line of literature. In Figure 4C, we also show that GFP-NLS nuclear to cytoplasmic ratio, here used as a marker of nuclear "permeability", changes in function of nuclear envelope tension as measured by our sensor. Since the reviewer

suggested to not state a direct link from NE tension, dilatation of nuclear pores and nuclear permeability, we avoided that in the revised version of our manuscript. We still mention it as a possible speculative mechanism. However, the fact that NE tension impacts on nuclear volume, besides being related to past literature, is clearly supported by our data and is an essential element of the model we proposed. We also showed that, to our knowledge, forces required to directly extend the nucleus are out of scale for the cell (see reply to reviewer 1 minor point 3).

2. There are important concerns about the N2FXm method that need to be addressed. Additional data and more detailed descriptions of the calibration method are needed so that the accuracy of the method can be evaluated (see below).

-The authors have not provided sufficient additional data to address the multiple concerns of this method. There is still little data regarding its accuracy.

-Issue of organelle volume in the cytoplasm was not addressed.

As we show in the original rebuttal letter (see reply to reviewer 2 point 1-i and reviewer 3 major 2), the linear correlation between GFP-NES signal and cytoplasm optical height, as quantified with standard FXM, is very good (see rebuttal Figure 1 B). Any inhomogeneity in the distribution of GFP-NES signal in the cytoplasm, as could be generated by cytoplasmic organelles, would have perturbed this distribution.

-Is GFP-NES really excluded from the nucleus? Although the data in rebuttal Figure 1 is used for the authors to claim that this is not an issue, the data in rebuttal Figure 1 AC clearly shows measurable non-zero intensity in the nucleus in the confocal imaging in the image and the graph. The modeling in rebuttal Figure 1A does not rule out GFP-NES in the nucleus, and the graph in Figure 1A shows a non-zero intensity at the base of the graph. As the authors say, the leakage of GFP-NES into the nucleus would lead to inaccurate measurement of nuclear volume. For instance, it might explain the unexplained discrepancy between the measurements using confocal 3D reconstructions and the N2FXm.

In rebuttal Figure 1 A we show the radial distribution of the GFP-NES signal in the nucleus in wide-field images (the same used for N2FXm). It is clear that the signal is minimum at the center and increases toward the nuclear periphery. That is not compatible with a background of nuclear GFP-NES, that would be rather constant or maximum at the center and minimum at the periphery. On the contrary, this result is perfectly in line with what would be expected from a signal coming from the cytoplasm and then integrated over the cytoplasmic volume surrounding the nucleus. Clearly, the signal doesn't start from "zero" (as is not zero in confocal scan). This non zero line, could correspond to the portion of cell volume on top and below the nucleus center, or, as we clearly stated, could also be originated by nuclear GFP-NES. However, the impact of this background is minimal and does not affect our measurements. Indeed, as summarized in the rebuttal letter, we challenged our method with two independent controls. First, we compared population measurements of nuclear volumes performed with N2FXm and standard 3D confocal reconstructions, and we didn't record a significant shift in the distributions. Second, we measured in living cells the volume of artificial spherical objects (DAAMs particles). N2FXm measurements were in good agreement with theoretical volume computed by geometrical reconstruction.

-Do GFP-NES levels in the nuclear region change during drug or mechanical perturbations? The new data in rebuttal Figure 1C show that the distribution of intensity values certainly change (at least the variability increases) suggesting that at least for a good portion of cells that this might be a real issue. Changes in GFP-NES in the nucleus would further raise serious questions into the validity of the measurements.

We apologize for lack of clarity, we did not specify that we statistically tested the difference, and it is not significant (t-test vs Ctrl, p-values: Y-comp=0.140; Lat=0.0618).

3. There are also concerns about the FRET-based nuclear envelope force sensor.

-The authors should clarify the description of the force sensor. This description could be similar to what they explained to reviewer 1: "The FRET tension sensor we developed is designed on Nesprin1, one of the components of the LINC complex. It spans from the inner nuclear envelope, where it binds SUN domain containing proteins, to the cytoplasm, where it binds to actin. Therefore, it is sensitive only to forces exerted by the actin cytoskeleton to the NE." Importantly, it should be clarified that this FRET probe most likely does not sense membrane tension (parallel to the membrane) but rather forces by actin directly attached to the Nesprin probe (perpendicular to the membrane).

We thank the reviewer for this suggestion and we improved the description of the sensor in the revised version of our manuscript accordingly. We now also clearly state its possible limitations. As suggested we added in the main text: "This tension sensor spans from the inner NE, where it binds SUN domain containing proteins, to the cytoplasm, where it binds to actin. Therefore, it is sensitive only to forces exerted by the cytoskeleton to the NE and does not sense tension parallel to the membrane (23)."

4. One of the most surprising results is the linear growth of the nucleus vs. exponential growth of the cell volume. Other than pointing out the difference, this is not further investigated. More data are needed to confirm and develop this observation.

-New data on this point only further questions of the validity of this claim of linear growth. The graphs used to support linear growth of the nucleus (Figure 2G and rebuttal Figure 1E) are inconsistent with this conclusion. Rebuttal Figure 1E shows that the points for each cell type simply do not obviously fit a flat line (linear) or a line with positive slope (exponential). In Figure 2G, this deviation from a flat line is hidden by inappropriate choice of Y-axis values but is apparent by close examination (Note that this graph shows the same points as Rebuttal Figure 1E). Hence, the data do not support their conclusion that nuclear growth is linear, which is considered a major take-home of this work.

Rebuttal Figure 1E was produced to reply to a reviewer 3 minor point. Please, note that this plot does not show at all the same points of figure 2G. Here, conditional average is not performed over nuclear volume, but, as requested by the reviewer, over cell volume. Hence, the two plots are different, and they address two different questions. Nuclear Volume Growth Speed is indeed here averaged over bins of cell volume and not over bins of nuclear volume. Nothing is hidden by inappropriate scaling. Interestingly, the shape of the curve in Rebuttal Figure 1E is not trivial and we thank the reviewer for pointing our attention to this point. While it deserves more investigation, we can surely conclude that this curve is clearly more similar to a flat line than to an asymptotically growing one. These results are not affecting the

conclusions of Figure 2G. Moreover, they confirm a robust decoupling of nuclear and cell growth dynamics in single cells, a main point of our work. Regarding the flat line fits in Figure 2G and Supplementary Figure 2B, distributions of fit residuals are normal and their means are not statistically different from zero.

Other issues

Figure 1D. The Y-axis was changed in response to reviewers. However, the axis change was very crudely done, which in effect visually changes the values of the graph.

We apologize, our mistake. When we adjusted the size of text in the plot axis assembling the figure we shifted it by a few pixels. We thank the reviewer for spotting this mistake and we corrected it in the revised version of our manuscript.

The information on error bars, number of replicates, N must be provided for every experiment shown.

All this info is now in figure legends or in the method session.

Many details in the methods seem to be lacking. For instance, what were the concentration and time of treatment of ivermectin? How were the osmotic shocks performed.

This information was in the Supplementary Materials. In the revised version of our manuscript we renamed the relative section as: "Drug, adhesion and osmotic perturbations experiments". As stated, cells were treated with ivermectin (Sigma Aldrich) at 10 μ M for 24 h. For hyper-osmotic shock experiments, D-sucrose (Carlo Erba reagents) was resuspended in pure water at 500 mM and then added to cells.

Reviewer #1 (Remarks to the Author):

This paper experimentally explores the effects of various biochemical and mechanobiological perturbations of single cells on the ratio of their nuclear to cytoplasmic volumes. It is shown that changes in cytoskeletal forces exerted on the nucleus (via latrunculin or ROCK inhibition) changes the NC ratio in a manner that is not simply due to the direct effect of those forces on the osmotic pressure balance. In addition, it is shown that while the NC ratio in many cell types (under hyperosmotic conditions) is constant during most of interphase, this ratio varies considerably during cell division. Interestingly, it is found that right after cell division, the cytoplasmic volume increases exponentially, while the nuclear volume increases only linearly with time. The results will stimulate further experiments and theoretical models and should be published in Nature Communications, pending the following changes.

We are grateful for the positive remarks, and for the positive recommendation.

We believe that the authors overstate the impact of their results with respect to the concepts presented in Refs. 7, 16, and 17. In particular, the present manuscript uses the term “falsify” (once) “or “falsified” (twice) to relate their findings to the presumably naïve “osmotic” models in these references. This, besides being polemical, is not correct and does not reflect the subtlety of the ideas previously presented. While the NC volumetric ratio is indeed determined by osmotic pressure differences, the proteins involved are those whose active transport through the nuclear pores, results in differing concentrations in the cytoplasm and nucleus. The import rate and export rates can be different for systems with active transport, resulting in concentrations of proteins that are directly related to these rates. When various perturbations are applied to the cell, including changes in the cytoskeletal forces that act on the nucleus, those transport rates are modified, with – in general – different changes for the import and export rates. Thus while it is true that the naïve osmotic effect of changes in cytoskeletal force are not significant in the experiments described, there is an important indirect effect of those forces on the transport rates that determine the protein concentrations in the nucleus and cytoplasm, and hence the osmotic pressure difference and NC ratio. This was indeed a major point of these references which seems to have been missed in the present manuscript.

Nevertheless, the paper is interesting and presents several novel results about how the NC ratio changes during the cell cycle and under biochemical or mechanobiological changes, which justify publication in Nature Communications. However, before we can formally recommend publication, the interpretation of these changes with respect to such perturbations should be rewritten to accurately incorporate the previously discussed changes in osmotic pressures due to active transport and to qualitatively relate these transport changes to the perturbations applied in the present manuscript. In addition, we request the authors to address the following major and minor comments.

We apologize to the reviewer for our miscommunication. Our intent was not to go against previous models (which of course had different scopes and were not informed by our data), and we meant our statement referred exclusively to the ingredients of our own model, which is a naive version of the osmotic model, taking into account just few essential ingredients

and making some simplifications in order to keep a minimal number of parameters. We have rephrased the statements in order to avoid any misunderstanding, and also added a statement on the results of the cited refs. Specifically we added this sentence in the discussion section: “When perturbations are applied to the cell, including changes in the cytoskeletal forces that act on the nucleus, transport rates can be modified, with – in general – different changes for the import and export rates. Thus while it is true that the naïve osmotic effect of changes in cytoskeletal force are not significant in our experiments, there is an important indirect effect of those forces on the transport rates that determine the protein concentrations in the nucleus and cytoplasm, and hence the osmotic pressure difference and NC ratio. This was a major point of previous studies [REFS 7-16-17].”

Major comments:

1. At the bottom of page 5, the authors write “Our model also shows that the same behavior is expected for a non-negligible constant surface tension...”. Compared to what scale is the surface tension non negligible? The work by Finan et al. (Annals of Biomedical Engineering, 2009) presents the notion of NE that is stretched compared to one that is relaxed. The authors should relate the findings of their model with respect to the work of Finan et al.

We thank the reviewer for this comment and we refined the inaccurate statement in the current version of the manuscript. We have explored constitutive surface tensions in the range $1e^{-6}$ to $1e^{-2}$ N/m. The value around $2e^{-2}$ N/m found by Finan and coworkers by a fit of their model can be compared with the upper bound of the constitutive surface tension that we explored. Hence, our results should apply (see e.g. fig SM3 in the SI appendix). It is important to state that their richer model (based on the swelling of porous gels) contains other parameters, hence although the basic underlying physical picture of our simpler model is similar, it is possible that the values of the parameters do not map exactly between the two frameworks. Finally, we note that while we explore for simplicity our tension-biased transport model in a regime of negligible tension, this choice has no impact on our main statement, as the choice of a finite/large constitutive tension would make it equally difficult for osmotic and mechanical extensile forces to swell the nucleus.

2. The authors conducted three types of experiments to modulate the forces that are exerted on the cell and are transduced into the nucleus: hyperosmotic shock, cell detachment, and cell spreading. These types of experiments were thoroughly investigated in previous works such as Finan et al. (mentioned above), Guo et al. (ref. 19 in the manuscript), and others. However, discussion that relates the results of the manuscript to these previous works is lacking. The authors should present their experimental results in a wider context and relate them to these previous works.

The work of Finan and coworkers has the merit of showing that the volume increase of the nucleus upon hypo-osmotic shocks, saturates and does not follow a linear increase as the cell volume does (Ponder plot). The simplest interpretation of that is that part of the pressure is compensated by nuclear envelope tension, leading to less volume increase than expected when tension is negligible.

We could say that nuclear envelope stress can have two effects, which can act on different timescales: one on nuclear import, which we focus on in our study, which makes nuclear volume increase when the envelope is more stretched compared to a reference state, and one that has a direct mechanical effect, and balances the difference of osmotic pressure between the nucleus and cytoplasm, by which increasing tension leads to a decrease of nuclear volume for a given osmotic balance at steady state. In our study, due to cell spreading, we can hypothesize that we are in a regime of low effective constitutive tension, and the import effect dominates.

3. At the bottom of page 37, the authors relate the tension to external force by the equation $F_{ext} = 8\pi RN \sigma_{ext}$. The authors should explain this relation as its derivation is unclear. Perhaps a cartoon showing the force, tension, and the relation between them can make the derivation clearer.

We apologize for the missing elements in our argument. We have clarified the assumptions, and we have specified in the revised text that we intend this relation as a simple estimate relating a scenario of tension of external origin to the total magnitude of the radial forces that are needed to generate the equivalent mechanical pressure difference.

4. The authors model the effect of force on nuclear import using an exponential relation. The author should discuss the underlying biophysical reasoning of this dependence or state it is just a fitting function.

We agree that this assumption needs an explanation - which was pointed out also by reviewer 2. We have made it clear in the text that we have assumed an activated process. This assumption does not really affect the quantitative aspects, since other functional forms would be equivalent in the relevant range of values.

5. In their theoretical model, the authors treat the external force as a scalar although by definition force is not a scalar. Furthermore, the authors assume that the nucleus is spherically symmetric, which is not true in general, when deformed by non-isotropic external forces (that may have shear components).

We have clarified these aspects. We chose the spherically symmetric case to simplify the model from parameters that could not be fixed reliably based on our data. Given the assumption, the force field only has one relevant degree of freedom, hence the sloppy notation (now clarified). Additionally, we have clarified that we mean the model in this study more as a conceptual guide to understand the experiments than as a full-blown theoretical description. The purpose is to isolate and select scenarios, hence we systematically reverted to the simplest assumptions.

6. The authors state that they assume the forces that are exerted on the nucleus are compressive. In that case and when the forces are spherically symmetric, the contribution of the external forces to the tension should be negative. Is this the case? This should be clearly explicated since this affects the way the exponential relation between the surface tension and nucleocytoplasmic ratio is interpreted.

We have clarified this part in the revised SI text. In our notation the contribution of compressive external forces to the tension is positive (as they both contribute to forces pointing inwards). We analyzed both cases of positive and negative contribution of external forces to the tension, but given the previous experimental literature [e.g. <https://doi.org/10.1073%2Fpnas.0908686106>] we believe that the first scenario could be more likely.

Minor comments:

1. Can there be forces that are exerted on the nucleus which will not be detected by stretching of the LINC complex (e.g. squashing of the nucleus)?

This observation is correct. The FRET tension sensor we developed is designed on Nesprin1, one of the components of the LINC complex. It spans from the inner nuclear envelope, where it binds SUN domain containing proteins, to the cytoplasm, where it binds to actin. Therefore, it is sensitive only to forces exerted by the cytoskeleton to the NE. If external forces applied to the cell indirectly also affect the link between actin and SUN proteins at the NE, then the sensor can detect it. More detail about the FRET sensor can be found in Poli et al 2023 [<https://doi.org/10.1038/s41467-023-37064-0>]. Please consider also our response to point 3 below.

2. Some works (e.g. Jahed et al., Journal of Cell Science 2016) suggest that the cytoskeleton and LINC complex are indirectly involved in nucleocytoplasmic transport by modulating the Ran protein gradients. Is it possible that the nucleocytoplasmic transport in the latrunculin-treated or the ROCK inhibited cells is affected via this pathway rather than mechanical “expansion” of the NPCs? Can the author comment on this possibility.

We do not exclude that there may be additional/other mechanisms at play, as suggested by the reviewer. However, we have observed a strong correlation between nuclear envelope tension, as measured by our FRET sensor, and the mean intensity ratio of GFP-NLS between the nucleus and cytoplasm (used as a marker of protein nuclear incorporation efficiency) (Fig. 4C). This dependency is essential for our model, independently of whether this is due to nuclear pores expansion or to other possible mechanisms.

3. At the bottom of page 6 the authors write that the cytoskeletal forces are compressive. However, these can also be expansive if the nucleus is laterally stretched. Can the authors provide a reference to a work that distinguish between these two possible modes of nuclear deformation?

The reviewer’s observation is correct. It could be, in certain circumstances, that positive forces are exerted to the nucleus. With our model, indeed, we analyzed the effect on nuclear volume of positive as well as negative external forces (always with the assumption of a spherically symmetric nucleus). In both cases variations in nuclear volume induced by direct forces are negligible with respect to the ones arising from changes in osmolarity (less than 10% for σ_0 in the range 10^{-6} to 10^{-2} N/m).

We now modify that sentence in: “Based on the model, we reasoned that this was very unlikely to be a direct effect, as, independently of the direction, compressive or stretching, the force required to change nuclear size would be abnormally large (see SI appendix) [24, Kalukula et al, Nature Reviews Molecular Cell Biology, (2022), 583-602, 23(9), DOI: <https://doi.org/10.1038/s41580-022-00480-z>].”

4. The authors mention that the estimate of the NE tension (10^{-6} N/m) is a lower bound due to fluctuations of active origin of the NE shape. Are there evidence for such active fluctuations? While this does not change the conclusion of the authors, it is worthwhile to give a reference for such active processes that facilitate NE shape fluctuations.

We agree and we have added this statement: “The apparent tension of the nucleus, estimating σ_N or σ_0 is found to be around 10^{-6} N/m in nuclear shape fluctuations experiments (Chu et al., 2017; Introini et al., 2021).”

Reviewer #2 (Remarks to the Author):

Pennacchio et al extended the powerful fluorescence exclusion microscopy, FXm, to simultaneously measure cell volume and nuclear volume – a new method they coined “N2FXm”. This novel method is sound and allowed the authors to study the nuclear-to-cell volumetric (NC) ratio. First, they found that the NC ratio was not constant during the cell cycle, being particularly impacted by mitosis. Second, they showed that the cell and nucleus did not follow the same growth laws. Third, they discovered that the NC ratio could not be explained solely by an osmotic balance and that cytoskeletal forces exerted onto the nuclear envelope could bias nuclear import and nuclear osmotic pressure, thus also participating in setting the NC ratio.

The manuscript is clearly written, and the data are of high quality and interest, shading light on novel regulatory mechanisms of nuclear volume. I would greatly recommend publication should the following questions be addressed.

We thank the reviewer for the overall positive judgment of our work and for the enthusiastic recommendation.

1/ The N2FXm method

I found the N2FXm method extremely powerful and relevant. Although most of my concerns are addressed in the methods or in the paper, I have one additional question and one recommendation for the authors:

i) Have the authors checked that there is no GFP-NES in the nucleus, and if the signal is homogeneous in the cytosol? Depending on the cell state/strength of the NES, there may still be some GFP in the nucleus, which would affect the measurement – especially if this bias is cell cycle-dependent, through for instance NE tension impacting nuclear export. Even if non-significant, there seems to be a small difference in the nuclear volume measurement between the 3D reconstructed and the N2FXm in Fig. S1F. Moreover, this could also be important in the perturbation experiments, impacting the actual measurement.

We thank the referee for these relevant remarks, which were also raised by reviewer 3. GFP-NES background in the nucleus, indeed, could affect the precision of N2FXm. Similarly, dishomogeneity in the distribution of GFP-NES in the cytoplasm, could perturb the second calibration, then impacting on nuclear volume estimates. We now explicitly discuss these possible drawbacks in the revised version of our manuscript. Additionally, following the reviewer's advice, we evaluated the distribution of the GFP-NES signal in correspondence of the nucleus (Rebuttal-Fig1.A). The mean radial intensity profile is minimum at the center and increases toward the periphery. In low magnification widefield images, the signal background simply originated by the presence of GFP-NES in the nucleus is expected to be rather constant or proportional to nucleus depth, then maximum at the center and minimum at the periphery. The profile we measured (middle panel in Rebuttal-Fig1.A) is instead compatible with a signal coming from the cytoplasm and then integrated over the cytoplasmic volume surrounding the nucleus. In the third plot in Rebuttal-Fig1.A we show the expected theoretical radial profile for a spherical nucleus (of unitary radius) embedded in a labeled cytoplasm:

$$f(r) = \int_0^r (1 - \sqrt{1-x^2}) dx = r - \frac{1}{2} * \arcsin(r) - \frac{1}{2} * r * \sqrt{1-r^2}.$$

This is compatible with the experimental profile we measured.

Importantly, we cannot exclude that a small portion of the signal we record in correspondence to the nucleus comes from inside the nucleus. Indeed we recorded a background line on the radial profile that could have originated both by GFP-NES on top and bottom of the nucleus or by GFP-NES in the nucleus. However, when we evaluate in a confocal section (Rebuttal-Fig1.C) the proportion between cytoplasmic and nuclear GFP-NES signal, it becomes clear that the majority of the signal comes from the cytoplasm. In the mid-nucleus confocal sections, the mean intensity of GFP-NES in the cytoplasm is approximately 5 times higher than in the nucleus (Rebuttal-Fig1.C). Importantly, this ratio is not perturbed by drug treatments (Rebuttal-Fig1.C).

To allow the reviewers to judge directly if and how inhomogeneity in the distribution of GFP-NES signal in the cytoplasm could impact our measurements we plotted a representative example of a cumulative distribution of R^2 values, relative to more than 150 frames over 10 cells, for linear fits between GFP-NES cytoplasmic intensities and calibrated optical heights (Rebuttal-Fig1.B and ref to Figure 1.C). R^2 distribution is strongly biased towards 1, median = 0.93, sd = 0.14. Significant inhomogeneity in the distribution of GFP-NES signal, such as local accumulations or depauperation, would have perturbed the linear dependency between these two quantities and then the distribution of R^2 values.

All together these considerations suggest that GFP-NES background in the nucleus and as well as possible dishomogeneity in the distribution of GFP-NES in the cytoplasm and its impact on second calibration, are neglectable/minimal. This is probably due to the 'low' sensitivity and resolution of the imaging system settings as they are necessary for the N2FXm method to properly work, basically low numerical aperture and low magnification objective. Actually, it is important to notice that we challenged our method with two independent controls. First, we compared our nuclear volume measurements with 3D confocal reconstruction of nuclei, and we didn't record a significant shift in the distribution of nuclear volume estimates of the same cell population. Second, we measure in living cells the volume of artificial spherical objects (DAAMs particles). N2FXm measurements were in good agreement with theoretical volume computed by geometrical reconstruction. These two complementary validations well support the goodness of our method.

ii) I believe one important point is that the GFP calibration is done at every time point and for each cell, thus limiting temporal fluctuations or cell-cell heterogeneity in GFP concentration. As such, it may be better to directly state it in the main text, and not only in the methods.

We thank the reviewer for this comment and as suggested we now specified it in the main text.

2/ Nuclear growth and NC ratio

I have a few questions regarding nuclear volume growth and subsequent NC ratio values, as well as their interpretations based on the model proposed by the authors:

i) The NC ratio seems rather constant in each cell line from -500 min to 0 min. First, can the authors provide a y-axis zoom before and after mitosis, to have a better visualization (Figs. S1L & 1F)? Second, it seems constant even though nuclear volume does not increase exponentially. Can the authors comment on this point?

To enhance the difference in NC ratios we showed it at its maximum, between PME (post mitotic expansion, nucleus life start) and NEB (nuclear envelope breakdown, nucleus life end), please see plot in Fig2.F. NC ratios are not constant and typically increase by at least 20% along the nucleus lifetime. We added this quantification in the main text of the revised version of our manuscript. As suggested, we also added in Fig2.F a graphical representation of relative paired t-tests results.

ii) Can the authors speculate (and potentially expand the discussion on this point) as to why the nucleus would grow linearly in time? Per the authors' model, if it does not increase exponentially, this could for instance mean either that import/export is cell cycle-dependent such that nuclear osmotic pressure is not constant over the cell cycle. This could for instance be the case if NE tension changes during the cell cycle (see the question on tension in point 3/ below).

The average linear growth on cell-cycle time scale is an intriguing fact that we report experimentally, but whose explanation goes beyond our scopes. Most of our data focus on effects on time scales of minutes to less than an hour, and our explanations are fully coherent on these time scales. Cell-cycle time scales involve different processes and likely different explanations are necessary.

This said, we can reason on how the data relate to the prediction of different simple models.

Let us suppose first a scenario of perfect osmotic force balance. If the tension were always negligible, the volume of the nucleus would grow exponentially as the total volume. Our analyses suggest that the tension would have to be too large to have a relevant impact on nuclear size and explain the average linear growth.

If we consider our model coupling tension with transport, we can ask whether extrapolating it to cell-cycle time scales would support linear growth. In this model the observed tension with cell cycle would increase nuclear import, causing the nucleus to swell more than predicted by osmotic balance. Initially, it would grow faster than the cytoplasm, as it does in our data (please see also the reply to point 4 of Reviewer#3). However, without more specific information our model cannot be easily used to suggest that this growth would be linear, which will require a specific and gradual release of external tension.

Recently, Rollin and coworkers have proposed a mechanism that could partly explain this trend (<https://www.biorxiv.org/content/10.1101/2022.08.01.502021v1.full>). In this preprint the authors claim a role of counterion release by chromatin folding. According to their model, the NC ratio (formula 15) can be intermediate between an expression that is the ratio between nuclear proteins and cytoplasmic proteins (NC1, the prediction of pure osmotic force balance) and one (NC2, larger) that depends on DNA charge, which is constant during

G1 phase, while the number of proteins in the nucleus grows with time, so NC2 decreases with time. Hence, this prediction, valid only in G1, could generate the right qualitative trend.

More widely, nuclear envelope tension might have at least two effects (acting on different timescales): one on nuclear import, which we focus on here, which would make the volume increase when the envelope feels more tension, while one direct mechanical effect would lead to a decrease of nuclear volume for a given osmotic balance at steady state. For high enough nuclear envelope tension, the volume loss effect might dominate, while for low tension (as in our study, due to cell spreading), the import effect might dominate.

Altogether, these synergies of complex effects make it hard to predict the volume growth of the nucleus during the cell cycle: a complete model would be needed, plus many more experiments.

iii) Related to the previous point, it does seem that, after the spreading phase (phase III), there is no or little growth of the nuclear volume, followed by more rapid growth. This seems true for the cell lines in Fig. S1L. Has this been considered in the calculation of volumetric growth? Could it be that the binning hides different growth regimes of the nucleus during the cell cycle?

We agree with the reviewer that at faster time scales, there may be several intriguing cell-cycle dependent phenomena that we missed in this study. We hope to perform higher-resolution time-resolved studies in the near future to address this and other questions. In particular Venkova and coworkers (Venkova et al, eLife 2022: <https://doi.org/10.7554/eLife.72381>) have shown nontrivial osmotic behavior of the cytoplasm during the spreading phase, so the referee is correct that it is reasonable to expect interesting effects for nuclear volume during vs after post-mitotic cell spreading.

iv) Finally, can the authors speculate as to why the nucleus would decrease by roughly 2.5 times its volume at mitosis? This suggests that it grows “more than necessary”, and that this could be linked with cell cycle-dependent growth regimes.

In a pure dynamic osmotic coupling it is expected to follow the cell volume pace up to double. We speculate that tension coupling dynamic contributes to the extra accumulated nuclear volume. But other processes could also contribute to this unbalance.

3/ Force-biased import and nuclear envelope tension

I find the conclusions of the authors that the force-biased import plays a role in regulating nuclear volume very appealing. I have a few questions regarding the experiments and the model:

i) Since the measurement of NE tension is not a standard method (and even though Ref. 22 is in press, it is still unpublished), maybe the authors should give more details about this method in the main text.

We apologize for the lack of details relative to the NE tension FRET sensor we developed. The paper from Poli and al. is now accepted (<https://doi.org/10.1038/s41467-023-37064-0>).

In the reply to reviewers section of that paper there are also additional controls. We hope now all the information relative to that valuable tool is publicly available.

ii) Related to point 2/ above, have the authors checked how is NE tension regulated during the cell cycle? Could tension explain why nuclear volume increases linearly?

Nuclear envelope tension has already been reported to increase during the cell cycle [Introini et al., doi: <https://doi.org/10.1101/2021.11.25.469847>; Chu et al., doi: <https://doi.org/10.1073/pnas.170222611>; Lomakin et al., doi: [10.1126/science.aba2894](https://doi.org/10.1126/science.aba2894)]. Interestingly, also assuming that NE tension increases linearly with time, that is not sufficient to explain how nuclear volume grows. The relationship between NE tension and nuclear volume, as previously stated, is complex and it could be both positive and negative. Positive, since an increase in NE tension would facilitate diffusion of molecules into the nucleus, affecting the osmotic equilibrium between the nucleus and the cytoplasm. Negative, since NE membrane tension counterbalances nucleus inflation.

iii) Related to the hyperosmotic experiments: although at first I was satisfied with the result, I am now a bit lost. It is known that a hyperosmotic shock rapidly induces changes in the cytoskeleton (see for instance Thirone, 2009, PMC5047760). As such, there should be an effect on NE tension, and by the authors' model, an effect on the nucleus volume, but it does not seem to be the case. Can the authors comment on this point? Maybe the effect is only on the cortex and not translated to the NE envelop?

This is an excellent point and indeed, as the reviewer, we were puzzled by that result. Interestingly, looking carefully at the movies, we noticed that upon hyperosmotic shock cells shrink immediately but mainly by lowering their height and not by changing their spreading area. We can speculate that this kind of compression response is much faster and reversible than restructuring of adhesion and contraction of the cell base. This seems to imply that disassembly of focal adhesions might happen later as a slower response/adaptation.

iv) Per the authors' model, it is my understanding that growing cells on substrata of different rigidities should impact the NC ratio. I did not find anything on this point in the literature. Do the authors know something about this?

We agree with the reviewer, our model indeed predicts that substrate rigidity, impacting on cell contractility and NE tension, would affect NC ratio. While the situation is of course way more complex than our model can describe, this is a very interesting point that we would like to address more in a future study. Interestingly, recent work from Pundel, Blowes and Connelly (<https://doi.org/10.1002/adv.202105545>), shows that extracellular physical cues affect nuclear and nucleolar volumes. Their results may be related to our model.

v) Related to the mathematical model: I appreciated the model and its explanation/parametrization strategy. I had one question on the choice of the exponential dependence of the nuclear concentration of macromolecules with external stress: why this choice? Why not linear in the first place? Especially because it seems that it is linearized in the equation defining V_n at the bottom of p. 5.

We thank the reviewer for the positive remarks on the model. We chose the exponential dependence under the hypothesis that biased transport due to the stress felt by the nuclear pores could be an activated process. However, as the reviewer correctly points out, this choice is equivalent to a linear-response assumption, and of course the data do not allow us to formulate any claim regarding this assumption. In the revised text we provided an explanation for our assumption and explicitly stated that to our scopes the assumption of linear response would be equivalent.

Minor comments

i) Figures

- adding the legend (control vs. ivermectin) on Fig. S3D would help the reader
- which condition is ivermectin in Fig. 4C? in general, what are the conditions in this plot?
- maybe ivermectin could be its own subfigure on Fig. 4 to help the reader and avoid going back to Fig. 3
- Y-axis in Fig. S2B: unit should be $\mu\text{m}^3/\text{h}$ and not μm^3

We apologize for the lack of clarity and mistakes. We improved the plot and amended the relative legend.

ii) Although the manuscript is nicely written, I found several typos, in particular in the appendix. I may have missed some, but here is a list of those I found:

- in methods section “cell transfection”, problem unit polybrene
- in methods section “N2FXm”, GFP-nes instead of NES
- in methods section “DAAM particle measurement”, typo on “technique”
- in appendix, “left” and “right” hand side are inverted I believe
- in appendix, “nucleus surface tension and e contribution” → a
- in appendix, page 2, $c^{\text{ions_out}}$ c_{out} instead of just $c^{\text{ions_out}}$ (I believe)
- in appendix, section “estimates”, in order TO confirm, prOessure
- in appendix, section “force biased”, there is a verb lacking in first sentence
- p. 6 repetition “the the”

We apologize for these mistakes and we thank the reviewer for spotting them.

Reviewer #3 (Remarks to the Author):

This manuscript describes experiments and modeling of the regulation of nuclear size by osmotic forces and mechanical properties of the nuclear envelope in cultured mammalian cells. The study of the NC ratio in mammalian cells has been hampered by a lack of accurate volume measurements. This manuscript introduces an elegant new method for estimating nuclear/ cytoplasmic volume using a fluorescence exclusion-based approach, as well as a new FRET sensor for measuring nuclear envelope tension in cells. The work investigates NC ratio changes through the cell cycle and upon perturbations such as osmotic shock and cellular detachment from the substrate. The data support a mechano-osmotic model that implicates osmotic pressure and nuclear transport as critical components for nuclear size control.

In general, this work has potential to be an important advance in the field, but it is currently too preliminary for publication. There are many significant concerns described below. These include insufficient characterization of the new methods and a general tendency for over interpretation leading to inappropriate conclusions. They present some interesting initial observations without establishing sufficient context or additional insights. Some limited experimental data and careful rewriting will be needed to improve this manuscript.

We thank the reviewer for seeing potential in our work.

Major comments

1. The title, summary and parts of the abstract are not appropriate as key statements are not sufficiently backed by the data in this paper. For example, the Summary statement is: "Cytoskeletal forces exerted on the nuclear envelope impact on nuclear volume through modulation of force-coupled nucleo-cytoplasmic transport. However, there is little in this paper on cytoskeletal forces or nuclear transport. The closest test is an effect with ivermectin. However this result does not directly show that nuclear transport rates have changed. There is also no clear data on cytoskeletal forces on nuclear import; small effects with LatA are shown, but these may be highly pleiotropic and cannot be just assumed to be actin effects on the nuclear envelope. Their statement in the last paragraph "forces exerted by the cytoskeleton, affecting NE tension, impact on nuclear pores size (26–28)" have not been conclusively demonstrated in these cited papers and have not been proven here. While the authors certainly may hypothesize upon force-effects on nuclear transport and pores, these statements should not be the central conclusion of this paper. I think that the authors should address these concerns by judicious rewriting the wording of their conclusions, rather than being asked to perform the large number of experiments needed to support claims.

We agree with the reviewer that it is not the scope of our work to prove that forces exerted by the cytoskeleton affect nuclear transport by impacting on nuclear pores size. The phenomenon of tension-biased transport that we leverage to explain our data appears to be well supported by the literature. Indeed, the idea of tension-biased transport was introduced, supported and quantified in a seminal paper from the Roca-Cusachs lab (Elosegui-Artola et al., Cell 2017, ref. 27). The Roca-Cusachs lab recently published an elegant and systematic report on how cytoskeletal forces affect nuclear cytoplasmic transport (Andreu et al., NCB 2022, ref. 25). The idea that this process could be coupled to pore size was supported by two later independent structural studies, Zimmerli et al. (Science 2021, ref.26 ) and Schuller

et al. (Nature 2021, ref. 28), both showing that nuclear pores size depends on the cell's tensional state. However, this is not essential for us. In our study, Fig4.C, shows a clear correlation between NE tension and GFP.nls NC ratio, here used as a marker of nuclear cytoplasmic transport efficiency. As the reviewer highlights, based on our data it is reasonable to conclude that cytoplasmic forces, affecting NE tension, impact on nuclear cytoplasmic transport and nuclear volume. We completely agree that our data does not add anything on the question of nuclear pores size changes. We rephrased our conclusions accordingly in the revised manuscript.

2. There are important concerns about the N2FXm method that need to be addressed. Additional data and more detailed descriptions of the calibration method are needed so that the accuracy of the method can be evaluated (see below). The authors statement that the GFP "localizes in the entire cytoplasm except the nucleus" is not true. For example, other organelles besides the nucleus may also exclude the cytoplasmic GFP; lysozymes, mitochondria, lipid droplets etc., might represent 2-10% of cellular volume. This issue and how it may affect NC ratio measurements using N2JXm certainly needs to be addressed. Can the authors determine if these organelle volumes significantly affect the NC ratio measurements? Another concern is whether NES-GFP is truly all excluded from the nucleoplasm. Can the authors include a control using high-resolution confocal sections to show that there is no significant nuclear signal? If there is some nuclear signal, could they determine how that would affect the accuracy measurements? They should also show whether the localization of the NES-GFP marker is affected by their perturbations. For instance, mechanical or drug perturbations could cause entry of the NES-GFP in the nucleus, which could potentially affect their measurements and conclusions.

We thank the referee for these relevant remarks, which were also raised by reviewer 2. GFP-NES background in the nucleus, indeed, could affect the precision of N2FXm. Similarly, dishomogeneity in the distribution of GFP-NES in the cytoplasm, could perturb the second calibration, then impacting on nuclear volume estimates. We now explicitly discuss these possible drawbacks in the revised version of our manuscript. Additionally, following the reviewer's advice, we evaluated the distribution of the GFP-NES signal in correspondence of the nucleus (Rebuttal-Fig1.A). The mean radial intensity profile is minimum at the center and increases toward the periphery. In low magnification widefield images, the signal background simply originated by the presence of GFP-NES in the nucleus is expected to be rather constant or proportional to nucleus depth, then maximum at the center and minimum at the periphery. The profile we measured (middle panel in Rebuttal-Fig1.A) is instead compatible with a signal coming from the cytoplasm and then integrated over the cytoplasmic volume surrounding the nucleus. In the third plot in Rebuttal-Fig1.A we show the expected theoretical radial profile for a spherical nucleus (of unitary radius) embedded in a labeled cytoplasm:

$$f(r) = \int_0^r (1 - \sqrt{1-x^2}) dx = r - \frac{1}{2} * \arcsin(r) - \frac{1}{2} * r * \sqrt{1-r^2}.$$

This is compatible with the experimental profile we measured.

Importantly, we cannot exclude that a small portion of the signal we record in correspondence to the nucleus comes from inside the nucleus. Indeed we recorded a

background line on the radial profile that could have originated both by GFP-NES on top and bottom of the nucleus or by GFP-NES in the nucleus. However, when we evaluate in a confocal section (Rebuttal-Fig1.C) the proportion between cytoplasmic and nuclear GFP-NES signal, it becomes clear that the majority of the signal comes from the cytoplasm. In the mid-nucleus confocal sections, the mean intensity of GFP-NES in the cytoplasm is approximately 5 times higher than in the nucleus (Rebuttal-Fig1.C). Importantly, this ratio is not perturbed by drug treatments (Rebuttal-Fig1.C).

To allow the reviewers to judge directly if and how inhomogeneity in the distribution of GFP-NES signal in the cytoplasm could impact our measurements we plotted a representative example of a cumulative distribution of R^2 values, relative to more than 150 frames over 10 cells, for linear fits between GFP-NES cytoplasmic intensities and calibrated optical heights (Rebuttal-Fig1.B and ref to Figure 1.C). R^2 distribution is strongly biased towards 1, median = 0.93, sd = 0.14. Significant inhomogeneity in the distribution of GFP-NES signal, such as local accumulations or depauperation, would have perturbed the linear dependency between these two quantities and then the distribution of R^2 values.

All together these considerations suggest that GFP-NES background in the nucleus and as well as possible dishomogeneity in the distribution of GFP-NES in the cytoplasm and its impact on second calibration, are neglectable/minimal. This is probably due to the 'low' sensitivity and resolution of the imaging system settings as they are necessary for the N2FXm method to properly work, basically low numerical aperture and low magnification objective. Actually, it is important to notice that we challenged our method with two independent controls. First, we compared our nuclear volume measurements with 3D confocal reconstruction of nuclei, and we didn't record a significant shift in the distribution of nuclear volume estimates of the same cell population. Second, we measure in living cells the volume of artificial spherical objects (DAAMs particles). N2FXm measurements were in good agreement with theoretical volume computed by geometrical reconstruction. These two complementary validations well support the goodness of our method.

3. There are also concerns about the FRET-based nuclear envelope force sensor. This is a new sensor developed by this group, and its development is described only in this work and in a preprint. However, neither manuscript presents sufficient background characterization of this probe to demonstrate that it is acting as a force sensor at the nuclear envelope. Although the handful of data points are consistent, more quantitative and systematic controls would give more confidence that this sensor indeed measures force at the nuclear envelope.

We apologize for the lack of details relative to the NE tension FRET sensor we developed. The paper from Poli and al. is now accepted (DOI: 10.1038/s41467-023-37064-0). In the reply to reviewers section of that paper there are also additional controls. We hope now all the information relative to that valuable tool is publicly available.

4. One of the most surprising results is the linear growth of the nucleus vs. exponential growth of the cell volume. Other than pointing out the difference, this is not further investigated. More data are needed to confirm and develop this observation. One simple prediction is that NC ratios systematically fall as cells grow larger during interphase (however, this is not apparent in Figure 1F). Could this difference arise from some

systematic error in the N2FXm measurements? The reader is left without any context on how to think about this linear growth rate, or whether it carries much significance.

We agree with the reviewer: the observation that cell and nucleus volumes don't grow at the same pace is probably the most relevant result of our study. Most of our data focus on effects on time scales of minutes to less than an hour, and our explanations are fully coherent on these time scales. Cell-cycle time scales involve different processes and likely different explanations are necessary. Please see also reply to point ii of Reviewer#2. The fact that nuclear volume growth speed is constant (and independent of nuclear size) doesn't necessarily mean it is always slower than the cytoplasmic one. Indeed, while the cytoplasm doubles in volume during the cell cycle, the nucleus grows approximately 2.5 times (Fig2.E). To better clarify this point, we simulated time dependent volume growth curves for cyto and nucleus, and relative NC ratio (Rebuttal-Fig1.D). The cyto volume is assumed to grow exponentially while the nucleus linearly. The parameters used for the simulations, volume growth speed and minimal volume, are the ones obtained from the fit of experimental averages volume growth speed for RPE1 cell (Fig2.G). As the reviewer pointed out, at long time points, the NC ratio will collapse. However, for a time interval in the scale of a cell cycle and with the parameters obtained fitting the experimental data, the NC ratio is initially growing and lately starts to decay. For time points over the average span of a cell cycle, the NC ratio will certainly tend to zero. It would be for sure interesting to test our model in cells going to senescence and investigate the NC ratio dynamics in those conditions. This is however behind the scope of this work.

Moreover, we evaluated the time averaged values of cytoplasm and nucleus normalized volume growth speeds for all cell lines analyzed (Rebuttal-Fig1.D). Normalized average nucleus volume growth speeds are comparable, and typically higher (with the exception of the MCF7 cell line), than the one of the cytoplasm. This explains why the NC ratio doesn't systematically fall in the time scale of a cell cycle.

As suggested, in the new version of our manuscript we further discuss the linear versus exponential volume growth speed.

5. Another result is the apparent loss of nuclear volume after mitosis. Has it been confirmed by other methods or in other papers? Could the results be explained by the behavior of the NES marker at these cell cycle stages? If NC ratios are low after mitosis and then fall over interphase because of linear growth, then how does the nuclear volume ever catch up? How are NC ratios then maintained over multiple cell cycles? In Figure S2A, there is an example where nuclear growth suddenly speeds up after 50 frames (what is the frame rate?). Is this transition typical (as it is not seen in the averaged data)? Does it correspond to a known cell cycle transition? In general, these data are not discussed sufficiently in context to what is currently known about these processes. If the authors hope to describe thoroughly the NC ratio changes over the cell cycle, these issues should be clarified. Overall, the measurements of nuclear volume just before and after mitoses in Figure 2 represent a weak point in the manuscript and could even be deleted (or placed in another manuscript).

It is certainly true that our method can measure the volume of the nucleus only when this compartment is properly defined. In the few frames between nuclear envelope breakdown (NEB) and nucleus sealing after mitosis, centrally preceding nucleus postmitotic expansion (PME), the nucleus is not an isolated compartment anymore and can't be defined. During

this time interval, GFP-NES diffuses inside the nucleus and our method measures a residual volume that is still not accessible to it. We expressly clarify this point in the updated method session of our manuscript. For this reason, when we want to compare nuclear volume measurements before and after mitosis we compare them at NEB and PME, time points when the nucleus is still integer and when it is surely integer again, respectively. We apologize for the lack of explanation, nuclear post mitotic expansion (PME), the fast nuclear volume increase after mitosis, has been previously described, please see as example: Gerlich et al., NCB 2001; Baarlink et al., NCB 2017 and Krippner et al., EMBO Reports 2020. We now introduce PME with a relevant reference in the revised version of our manuscript.

Regarding how the nucleus can “catch up” its volume after mitosis, please see the reply to point 4.

The single-cell plot in Fig.S2A was intended to show the quality of the original data. Analyzing at the single-cell level the relationship between specific cell cycle stage or transition with nuclear and cellular volume growth is beyond the scope, and the reach, of this work. This goal can be potentially achieved with our method but will require a considerable amount of work in future studies.

Specific Comments

p.3 and Methods

More details of the calibration methods and evaluation of the accuracy of the approach are needed. First, can the authors provide an experimental version of Fig S1C (top right) to demonstrate linearity with the Texas red dextran marker they use?

We apologize for the lack of clarity, the plot in Fig1.C is indeed experimental. As previously mentioned, to better evaluate the robustness of fits along different frames and cells now in Rebuttal-Fig1.B we also show the cumulative distribution of R^2 relative to linear fits of GFP-NES intensity in function of cytoplasm optical heights measured with standard FXm. Please notice that the calibration is performed for each cell and at each time point.

Second, more detailed descriptions of how the NES-GFP intensity is calibrated with the cell volume data is needed. Can the authors show in a figure a real-life example (not just schematics)?

Please see the reply to the previous point. A real-life example was already provided in the original version of our manuscript.

For calibration, more details are needed on what cytoplasmic portions of the cell is used for comparison? How well do the red and green intensities inversely correlate; can the authors add plot showing correlation of green and red intensities per pixel? How is the noise in the intensity measurements dealt with for?

We apologize for the lack of clarity. For the GFP-NES calibration we use the intensity of all cell pixels excluding the ones in the nuclear area, as defined by the H2B signal. A real example of that is in Fig1.C. The linear dependency is robust, as can be appreciated by the cumulative distribution of R^2 for several single time frame linear fits from different cells (Rebuttal-Fig1.B).

Inhomogeneity in the cytoplasm could have a large impact on the interpretation of these measurements. Can the authors test whether inhomogeneities are significant enough to affect the calibration and subsequent measurements.? For example if there are many lysosomes in one part of the cytoplasm, is that portion of the cell not used for calibration?

We agree with the reviewer, that is an important point. Please see the response to your second major point. It is true, essentially, accumulation as well as depauperation of GFP-NES in the cytoplasm, as could be generated by vesicles excluding or concentrating the fluorescent protein, could perturb the linear dependency between the GFP-NES and the calibrated optical heights. Distribution of R^2 in Rebuttal-Fig1.B shows that is not the case.

L.7 p3 , L.6 p4 & L.27 p.5

Authors use the same nomenclature "NC ratio" for ratio of the nuclear to cell volume as well as nuclear to cytoplasmic volume. One definition should be used consistently. Also, the term "NC volume" might be reconsidered.

We apologize for this mistake and we amended it in the updated version of our manuscript.

L.12 p.4

"We also compared nuclear volume distribution of RPE1 cells measured with both N2FXm and 3D confocal reconstruction (see materials and methods) obtaining similar results"

This wording implies that the same cells were imaged with the two methods; is this true? The bar graph appears to show a sizable difference, but the authors say that it is not significant. Can they provide the mean values and the standard error? Can the authors include an example of the confocal z stacks with their markers. (This might address whether there is any detectable NES-GFP in the nucleus).

We apologize for the lack of clarity. The referee is correct, we did not measure the same cells, we refer to the same cell population. Now we clarified that sentence as: "We independently measured nuclear volume distribution of RPE1 cells with both N2FXm and 3D confocal reconstruction (see materials and methods). The difference between the distributions is not statistically significant."

In boxplot in Fig. S1.F, the boxes extend from the first to the third quartiles. The middle line represents the median. The whiskers extend up or down to $1.5 \times \text{IQR}$, where IQR is the inter-quartile range.

L.23 p.4

"DCIS.com "

Are the authors referring to the cell line "MCF10 DCIS.COM"? This should be clarified. Also, in the methods, the authors refer to DCIS.com medium; is this correct?

We apologize for the non appropriate naming. In the main text and in the method section of our revised manuscript we now specify that in text, figure and legend we used the name "DCIS.com" as an abbreviation of "MCF10 DCIS.COM".

Figure1

Can Panel D and E be plotted with the same y-axis to facilitate comparison.

Yes, in the new version of figure 1 now the two plots have the same scale.

Figure 2A

As the nucleus breaks down during mitosis, it is confusing what is being measured with this assay when there is no intact nucleus. This should also be clarified in the text. Consider removing these values after NEB on the graph, or make them a lighter orange color to denote that it is not really the nuclear volume.

This is a very important point and as suggested we clarified it in the text. We agree with the reviewer, indeed at the envelope breakdown there is no nucleus anymore and GFP-NES diffuse in the nucleus. Interestingly, with our method we measure a residual volume that is still not accessible to it. We could only speculate that this is the space occupied by the chromatin.

L.3 p5 & Figure 2C

"We found that NEB systematically precedes the onset of cellular roundup by ~10-20 min."

This statement is not backed well by the corresponding graph. The time points in the graph of 20 min apart do not allow for an accurate assessment of timing, and three of the five strains have similar percentages at -20 and 0 time points. These data should be discussed in relationship to previously published data on this time.

We agree with the reviewer and indeed in the text we specified: "However, the temporal resolution of our experiments (10 min) was too small to precisely distinguish these two events."

L10. P.5

"This implies a non-constancy of the NC volume ratio between NEB and PME (Fig. 2F)"

Can the authors present a statistical test between the paired Nuc and Cyto ratio results presented in Figure E and the NC ratio at PME and NEB in panel F.

We thank the reviewer for this suggestion and we added test results in the updated version of Fig2.

L12 p5

The results on growth rates need to be better described. Currently it is presented as the last graph in Figure 2, along the division data, and discussed in the same paragraph as the cell division, but describes behavior on an entirely different magnitude time scale. This is confusing to the reader. At the very least, the data on growth should be discussed in separate paragraphs for division in the text, and ideally presented in separate figures. As mentioned above, more context to this result should be given.

As suggested we describe this result in a separate paragraph.

L.17 p5

"Overall, these results indicate that in mammalian cells the nucleus-cytoplasm volumetric coupling could not be simply defined by a pure osmotic equilibrium, which would lead, instead, to a constant value of the NC ratio (16, 17)."

There are certainly circumstances in which NC ratios can vary even if they only use osmotic mechanism. Reference 16 shows that if the nuclear growth rate speed is proportional to the cell volume then a pure osmotic theory can explain N/C ratio maintenance. Can the authors show the Vol growth speed for nuclei as a function of Norm volume for cells? Is it flat or linear? If it is not linear than the pure osmotic model cannot explain these results.

As suggested by the reviewer we plotted nuclear volume grow speed as a function of cell volume (Rebuttal-Fig1.E). Nuclear volume grow speed doesn't correlate with cyto volume.

L.28 p5

“As expected from a purely osmotic model”

Can the authors clarify what they define as a "purely osmotic model". Can they refer to an equation? They cite this pure osmotic model with references 16, 17, but these papers use models that also take in account membrane tension.

We apologize, we correct that sentence with:“As expected from a model considering only osmotic equilibrium and membrane tension (eq.2 in SI appendix)” .

L.30 p.5

“Our model also shows that the same behavior is expected for a non-negligible constant surface tension, with a small correction on the slope, but the expected nuclear volume changes due to external forces are [...]”

Can they refer to an equation?

We now refer to eq.6 in the SI Appendix.

Figure 3 H & I : The plots look very similar, even in the SD. Please confirm that the NC data shown are correct.

We thank the reviewer and we confirm that data plotted are correct.

L.11 p.6

“was mostly unaffected along hyperosmotic shocks (Fig. 3E).”

Can the authors conduct a test for the Norm Mean Inv FRETindex at t=0 min and after the hypertonic shock t>10 minutes. It looks like there is a significant difference

We thank the reviewer for noticing it and indeed there is a statistical difference. Now we added statistical test analysis to the plots, please see updated version of figure 3.

Figure 3

Panel E,J,O. Can these be plotted with the same y axis to facilitate comparison?

Panel C,D,E. Can the authors label when the hyper osmotic shock occurs on the graph?

Figure 3 H & I.The plots look very similar, even in the SD. Please confirm that the NC data shown are correct.

Thank you for these suggestions. Following the reviewer's advice we updated Fig.3.

L.19 p.6

“nuclear and cytoplasmic volumes are strongly decoupled, with nuclear volume variations coherent with the changes of forces exerted on the NE and not with the changes of cytoplasmic volume (Fig. 4A).”

It would be important to test if NES-GFP localization is altered during these experiments.

Please find in Rebuttal-Fig1.C single plane confocal images of GFP-NES upon drug treatments. Moreover, the mean intensity ratio between cytoplasm and nucleus is not perturbed by drugs treatments.

L.21 p.7

"specific transcription factor such as YAP, key regulator of organ growth and regeneration as well as of mechanotransduction, also are affected by this mechanism."

Please cite a reference for this statement.

We apologize for this omission. We now add also here citation to ref.27 as it should be.

References

Update Lemiere reference #16

Fix Deveri reference #17

Fix Zimmerli reference #26

Thank you, we updated these references in the new version of our manuscript.

Rebuttal-Fig1. A. Left, widefield representative image of RPE1 cell expressing GFP-NES; highlighted in white the nucleus area; scale bar $20 \mu\text{m}$. Middle, average experimental radial profile of GFP-NES mean intensity in the nucleus area. Right, theoretical prediction of radial

profile. B. Cumulative distribution over several frames and different cells of R^2 relative to linear fits of pixels GFP-NES intensity in the cytoplasm in function of the corresponding pixel optical height computed with traditional FXm. C. Left, single plane confocal representative image of RPE1 cell expressing GFP-NES; highlighted in white the line used for the relative intensity scan; scale bar $20 \mu\text{m}$. Middle, line intensity profile relative to the image on the left. Right, cytoplasm to nucleus ratio of GFP-NES mean intensity for ctrl, Y-compound and latrunculin treated cells with relative representative images, scale bars $20 \mu\text{m}$. D. Left, simulations of time dependent cytoplasm and nucleus volume growth curves. Parameters used for the simulation were extrapolated from the fit of experimental grow speed of RPE1 cells (main Fig2.G). Middle, nucleus to cytoplasm volume ratio as obtained from the simulation. Right, average normalized cytoplasm and nucleus volume grow speeds. E. Conditional average of nuclear grow speed in function of cytoplasm volume.

Reviewers' comments:

Reviewer #1 (Remarks to the Author):

The authors have responded to almost all of my comments but may have misunderstood my first point where I emphasized the role of active transport. Their proposed revision discusses osmotic pressure and nuclear envelope tension and points to papers that review those. However, what I suggested was that in the introduction the authors specifically point to previous research on how the osmotic effects are actively regulated by the fact that transport through the nuclear pores is not diffusive and passive, but requires active regulation. Thus, some proteins are transported at a high rate from the nucleus to the cytoplasm but transported with lower rates from the cytoplasm to the nucleus. [Passive transport such as transport via channels results in equal rates in both directions.] For active transport, this results in steady-state, in a lower concentration of those proteins in the nucleus. The reverse situation can occur for other proteins. Thus, the osmotic pressure differences may be due to the different concentrations of the various proteins in the cytoplasm and in the nucleus, due to the difference in their active transport rates to and fro. I still feel that this should be explicated, contrasted with passive transport and discussed with references in the introduction.

Reviewer #3 (Remarks to the Author):

The authors have not addressed sufficiently the major concerns of this work in this revision. While text revisions that soften and clarify the conclusions have improved the manuscript, there are still outstanding issues about the method as well as on the experiments on the distinction between linear and exponential growth. In this revision, there are no additional analyses or data, only a reciting of rebuttal arguments that have not changed our assessments.

The concerns about the methods, its accuracy and possible GFP-NES leakage have not been addressed.

The concerns about the distinction between linear and exponential growth of nuclear and cellular growth rates, which is a major novel take home of the paper, has also not been adequately addressed. Figure 2G is the problematic figure used as the primary basis for this conclusion. We still think that apparent difference in the growth rate patterns between the nucleus and cell shown in Figure 2G are due to inappropriate scaling of the Y-axis used to plot the nuclear growth rate. As the nuclear growth rate values are many times smaller than the cellular rate, there is a problem of plotting the absolute values on the same graph. To illustrate this point for the editors, we have taken the points on the Figure

2G graph and rescaled it ourselves (See submitted figure). When nuclear growth rates are plotted with proper Y axis values, it clearly shows the rates are quite variable with different nuclear sizes: a pattern that is clearly non-linear and not so different from the cell growth rate pattern. Why one of these should be fitted to a linear function and the other to an exponential is not clear. Why these plots are binned differently also needs explanation. Thus, this data do not support the major claim that the nuclear growth rate is linear.

Note: In Rebuttal figure 1E, similar non-linear nuclear growth rate patterns are also seen as a function of cellular volume; given the scaling between nucleus and cell size, this supports the significant variability in nuclear growth rates as a function of nuclear and cell size. (Note: in Figure 2G, the definition of the X axes used needs to be clarified in the figure legend).

Force sensor. We appreciate the rewording on the limitations of the nesprin force sensor. However, there is a question of how force from nesprin and cytoskeleton that is perpendicular to the membrane relates to tension parallel to the membrane which is presumably what regulates nuclear pore transport? The authors should be cautious in not equating these different forces.

(1)

(2)

(3)

Figure from Reviewer 3.

Rescaling of Figure 2G reveals non-linear nuclear growth pattern.

- 1) This is the current version of Figure 2G in the manuscript used to claim that nuclear growth rate is "linear".
- 2) We used a vector graphic software (Affinity Designer) to resize the Y axis of the nuclear growth data (yellow) to fit the range of data points.
- 3) With this proper scaling, the pattern of nuclear growth rates (yellow) looks similar to that of the cellular growth rates (blue). This challenges the current claim that nuclear growth and cellular growth rates follow different rules.

Reviewer #1 (Remarks to the Author):

The authors have responded to almost all of my comments but may have misunderstood my first point where I emphasized the role of active transport. Their proposed revision discusses osmotic pressure and nuclear envelope tension and points to papers that review those. However, what I suggested was that in the introduction the authors specifically point to previous research on how the osmotic effects are actively regulated by the fact that transport through the nuclear pores is not diffusive and passive, but requires active regulation. Thus, some proteins are transported at a high rate from the nucleus to the cytoplasm but transported with lower rates from the cytoplasm to the nucleus. [Passive transport such as transport via channels results in equal rates in both directions.] For active transport, this results in steady-state, in a lower concentration of those proteins in the nucleus. The reverse situation can occur for other proteins. Thus, the osmotic pressure differences may be due to the different concentrations of the various proteins in the cytoplasm and in the nucleus, due to the difference in their active transport rates to and fro. I still feel that this should be explicated, contrasted with passive transport and discussed with references in the introduction.

We thank the reviewer for this valuable suggestion and we modified the introduction of our manuscript accordingly.

Reviewer #3 (Remarks to the Author):

The authors have not addressed sufficiently the major concerns of this work in this revision. While text revisions that soften and clarify the conclusions have improved the manuscript, there are still outstanding issues about the method as well as on the experiments on the distinction between linear and exponential growth. In this revision, there are no additional analyses or data, only a reciting of rebuttal arguments that have not changed our assessments.

We thank the reviewer for the time and effort dedicated to the accurate evaluation of our work. We are sorry for the negative outcome, but we have provided new analyses and comments that we believe should address the reviewer's concerns, which are partially due to insufficient clarity and depth in our previous analyses.

The concerns about the methods, its accuracy and possible GFP-NES leakage have not been addressed.

We believe we did address this concern in full, although our explanations might have been insufficiently clear. We summarize here our analyses and reasoning to address the reviewer's concerns.

The accuracy of our method has been tested with two independent experiments. First, we measured in Rpe1 cells the distribution of nuclear volumes by 3D confocal reconstruction. This distribution is not statistically different to the one obtained with N2FXm (Supp. Fig1.G). Second, we employed spherical particles whose refractive index is matched to the one of the cell (this aspect is crucial), the "DAAM" particles provided by Daan Vorselen and previously published in Vorselen et al. 2020 [doi: <https://doi.org/10.1038/s41467-019-13804-z>]. When these particles are engulfed by the cells, we measured their volume by geometrical reconstruction (essentially measuring the diameter and computing the corresponding volume) and, in parallel, by N2FXm. The two measurements are in good agreement (Supp. Fig1.H-J).

As stated in the first and second rebuttal letter, we cannot exclude leakage of GFP-NES in the nucleus and honestly it is not reasonable to expect that the nucleus is totally impermeable to the dye. However, what matters is that this is a minor effect, and if GFP-NES in the nucleus is the main source of signal in the nuclear area, the radial distribution of the signal should be maximum at the center and minimum at the periphery. With the settings used for N2FXm, wide-field microscopy with low magnification and low numerical aperture, we measured the contrary (Rebuttal Fig1.A). Thus, we believe that it is very reasonable to conclude that, although a non-zero background exists, the dominant component of the measured GFP-NES signal in the nuclear area is coming from outside the nucleus.

Finally, regarding how possible organelles in the cytoplasm could affect the second calibration, we show that the linear correlation between GFP-NES signal and cytoplasm optical height, as quantified with standard FXM, is very good (Supp. Fig1.F). Any strong inhomogeneity in the cytoplasm should have perturbed this correlation, which instead we find to be very robust.

The concerns about the distinction between linear and exponential growth of nuclear and cellular growth rates, which is a major novel take home of the paper, has also not been

adequately addressed. Figure 2G is the problematic figure used as the primary basis for this conclusion. We still think that apparent difference in the growth rate patterns between the nucleus and cell shown in Figure 2G are due to inappropriate scaling of the Y-axis used to plot the nuclear growth rate. As the nuclear growth rate values are many times smaller than the cellular rate, there is a problem of plotting the absolute values on the same graph. To illustrate this point for the editors, we have taken the points on the Figure 2G graph and rescaled it ourselves (See submitted figure). When nuclear growth rates are plotted with proper Y axis values, it clearly shows the rates are quite variable with different nuclear sizes: a pattern that is clearly non-linear and not so different from the cell growth rate pattern. Why one of these should be fitted to a linear function and the other to an exponential is not clear. Why these plots are binned differently also needs explanation. Thus, this data do not support the major claim that the nuclear growth rate is linear.

We admit that the representation provided with plots in Fig2.G and Supp. Fig2.B (we normalized the x-axis for cytoplasm and nuclear volume growth rate comparison) was not the most suited to support our conclusion that the nuclear volume is – on average - not growing as exponentially as the cytoplasm (as a side note, the binning difference is due to their basis in nuclear and cell volumes). We apologize and we regret to not have provided more promptly a better quantification. We also agree with the reviewer that there are further details beyond this main trend, which, however, we believe is robust.

To better explain this point and dispel any doubts we now computed the specific growth rate, $1/V \cdot dV/dt$ by binned averages, independently for cell and nucleus, as previously done in Fig2.b of Cadart et al. [<https://doi.org/10.7554/eLife.70816>]. This alternative quantification clearly shows that (Rebuttal Fig2):

- I. the cytoplasm is compatible with exponential growth, but also with the cell-cycle trends observed previously (Cadart et al.) for HeLa cells;
- II. the nucleus (except for one outlier bin) shows a decreasing trend, meaning that its average growth must be sub-exponential;
- III. nucleus and cytoplasm do not grow at the same pace in terms of the average growth law followed by single cells.

It is important to notice that this last point is the core of our scientific message. Indeed, regardless of the details on the typical single-cell growth curve of the nucleus versus the cytoplasm, the novel and crucial finding is that the two compartments follow dramatically different growth rules, which we believe is beyond any debate (and is also very robust across cell lines in our data). This said, we agree with the referee that there are further details and sub-trends to be investigated regarding these single-cell growth laws.

In the new version of our manuscript we substitute the plots of volume growth speed with the ones of specific growth rate. Moreover, we toned down the conclusion that the nucleus grows on average at constant speed and we just propose it as a possible interpretation of the sub-exponential growth rate. This hypothesis is also supported by the fact that the residuals of the fit of the growth speed versus a constant value are normally distributed around zero.

Note: In Rebuttal figure 1E, similar non-linear nuclear growth rate patterns are also seen as a function of cellular volume; given the scaling between nucleus and cell size, this supports the significant variability in nuclear growth rates as a function of nuclear and cell size. (Note: in Figure 2G, the definition of the X axes used needs to be clarified in the figure legend).

We agree that there are cell-cycle sub-trends in the data, such as the cell-cycle dependent difference in exponential growth rate found by Cadart et al. However, as stated previously, the plots in Rebuttal Fig1.E clearly show that:

- I. for all the cell lines analyzed nuclear volume growth speed is not monotonically increasing with cell volume (thus its main trend is not compatible with exponential growth);
- II. the single-cell mean growth laws of nuclear and cytoplasm volume differ.

We stress once again that we agree with the reviewer that the shape of these curves is not trivial and there are further details that are interesting, but their interpretation, we believe, is beyond the scope of this paper. We have stated this point more clearly in the manuscript text. For a clarification about the X axes please see the response to the previous point.

Force sensor. We appreciate the rewording on the limitations of the nesprin force sensor. However, there is a question of how force from nesprin and cytoskeleton that is perpendicular to the membrane relates to tension parallel to the membrane which is presumably what regulates nuclear pore transport? The authors should be cautious in not equating these different forces.

We thank the reviewer for raising again this point. We are very conscious that the sensor we developed is measuring forces exerted by the cytoplasm to the nuclear envelope. Those are perpendicular rather than parallel to the membrane as we already expressly state in the description of the sensor.

Rebuttal Figure 2. Cell (top) and Nuclear (bottom) Volume Specific Growth Rates for Rpe1 cells.

Reviewer #1 (Remarks to the Author):

The authors have responded to the points we raised. We strongly suggest that they make the following final modifications related to these points before the paper is accepted.

We appreciate the recognition of our work during the revision process. We would like to express our gratitude to the reviewer for her/his helpful suggestions, which have significantly contributed to the improvement of our work.

1. Page 1 of remarks "We apologize...": No apology is needed, but the science should be explicated. I appreciate that that authors distinguish a "naive" tension model from one that accounts for active transport in the Discussion. However, I feel that since the active model (which also includes the "naive" model in the limit that the activity is set to zero) was previously published that it be contrasted with the model of the authors in the Introduction. I believe that this is a more accurate representation of the field and the contribution of the present paper that is a significant advance, and that we already said should, after revision, be published in Nat. Comm.

As suggested, we extended the introduction of the revised version of our manuscript. Now we added this sentence:

"For mammalian cells, the commonly accepted model considers cytoplasm and nucleus in mechanical equilibrium where the dominant forces are due to osmotic pressure, and nuclear envelope tension counteracting the growing tendency of the nucleus (9). Interestingly, a previous study on pig chondrocytes has already observed a differential behavior of the nucleus with respect to the cytoplasm in response to an osmotic perturbation, suggesting a possible decoupling of the two compartments, and described it by a model of a spherical gel with a membrane stress term subjected to osmotic loading (10)."

2. Major comments: Point 1: Similar to the spirit of the previous point, we think the work of Finan should be discussed in the Introduction. It does not detract from the very nice results of the present paper if the previous research is openly presented.

Please see the reply to the previous point.

3. Consistent with the spirit of points 1 and 2, we suggest that the authors explicate in the main text, the point they so nicely state in the response to Major comment: point 3, "We could say that nuclear.."

We thanks the reviewer for the suggestion and we added in the discussion of our revised manuscript this paragraph:

"More widely, nuclear envelope stress could have two possible independent effects, which can act on different timescales: one on nuclear import (and osmotic equilibrium), which we focus on in our study, which makes nuclear volume increase when the envelope is more stretched compared to a reference state; and one, that has a direct mechanical effect, and balances the difference of osmotic pressure between the nucleus and cytoplasm, by which

increasing tension leads to a decrease of nuclear volume for a given osmotic balance at steady state. In our study, due to cell spreading, we can hypothesize that we are in a regime of low effective constitutive tension, and the import effect dominates. A comprehensive understanding of how the combination of these complex opposite effects determine nucleus volume growth during the cell cycle would require further experiments and a clearer picture of the underlying mechanisms.”

4. We still do not understand why a compressive force on the nucleus (assuming, as do the authors, that it is transmitted mostly to the lamina) should result in an increase in the tension. Consider the opposite case of an extensile force -- for example, that the acto-myosin cytoskeleton is pulling on the nucleus: that pulling force would tend to expand the lamina and thus increase its tension. Thus, a compressive force on the nucleus might tend to buckle the lamina proteins and thus decrease the tension. The energy would then go into the buckling of the protein network, but the tension would be decreased; this is indeed the case for many biopolymers where stretching "costs" energy, but where compression results in buckling the network. We apologize if we are confused about this, but if the referee is confused, it might be that the readers will not follow the argument and the logic should be carefully explicated or modified as noted.

We apologize for the lack of clarity in the first rebuttal letter. For compressive cytoskeletal forces acting on the nucleus we intend forces that squeezing the nucleus as a whole, like compressing it from the top, would then generate an overall increase of NE tension. We agree with the reviewer that pointy acting forces, pinching inward the NE, are expected to buckle it and generate a local decrease of tension. It is difficult to figure out if these kinds of local forces are sufficient to then generate global effects. For the scope of our work, it is important to highlight and reiterate that direct forces, both positive and negative, are unlikely to deform the nucleus, the size of which is instead mainly set by mechano-regulated osmosis.

5. Minor points: point 4: We read what the authors added to the supplementary information and have a further comment. The fact that the fluctuations decrease may not be due to an decrease in the tension due to activity, but rather, to an increase in the "fluctuating forces" that act on the system due to activity, over and above the thermal fluctuating forces. In other words, the tension might remain the same, but since the fluctuations depend on the ratio of the tension and the magnitude of the fluctuating forces, the fluctuations might increase due to an effectively higher temperature, and not a lowered tension due to activity. The authors can see, for example, <https://journals.aps.org/prl/abstract/10.1103/PhysRevLett.106.238103>

The authors should consider and explicitly note this possibility.

This is exactly what we meant (see e.g. the introduction of Introini et al. 2013, where this point is explained in detail). We apologize for the lack of clarity, we modified the relative paragraph of the Appendix as follows:

“The apparent tension of the nucleus, estimating σ_N or σ_0 is found to be around 10^{-6} N/m in nuclear shape fluctuations experiments (Chu et al., 2017; Introini et al., 2023. However, this value is an underestimation of the actual tension, since it is affected by excess flickering due to transient deformations of active origin, since fluctuating forces due to activity decrease the effective tension with respect to the actual mechanical tension (see Introini et al., 2023 for a

detailed explanation). We can assume that this value is a lower bound for the real mechanical tension σ_0 ."

Reviewer #2 (Remarks to the Author):

I am mostly satisfied with the way the authors addressed my concerns. I believe that the novel methodology and its subsequent application in this study grant publication in Nature Communications. This methodology will certainly be applied by other research groups and stimulate investigations on nuclear volume regulation.

We appreciate the recognition of our work during the revision process. We would like to express our gratitude to the reviewer for her/his helpful suggestions, which have significantly contributed to the improvement of our work. Additionally, we are pleased by the reviewer's enthusiasm for our project and her/his recognition of the potential impact it could have in the community.

Reviewer #3 (Remarks to the Author):

In general, there are several major concerns of this work that have not been suitably addressed in this revision. Most of the concerns were discussed but not directly addressed. In fact, the additional data and information provided in this round further reveal the weaknesses in the data. Thus, there are still significant reservations about many aspects of this manuscript, including the validity of the N2FXm assay and the distinction of linear vs exponential growth rate. This work unfortunately is not ready for publication in any journal.

We would like to express our gratitude to the reviewer for her/his helpful suggestions, which have significantly contributed to the improvement of our work.

We are sorry but we disagree with the reviewer. The new quantifications and data provided in the first rebuttal letter, are not affecting the conclusions of our work and moreover they confirmed the observed decoupling between cellular and nuclear volume growth dynamics.

Major concerns

1. The title, summary and parts of the abstract are not appropriate as key statements are not sufficiently backed by the data in this paper.

Following editor and reviewer suggestion we propose to change the working title of our manuscript in: "Differential osmo-mechanical regulation of nuclear and cytoplasmic volume in mammalian cells".

We also toned down the abstract.

-The authors revised the text slightly to remove the implications that this work shows direct effects on nuclear pore size. However, the current title is still inappropriate as places emphasis on elements that have not been sufficiently tested in this work. As remarked in the previous round, the involvement of nuclear import (the conclusion of the title) is supported only by an effect of ivermectin, an inhibitor of α/β importin- based nuclear transport. Broader effects of actin inhibition are not considered. Thus, there is insufficient experimental data to make this the take-home message. To do so would require new experiments that directly document a change in nuclear import rate that is dependent on nesprin and its ability to bind to actin at the nuclear envelope.

As stated in the original rebuttal letter, it is not the scope of our work to demonstrate the existence of a tension-biased nuclear transport. This mechanism has been conceived and quantitatively described in two seminal papers from Roca-Cusachs lab (Elosegui-Artola et al., Cell 2017 and Andreu et al., NCB 2022). Moreover, two structural papers from different labs also suggest that cyto-skeletal tension could stretch nuclear pores (Zimmerli et al., Science 2021 and Schuller et al., Nature 2021). Although of course it is possible that other processes are in place, this evidence is well in line with Roca-Cusachs model. In our work we rely on this well established line of literature. In Figure 4C, we also show that GFP-NLS nuclear to cytoplasmic ratio, here used as a marker of nuclear "permeability", changes in function of nuclear envelope tension as measured by our sensor. Since the reviewer

suggested to not state a direct link from NE tension, dilatation of nuclear pores and nuclear permeability, we avoided that in the revised version of our manuscript. We still mention it as a possible speculative mechanism. However, the fact that NE tension impacts on nuclear volume, besides being related to past literature, is clearly supported by our data and is an essential element of the model we proposed. We also showed that, to our knowledge, forces required to directly extend the nucleus are out of scale for the cell (see reply to reviewer 1 minor point 3).

2. There are important concerns about the N2FXm method that need to be addressed. Additional data and more detailed descriptions of the calibration method are needed so that the accuracy of the method can be evaluated (see below).

-The authors have not provided sufficient additional data to address the multiple concerns of this method. There is still little data regarding its accuracy.

-Issue of organelle volume in the cytoplasm was not addressed.

As we show in the original rebuttal letter (see reply to reviewer 2 point 1-i and reviewer 3 major 2), the linear correlation between GFP-NES signal and cytoplasm optical height, as quantified with standard FXM, is very good (see rebuttal Figure 1 B). Any inhomogeneity in the distribution of GFP-NES signal in the cytoplasm, as could be generated by cytoplasmic organelles, would have perturbed this distribution.

-Is GFP-NES really excluded from the nucleus? Although the data in rebuttal Figure 1 is used for the authors to claim that this is not an issue, the data in rebuttal Figure 1 AC clearly shows measurable non-zero intensity in the nucleus in the confocal imaging in the image and the graph. The modeling in rebuttal Figure 1A does not rule out GFP-NES in the nucleus, and the graph in Figure 1A shows a non-zero intensity at the base of the graph. As the authors say, the leakage of GFP-NES into the nucleus would lead to inaccurate measurement of nuclear volume. For instance, it might explain the unexplained discrepancy between the measurements using confocal 3D reconstructions and the N2FXm.

In rebuttal Figure 1 A we show the radial distribution of the GFP-NES signal in the nucleus in wide-field images (the same used for N2FXm). It is clear that the signal is minimum at the center and increases toward the nuclear periphery. That is not compatible with a background of nuclear GFP-NES, that would be rather constant or maximum at the center and minimum at the periphery. On the contrary, this result is perfectly in line with what would be expected from a signal coming from the cytoplasm and then integrated over the cytoplasmic volume surrounding the nucleus. Clearly, the signal doesn't start from "zero" (as is not zero in confocal scan). This non zero line, could correspond to the portion of cell volume on top and below the nucleus center, or, as we clearly stated, could also be originated by nuclear GFP-NES. However, the impact of this background is minimal and does not affect our measurements. Indeed, as summarized in the rebuttal letter, we challenged our method with two independent controls. First, we compared population measurements of nuclear volumes performed with N2FXm and standard 3D confocal reconstructions, and we didn't record a significant shift in the distributions. Second, we measured in living cells the volume of artificial spherical objects (DAAMs particles). N2FXm measurements were in good agreement with theoretical volume computed by geometrical reconstruction.

-Do GFP-NES levels in the nuclear region change during drug or mechanical perturbations? The new data in rebuttal Figure 1C show that the distribution of intensity values certainly change (at least the variability increases) suggesting that at least for a good portion of cells that this might be a real issue. Changes in GFP-NES in the nucleus would further raise serious questions into the validity of the measurements.

We apologize for lack of clarity, we did not specify that we statistically tested the difference, and it is not significant (t-test vs Ctrl, p-values: Y-comp=0.140; Lat=0.0618).

3. There are also concerns about the FRET-based nuclear envelope force sensor.

-The authors should clarify the description of the force sensor. This description could be similar to what they explained to reviewer 1: "The FRET tension sensor we developed is designed on Nesprin1, one of the components of the LINC complex. It spans from the inner nuclear envelope, where it binds SUN domain containing proteins, to the cytoplasm, where it binds to actin. Therefore, it is sensitive only to forces exerted by the actin cytoskeleton to the NE." Importantly, it should be clarified that this FRET probe most likely does not sense membrane tension (parallel to the membrane) but rather forces by actin directly attached to the Nesprin probe (perpendicular to the membrane).

We thank the reviewer for this suggestion and we improved the description of the sensor in the revised version of our manuscript accordingly. We now also clearly state its possible limitations. As suggested we added in the main text: "This tension sensor spans from the inner NE, where it binds SUN domain containing proteins, to the cytoplasm, where it binds to actin. Therefore, it is sensitive only to forces exerted by the cytoskeleton to the NE and does not sense tension parallel to the membrane (23)."

4. One of the most surprising results is the linear growth of the nucleus vs. exponential growth of the cell volume. Other than pointing out the difference, this is not further investigated. More data are needed to confirm and develop this observation.

-New data on this point only further questions of the validity of this claim of linear growth. The graphs used to support linear growth of the nucleus (Figure 2G and rebuttal Figure 1E) are inconsistent with this conclusion. Rebuttal Figure 1E shows that the points for each cell type simply do not obviously fit a flat line (linear) or a line with positive slope (exponential). In Figure 2G, this deviation from a flat line is hidden by inappropriate choice of Y-axis values but is apparent by close examination (Note that this graph shows the same points as Rebuttal Figure 1E). Hence, the data do not support their conclusion that nuclear growth is linear, which is considered a major take-home of this work.

Rebuttal Figure 1E was produced to reply to a reviewer 3 minor point. Please, note that this plot does not show at all the same points of figure 2G. Here, conditional average is not performed over nuclear volume, but, as requested by the reviewer, over cell volume. Hence, the two plots are different, and they address two different questions. Nuclear Volume Growth Speed is indeed here averaged over bins of cell volume and not over bins of nuclear volume. Nothing is hidden by inappropriate scaling. Interestingly, the shape of the curve in Rebuttal Figure 1E is not trivial and we thank the reviewer for pointing our attention to this point. While it deserves more investigation, we can surely conclude that this curve is clearly more similar to a flat line than to an asymptotically growing one. These results are not affecting the

conclusions of Figure 2G. Moreover, they confirm a robust decoupling of nuclear and cell growth dynamics in single cells, a main point of our work. Regarding the flat line fits in Figure 2G and Supplementary Figure 2B, distributions of fit residuals are normal and their means are not statistically different from zero.

Other issues

Figure 1D. The Y-axis was changed in response to reviewers. However, the axis change was very crudely done, which in effect visually changes the values of the graph.

We apologize, our mistake. When we adjusted the size of text in the plot axis assembling the figure we shifted it by a few pixels. We thank the reviewer for spotting this mistake and we corrected it in the revised version of our manuscript.

The information on error bars, number of replicates, N must be provided for every experiment shown.

All this info is now in figure legends or in the method session.

Many details in the methods seem to be lacking. For instance, what were the concentration and time of treatment of ivermectin? How were the osmotic shocks performed.

This information was in the Supplementary Materials. In the revised version of our manuscript we renamed the relative section as: "Drug, adhesion and osmotic perturbations experiments". As stated, cells were treated with ivermectin (Sigma Aldrich) at 10 μ M for 24 h. For hyper-osmotic shock experiments, D-sucrose (Carlo Erba reagents) was resuspended in pure water at 500 mM and then added to cells.

Reviewer #1 (Remarks to the Author):

This paper experimentally explores the effects of various biochemical and mechanobiological perturbations of single cells on the ratio of their nuclear to cytoplasmic volumes. It is shown that changes in cytoskeletal forces exerted on the nucleus (via latrunculin or ROCK inhibition) changes the NC ratio in a manner that is not simply due to the direct effect of those forces on the osmotic pressure balance. In addition, it is shown that while the NC ratio in many cell types (under hyperosmotic conditions) is constant during most of interphase, this ratio varies considerably during cell division. Interestingly, it is found that right after cell division, the cytoplasmic volume increases exponentially, while the nuclear volume increases only linearly with time. The results will stimulate further experiments and theoretical models and should be published in Nature Communications, pending the following changes.

We are grateful for the positive remarks, and for the positive recommendation.

We believe that the authors overstate the impact of their results with respect to the concepts presented in Refs. 7, 16, and 17. In particular, the present manuscript uses the term “falsify” (once) “or “falsified” (twice) to relate their findings to the presumably naïve “osmotic” models in these references. This, besides being polemical, is not correct and does not reflect the subtlety of the ideas previously presented. While the NC volumetric ratio is indeed determined by osmotic pressure differences, the proteins involved are those whose active transport through the nuclear pores, results in differing concentrations in the cytoplasm and nucleus. The import rate and export rates can be different for systems with active transport, resulting in concentrations of proteins that are directly related to these rates. When various perturbations are applied to the cell, including changes in the cytoskeletal forces that act on the nucleus, those transport rates are modified, with – in general – different changes for the import and export rates. Thus while it is true that the naïve osmotic effect of changes in cytoskeletal force are not significant in the experiments described, there is an important indirect effect of those forces on the transport rates that determine the protein concentrations in the nucleus and cytoplasm, and hence the osmotic pressure difference and NC ratio. This was indeed a major point of these references which seems to have been missed in the present manuscript.

Nevertheless, the paper is interesting and presents several novel results about how the NC ratio changes during the cell cycle and under biochemical or mechanobiological changes, which justify publication in Nature Communications. However, before we can formally recommend publication, the interpretation of these changes with respect to such perturbations should be rewritten to accurately incorporate the previously discussed changes in osmotic pressures due to active transport and to qualitatively relate these transport changes to the perturbations applied in the present manuscript. In addition, we request the authors to address the following major and minor comments.

We apologize to the reviewer for our miscommunication. Our intent was not to go against previous models (which of course had different scopes and were not informed by our data), and we meant our statement referred exclusively to the ingredients of our own model, which is a naive version of the osmotic model, taking into account just few essential ingredients

and making some simplifications in order to keep a minimal number of parameters. We have rephrased the statements in order to avoid any misunderstanding, and also added a statement on the results of the cited refs. Specifically we added this sentence in the discussion section: “When perturbations are applied to the cell, including changes in the cytoskeletal forces that act on the nucleus, transport rates can be modified, with – in general – different changes for the import and export rates. Thus while it is true that the naïve osmotic effect of changes in cytoskeletal force are not significant in our experiments, there is an important indirect effect of those forces on the transport rates that determine the protein concentrations in the nucleus and cytoplasm, and hence the osmotic pressure difference and NC ratio. This was a major point of previous studies [REFS 7-16-17].”

Major comments:

1. At the bottom of page 5, the authors write “Our model also shows that the same behavior is expected for a non-negligible constant surface tension...”. Compared to what scale is the surface tension non negligible? The work by Finan et al. (Annals of Biomedical Engineering, 2009) presents the notion of NE that is stretched compared to one that is relaxed. The authors should relate the findings of their model with respect to the work of Finan et al.

We thank the reviewer for this comment and we refined the inaccurate statement in the current version of the manuscript. We have explored constitutive surface tensions in the range $1e^{-6}$ to $1e^{-2}$ N/m. The value around $2e^{-2}$ N/m found by Finan and coworkers by a fit of their model can be compared with the upper bound of the constitutive surface tension that we explored. Hence, our results should apply (see e.g. fig SM3 in the SI appendix). It is important to state that their richer model (based on the swelling of porous gels) contains other parameters, hence although the basic underlying physical picture of our simpler model is similar, it is possible that the values of the parameters do not map exactly between the two frameworks. Finally, we note that while we explore for simplicity our tension-biased transport model in a regime of negligible tension, this choice has no impact on our main statement, as the choice of a finite/large constitutive tension would make it equally difficult for osmotic and mechanical extensile forces to swell the nucleus.

2. The authors conducted three types of experiments to modulate the forces that are exerted on the cell and are transduced into the nucleus: hyperosmotic shock, cell detachment, and cell spreading. These types of experiments were thoroughly investigated in previous works such as Finan et al. (mentioned above), Guo et al. (ref. 19 in the manuscript), and others. However, discussion that relates the results of the manuscript to these previous works is lacking. The authors should present their experimental results in a wider context and relate them to these previous works.

The work of Finan and coworkers has the merit of showing that the volume increase of the nucleus upon hypo-osmotic shocks, saturates and does not follow a linear increase as the cell volume does (Ponder plot). The simplest interpretation of that is that part of the pressure is compensated by nuclear envelope tension, leading to less volume increase than expected when tension is negligible.

We could say that nuclear envelope stress can have two effects, which can act on different timescales: one on nuclear import, which we focus on in our study, which makes nuclear volume increase when the envelope is more stretched compared to a reference state, and one that has a direct mechanical effect, and balances the difference of osmotic pressure between the nucleus and cytoplasm, by which increasing tension leads to a decrease of nuclear volume for a given osmotic balance at steady state. In our study, due to cell spreading, we can hypothesize that we are in a regime of low effective constitutive tension, and the import effect dominates.

3. At the bottom of page 37, the authors relate the tension to external force by the equation $F_{ext} = 8\pi RN \sigma_{ext}$. The authors should explain this relation as its derivation is unclear. Perhaps a cartoon showing the force, tension, and the relation between them can make the derivation clearer.

We apologize for the missing elements in our argument. We have clarified the assumptions, and we have specified in the revised text that we intend this relation as a simple estimate relating a scenario of tension of external origin to the total magnitude of the radial forces that are needed to generate the equivalent mechanical pressure difference.

4. The authors model the effect of force on nuclear import using an exponential relation. The author should discuss the underlying biophysical reasoning of this dependence or state it is just a fitting function.

We agree that this assumption needs an explanation - which was pointed out also by reviewer 2. We have made it clear in the text that we have assumed an activated process. This assumption does not really affect the quantitative aspects, since other functional forms would be equivalent in the relevant range of values.

5. In their theoretical model, the authors treat the external force as a scalar although by definition force is not a scalar. Furthermore, the authors assume that the nucleus is spherically symmetric, which is not true in general, when deformed by non-isotropic external forces (that may have shear components).

We have clarified these aspects. We chose the spherically symmetric case to simplify the model from parameters that could not be fixed reliably based on our data. Given the assumption, the force field only has one relevant degree of freedom, hence the sloppy notation (now clarified). Additionally, we have clarified that we mean the model in this study more as a conceptual guide to understand the experiments than as a full-blown theoretical description. The purpose is to isolate and select scenarios, hence we systematically reverted to the simplest assumptions.

6. The authors state that they assume the forces that are exerted on the nucleus are compressive. In that case and when the forces are spherically symmetric, the contribution of the external forces to the tension should be negative. Is this the case? This should be clearly explicated since this affects the way the exponential relation between the surface tension and nucleocytoplasmic ratio is interpreted.

We have clarified this part in the revised SI text. In our notation the contribution of compressive external forces to the tension is positive (as they both contribute to forces pointing inwards). We analyzed both cases of positive and negative contribution of external forces to the tension, but given the previous experimental literature [e.g. <https://doi.org/10.1073%2Fpnas.0908686106>] we believe that the first scenario could be more likely.

Minor comments:

1. Can there be forces that are exerted on the nucleus which will not be detected by stretching of the LINC complex (e.g. squashing of the nucleus)?

This observation is correct. The FRET tension sensor we developed is designed on Nesprin1, one of the components of the LINC complex. It spans from the inner nuclear envelope, where it binds SUN domain containing proteins, to the cytoplasm, where it binds to actin. Therefore, it is sensitive only to forces exerted by the cytoskeleton to the NE. If external forces applied to the cell indirectly also affect the link between actin and SUN proteins at the NE, then the sensor can detect it. More detail about the FRET sensor can be found in Poli et al 2023 [<https://doi.org/10.1038/s41467-023-37064-0>]. Please consider also our response to point 3 below.

2. Some works (e.g. Jahed et al., Journal of Cell Science 2016) suggest that the cytoskeleton and LINC complex are indirectly involved in nucleocytoplasmic transport by modulating the Ran protein gradients. Is it possible that the nucleocytoplasmic transport in the latrunculin-treated or the ROCK inhibited cells is affected via this pathway rather than mechanical “expansion” of the NPCs? Can the author comment on this possibility.

We do not exclude that there may be additional/other mechanisms at play, as suggested by the reviewer. However, we have observed a strong correlation between nuclear envelope tension, as measured by our FRET sensor, and the mean intensity ratio of GFP-NLS between the nucleus and cytoplasm (used as a marker of protein nuclear incorporation efficiency) (Fig. 4C). This dependency is essential for our model, independently of whether this is due to nuclear pores expansion or to other possible mechanisms.

3. At the bottom of page 6 the authors write that the cytoskeletal forces are compressive. However, these can also be expansive if the nucleus is laterally stretched. Can the authors provide a reference to a work that distinguish between these two possible modes of nuclear deformation?

The reviewer's observation is correct. It could be, in certain circumstances, that positive forces are exerted to the nucleus. With our model, indeed, we analyzed the effect on nuclear volume of positive as well as negative external forces (always with the assumption of a spherically symmetric nucleus). In both cases variations in nuclear volume induced by direct forces are negligible with respect to the ones arising from changes in osmolarity (less than 10% for σ_0 in the range 10^{-6} to 10^{-2} N/m).

We now modify that sentence in: “Based on the model, we reasoned that this was very unlikely to be a direct effect, as, independently of the direction, compressive or stretching, the force required to change nuclear size would be abnormally large (see SI appendix) [24, Kalukula et al, Nature Reviews Molecular Cell Biology, (2022), 583-602, 23(9), DOI: <https://doi.org/10.1038/s41580-022-00480-z>].”

4. The authors mention that the estimate of the NE tension (10^{-6} N/m) is a lower bound due to fluctuations of active origin of the NE shape. Are there evidence for such active fluctuations? While this does not change the conclusion of the authors, it is worthwhile to give a reference for such active processes that facilitate NE shape fluctuations.

We agree and we have added this statement: “The apparent tension of the nucleus, estimating σ_N or σ_0 is found to be around 10^{-6} N/m in nuclear shape fluctuations experiments (Chu et al., 2017; Introini et al., 2021).”

Reviewer #2 (Remarks to the Author):

Pennacchio et al extended the powerful fluorescence exclusion microscopy, FXm, to simultaneously measure cell volume and nuclear volume – a new method they coined “N2FXm”. This novel method is sound and allowed the authors to study the nuclear-to-cell volumetric (NC) ratio. First, they found that the NC ratio was not constant during the cell cycle, being particularly impacted by mitosis. Second, they showed that the cell and nucleus did not follow the same growth laws. Third, they discovered that the NC ratio could not be explained solely by an osmotic balance and that cytoskeletal forces exerted onto the nuclear envelope could bias nuclear import and nuclear osmotic pressure, thus also participating in setting the NC ratio.

The manuscript is clearly written, and the data are of high quality and interest, shading light on novel regulatory mechanisms of nuclear volume. I would greatly recommend publication should the following questions be addressed.

We thank the reviewer for the overall positive judgment of our work and for the enthusiastic recommendation.

1/ The N2FXm method

I found the N2FXm method extremely powerful and relevant. Although most of my concerns are addressed in the methods or in the paper, I have one additional question and one recommendation for the authors:

i) Have the authors checked that there is no GFP-NES in the nucleus, and if the signal is homogeneous in the cytosol? Depending on the cell state/strength of the NES, there may still be some GFP in the nucleus, which would affect the measurement – especially if this bias is cell cycle-dependent, through for instance NE tension impacting nuclear export. Even if non-significant, there seems to be a small difference in the nuclear volume measurement between the 3D reconstructed and the N2FXm in Fig. S1F. Moreover, this could also be important in the perturbation experiments, impacting the actual measurement.

We thank the referee for these relevant remarks, which were also raised by reviewer 3. GFP-NES background in the nucleus, indeed, could affect the precision of N2FXm. Similarly, dishomogeneity in the distribution of GFP-NES in the cytoplasm, could perturb the second calibration, then impacting on nuclear volume estimates. We now explicitly discuss these possible drawbacks in the revised version of our manuscript. Additionally, following the reviewer's advice, we evaluated the distribution of the GFP-NES signal in correspondence of the nucleus (Rebuttal-Fig1.A). The mean radial intensity profile is minimum at the center and increases toward the periphery. In low magnification widefield images, the signal background simply originated by the presence of GFP-NES in the nucleus is expected to be rather constant or proportional to nucleus depth, then maximum at the center and minimum at the periphery. The profile we measured (middle panel in Rebuttal-Fig1.A) is instead compatible with a signal coming from the cytoplasm and then integrated over the cytoplasmic volume surrounding the nucleus. In the third plot in Rebuttal-Fig1.A we show the expected theoretical radial profile for a spherical nucleus (of unitary radius) embedded in a labeled cytoplasm:

$$f(r) = \int_0^r (1 - \sqrt{1-x^2}) dx = r - \frac{1}{2} * \arcsin(r) - \frac{1}{2} * r * \sqrt{1-r^2}.$$

This is compatible with the experimental profile we measured.

Importantly, we cannot exclude that a small portion of the signal we record in correspondence to the nucleus comes from inside the nucleus. Indeed we recorded a background line on the radial profile that could have originated both by GFP-NES on top and bottom of the nucleus or by GFP-NES in the nucleus. However, when we evaluate in a confocal section (Rebuttal-Fig1.C) the proportion between cytoplasmic and nuclear GFP-NES signal, it becomes clear that the majority of the signal comes from the cytoplasm. In the mid-nucleus confocal sections, the mean intensity of GFP-NES in the cytoplasm is approximately 5 times higher than in the nucleus (Rebuttal-Fig1.C). Importantly, this ratio is not perturbed by drug treatments (Rebuttal-Fig1.C).

To allow the reviewers to judge directly if and how inhomogeneity in the distribution of GFP-NES signal in the cytoplasm could impact our measurements we plotted a representative example of a cumulative distribution of R^2 values, relative to more than 150 frames over 10 cells, for linear fits between GFP-NES cytoplasmic intensities and calibrated optical heights (Rebuttal-Fig1.B and ref to Figure 1.C). R^2 distribution is strongly biased towards 1, median = 0.93, sd = 0.14. Significant inhomogeneity in the distribution of GFP-NES signal, such as local accumulations or depauperation, would have perturbed the linear dependency between these two quantities and then the distribution of R^2 values.

All together these considerations suggest that GFP-NES background in the nucleus and as well as possible dishomogeneity in the distribution of GFP-NES in the cytoplasm and its impact on second calibration, are neglectable/minimal. This is probably due to the 'low' sensitivity and resolution of the imaging system settings as they are necessary for the N2FXm method to properly work, basically low numerical aperture and low magnification objective. Actually, it is important to notice that we challenged our method with two independent controls. First, we compared our nuclear volume measurements with 3D confocal reconstruction of nuclei, and we didn't record a significant shift in the distribution of nuclear volume estimates of the same cell population. Second, we measure in living cells the volume of artificial spherical objects (DAAMs particles). N2FXm measurements were in good agreement with theoretical volume computed by geometrical reconstruction. These two complementary validations well support the goodness of our method.

ii) I believe one important point is that the GFP calibration is done at every time point and for each cell, thus limiting temporal fluctuations or cell-cell heterogeneity in GFP concentration. As such, it may be better to directly state it in the main text, and not only in the methods.

We thank the reviewer for this comment and as suggested we now specified it in the main text.

2/ Nuclear growth and NC ratio

I have a few questions regarding nuclear volume growth and subsequent NC ratio values, as well as their interpretations based on the model proposed by the authors:

i) The NC ratio seems rather constant in each cell line from -500 min to 0 min. First, can the authors provide a y-axis zoom before and after mitosis, to have a better visualization (Figs. S1L & 1F)? Second, it seems constant even though nuclear volume does not increase exponentially. Can the authors comment on this point?

To enhance the difference in NC ratios we showed it at its maximum, between PME (post mitotic expansion, nucleus life start) and NEB (nuclear envelope breakdown, nucleus life end), please see plot in Fig2.F. NC ratios are not constant and typically increase by at least 20% along the nucleus lifetime. We added this quantification in the main text of the revised version of our manuscript. As suggested, we also added in Fig2.F a graphical representation of relative paired t-tests results.

ii) Can the authors speculate (and potentially expand the discussion on this point) as to why the nucleus would grow linearly in time? Per the authors' model, if it does not increase exponentially, this could for instance mean either that import/export is cell cycle-dependent such that nuclear osmotic pressure is not constant over the cell cycle. This could for instance be the case if NE tension changes during the cell cycle (see the question on tension in point 3/ below).

The average linear growth on cell-cycle time scale is an intriguing fact that we report experimentally, but whose explanation goes beyond our scopes. Most of our data focus on effects on time scales of minutes to less than an hour, and our explanations are fully coherent on these time scales. Cell-cycle time scales involve different processes and likely different explanations are necessary.

This said, we can reason on how the data relate to the prediction of different simple models.

Let us suppose first a scenario of perfect osmotic force balance. If the tension were always negligible, the volume of the nucleus would grow exponentially as the total volume. Our analyses suggest that the tension would have to be too large to have a relevant impact on nuclear size and explain the average linear growth.

If we consider our model coupling tension with transport, we can ask whether extrapolating it to cell-cycle time scales would support linear growth. In this model the observed tension with cell cycle would increase nuclear import, causing the nucleus to swell more than predicted by osmotic balance. Initially, it would grow faster than the cytoplasm, as it does in our data (please see also the reply to point 4 of Reviewer#3). However, without more specific information our model cannot be easily used to suggest that this growth would be linear, which will require a specific and gradual release of external tension.

Recently, Rollin and coworkers have proposed a mechanism that could partly explain this trend (<https://www.biorxiv.org/content/10.1101/2022.08.01.502021v1.full>). In this preprint the authors claim a role of counterion release by chromatin folding. According to their model, the NC ratio (formula 15) can be intermediate between an expression that is the ratio between nuclear proteins and cytoplasmic proteins (NC1, the prediction of pure osmotic force balance) and one (NC2, larger) that depends on DNA charge, which is constant during

G1 phase, while the number of proteins in the nucleus grows with time, so NC2 decreases with time. Hence, this prediction, valid only in G1, could generate the right qualitative trend.

More widely, nuclear envelope tension might have at least two effects (acting on different timescales): one on nuclear import, which we focus on here, which would make the volume increase when the envelope feels more tension, while one direct mechanical effect would lead to a decrease of nuclear volume for a given osmotic balance at steady state. For high enough nuclear envelope tension, the volume loss effect might dominate, while for low tension (as in our study, due to cell spreading), the import effect might dominate.

Altogether, these synergies of complex effects make it hard to predict the volume growth of the nucleus during the cell cycle: a complete model would be needed, plus many more experiments.

iii) Related to the previous point, it does seem that, after the spreading phase (phase III), there is no or little growth of the nuclear volume, followed by more rapid growth. This seems true for the cell lines in Fig. S1L. Has this been considered in the calculation of volumetric growth? Could it be that the binning hides different growth regimes of the nucleus during the cell cycle?

We agree with the reviewer that at faster time scales, there may be several intriguing cell-cycle dependent phenomena that we missed in this study. We hope to perform higher-resolution time-resolved studies in the near future to address this and other questions. In particular Venkova and coworkers (Venkova et al, eLife 2022: <https://doi.org/10.7554/eLife.72381>) have shown nontrivial osmotic behavior of the cytoplasm during the spreading phase, so the referee is correct that it is reasonable to expect interesting effects for nuclear volume during vs after post-mitotic cell spreading.

iv) Finally, can the authors speculate as to why the nucleus would decrease by roughly 2.5 times its volume at mitosis? This suggests that it grows “more than necessary”, and that this could be linked with cell cycle-dependent growth regimes.

In a pure dynamic osmotic coupling it is expected to follow the cell volume pace up to double. We speculate that tension coupling dynamic contributes to the extra accumulated nuclear volume. But other processes could also contribute to this unbalance.

3/ Force-biased import and nuclear envelope tension

I find the conclusions of the authors that the force-biased import plays a role in regulating nuclear volume very appealing. I have a few questions regarding the experiments and the model:

i) Since the measurement of NE tension is not a standard method (and even though Ref. 22 is in press, it is still unpublished), maybe the authors should give more details about this method in the main text.

We apologize for the lack of details relative to the NE tension FRET sensor we developed. The paper from Poli and al. is now accepted (<https://doi.org/10.1038/s41467-023-37064-0>).

In the reply to reviewers section of that paper there are also additional controls. We hope now all the information relative to that valuable tool is publicly available.

ii) Related to point 2/ above, have the authors checked how is NE tension regulated during the cell cycle? Could tension explain why nuclear volume increases linearly?

Nuclear envelope tension has already been reported to increase during the cell cycle [Introini et al., doi: <https://doi.org/10.1101/2021.11.25.469847>; Chu et al., doi: <https://doi.org/10.1073/pnas.170222611>; Lomakin et al., doi: [10.1126/science.aba2894](https://doi.org/10.1126/science.aba2894)]. Interestingly, also assuming that NE tension increases linearly with time, that is not sufficient to explain how nuclear volume grows. The relationship between NE tension and nuclear volume, as previously stated, is complex and it could be both positive and negative. Positive, since an increase in NE tension would facilitate diffusion of molecules into the nucleus, affecting the osmotic equilibrium between the nucleus and the cytoplasm. Negative, since NE membrane tension counterbalances nucleus inflation.

iii) Related to the hyperosmotic experiments: although at first I was satisfied with the result, I am now a bit lost. It is known that a hyperosmotic shock rapidly induces changes in the cytoskeleton (see for instance Thirone, 2009, PMC5047760). As such, there should be an effect on NE tension, and by the authors' model, an effect on the nucleus volume, but it does not seem to be the case. Can the authors comment on this point? Maybe the effect is only on the cortex and not translated to the NE envelop?

This is an excellent point and indeed, as the reviewer, we were puzzled by that result. Interestingly, looking carefully at the movies, we noticed that upon hyperosmotic shock cells shrink immediately but mainly by lowering their height and not by changing their spreading area. We can speculate that this kind of compression response is much faster and reversible than restructuring of adhesion and contraction of the cell base. This seems to imply that disassembly of focal adhesions might happen later as a slower response/adaptation.

iv) Per the authors' model, it is my understanding that growing cells on substrata of different rigidities should impact the NC ratio. I did not find anything on this point in the literature. Do the authors know something about this?

We agree with the reviewer, our model indeed predicts that substrate rigidity, impacting on cell contractility and NE tension, would affect NC ratio. While the situation is of course way more complex than our model can describe, this is a very interesting point that we would like to address more in a future study. Interestingly, recent work from Pundel, Blowes and Connelly (<https://doi.org/10.1002/adv.202105545>), shows that extracellular physical cues affect nuclear and nucleolar volumes. Their results may be related to our model.

v) Related to the mathematical model: I appreciated the model and its explanation/parametrization strategy. I had one question on the choice of the exponential dependence of the nuclear concentration of macromolecules with external stress: why this choice? Why not linear in the first place? Especially because it seems that it is linearized in the equation defining V_n at the bottom of p. 5.

We thank the reviewer for the positive remarks on the model. We chose the exponential dependence under the hypothesis that biased transport due to the stress felt by the nuclear pores could be an activated process. However, as the reviewer correctly points out, this choice is equivalent to a linear-response assumption, and of course the data do not allow us to formulate any claim regarding this assumption. In the revised text we provided an explanation for our assumption and explicitly stated that to our scopes the assumption of linear response would be equivalent.

Minor comments

i) Figures

- adding the legend (control vs. ivermectin) on Fig. S3D would help the reader
- which condition is ivermectin in Fig. 4C? in general, what are the conditions in this plot?
- maybe ivermectin could be its own subfigure on Fig. 4 to help the reader and avoid going back to Fig. 3
- Y-axis in Fig. S2B: unit should be $\mu\text{m}^3/\text{h}$ and not μm^3

We apologize for the lack of clarity and mistakes. We improved the plot and amended the relative legend.

ii) Although the manuscript is nicely written, I found several typos, in particular in the appendix. I may have missed some, but here is a list of those I found:

- in methods section "cell transfection", problem unit polybrene
- in methods section "N2FXm", GFP-nes instead of NES
- in methods section "DAAM particle measurement", typo on "technique"
- in appendix, "left" and "right" hand side are inverted I believe
- in appendix, "nucleus surface tension and e contribution" → a
- in appendix, page 2, $c^{\text{ions_out}}$ c_{out} instead of just $c^{\text{ions_out}}$ (I believe)
- in appendix, section "estimates", in order TO confirm, prOessure
- in appendix, section "force biased", there is a verb lacking in first sentence
- p. 6 repetition "the the"

We apologize for these mistakes and we thank the reviewer for spotting them.

Reviewer #3 (Remarks to the Author):

This manuscript describes experiments and modeling of the regulation of nuclear size by osmotic forces and mechanical properties of the nuclear envelope in cultured mammalian cells. The study of the NC ratio in mammalian cells has been hampered by a lack of accurate volume measurements. This manuscript introduces an elegant new method for estimating nuclear/ cytoplasmic volume using a fluorescence exclusion-based approach, as well as a new FRET sensor for measuring nuclear envelope tension in cells. The work investigates NC ratio changes through the cell cycle and upon perturbations such as osmotic shock and cellular detachment from the substrate. The data support a mechano-osmotic model that implicates osmotic pressure and nuclear transport as critical components for nuclear size control.

In general, this work has potential to be an important advance in the field, but it is currently too preliminary for publication. There are many significant concerns described below. These include insufficient characterization of the new methods and a general tendency for over interpretation leading to inappropriate conclusions. They present some interesting initial observations without establishing sufficient context or additional insights. Some limited experimental data and careful rewriting will be needed to improve this manuscript.

We thank the reviewer for seeing potential in our work.

Major comments

1. The title, summary and parts of the abstract are not appropriate as key statements are not sufficiently backed by the data in this paper. For example, the Summary statement is: "Cytoskeletal forces exerted on the nuclear envelope impact on nuclear volume through modulation of force-coupled nucleo-cytoplasmic transport. However, there is little in this paper on cytoskeletal forces or nuclear transport. The closest test is an effect with ivermectin. However this result does not directly show that nuclear transport rates have changed. There is also no clear data on cytoskeletal forces on nuclear import; small effects with LatA are shown, but these may be highly pleiotropic and cannot be just assumed to be actin effects on the nuclear envelope. Their statement in the last paragraph "forces exerted by the cytoskeleton, affecting NE tension, impact on nuclear pores size (26–28)" have not been conclusively demonstrated in these cited papers and have not been proven here. While the authors certainly may hypothesize upon force-effects on nuclear transport and pores, these statements should not be the central conclusion of this paper. I think that the authors should address these concerns by judicious rewriting the wording of their conclusions, rather than being asked to perform the large number of experiments needed to support claims.

We agree with the reviewer that it is not the scope of our work to prove that forces exerted by the cytoskeleton affect nuclear transport by impacting on nuclear pores size. The phenomenon of tension-biased transport that we leverage to explain our data appears to be well supported by the literature. Indeed, the idea of tension-biased transport was introduced, supported and quantified in a seminal paper from the Roca-Cusachs lab (Elosegui-Artola et al., Cell 2017, ref. 27). The Roca-Cusachs lab recently published an elegant and systematic report on how cytoskeletal forces affect nuclear cytoplasmic transport (Andreu et al., NCB 2022, ref. 25). The idea that this process could be coupled to pore size was supported by two later independent structural studies, Zimmerli et al. (Science 2021, ref.26 ) and Schuller

et al. (Nature 2021, ref. 28), both showing that nuclear pores size depends on the cell's tensional state. However, this is not essential for us. In our study, Fig4.C, shows a clear correlation between NE tension and GFP.nls NC ratio, here used as a marker of nuclear cytoplasmic transport efficiency. As the reviewer highlights, based on our data it is reasonable to conclude that cytoplasmic forces, affecting NE tension, impact on nuclear cytoplasmic transport and nuclear volume. We completely agree that our data does not add anything on the question of nuclear pores size changes. We rephrased our conclusions accordingly in the revised manuscript.

2. There are important concerns about the N2FXm method that need to be addressed. Additional data and more detailed descriptions of the calibration method are needed so that the accuracy of the method can be evaluated (see below). The authors statement that the GFP "localizes in the entire cytoplasm except the nucleus" is not true. For example, other organelles besides the nucleus may also exclude the cytoplasmic GFP; lysozymes, mitochondria, lipid droplets etc., might represent 2-10% of cellular volume. This issue and how it may affect NC ratio measurements using N2JXm certainly needs to be addressed. Can the authors determine if these organelle volumes significantly affect the NC ratio measurements? Another concern is whether NES-GFP is truly all excluded from the nucleoplasm. Can the authors include a control using high-resolution confocal sections to show that there is no significant nuclear signal? If there is some nuclear signal, could they determine how that would affect the accuracy measurements? They should also show whether the localization of the NES-GFP marker is affected by their perturbations. For instance, mechanical or drug perturbations could cause entry of the NES-GFP in the nucleus, which could potentially affect their measurements and conclusions.

We thank the referee for these relevant remarks, which were also raised by reviewer 2. GFP-NES background in the nucleus, indeed, could affect the precision of N2FXm. Similarly, dishomogeneity in the distribution of GFP-NES in the cytoplasm, could perturb the second calibration, then impacting on nuclear volume estimates. We now explicitly discuss these possible drawbacks in the revised version of our manuscript. Additionally, following the reviewer's advice, we evaluated the distribution of the GFP-NES signal in correspondence of the nucleus (Rebuttal-Fig1.A). The mean radial intensity profile is minimum at the center and increases toward the periphery. In low magnification widefield images, the signal background simply originated by the presence of GFP-NES in the nucleus is expected to be rather constant or proportional to nucleus depth, then maximum at the center and minimum at the periphery. The profile we measured (middle panel in Rebuttal-Fig1.A) is instead compatible with a signal coming from the cytoplasm and then integrated over the cytoplasmic volume surrounding the nucleus. In the third plot in Rebuttal-Fig1.A we show the expected theoretical radial profile for a spherical nucleus (of unitary radius) embedded in a labeled cytoplasm:

$$f(r) = \int_0^r (1 - \sqrt{1-x^2}) dx = r - \frac{1}{2} * \arcsin(r) - \frac{1}{2} * r * \sqrt{1-r^2}.$$

This is compatible with the experimental profile we measured.

Importantly, we cannot exclude that a small portion of the signal we record in correspondence to the nucleus comes from inside the nucleus. Indeed we recorded a

background line on the radial profile that could have originated both by GFP-NES on top and bottom of the nucleus or by GFP-NES in the nucleus. However, when we evaluate in a confocal section (Rebuttal-Fig1.C) the proportion between cytoplasmic and nuclear GFP-NES signal, it becomes clear that the majority of the signal comes from the cytoplasm. In the mid-nucleus confocal sections, the mean intensity of GFP-NES in the cytoplasm is approximately 5 times higher than in the nucleus (Rebuttal-Fig1.C). Importantly, this ratio is not perturbed by drug treatments (Rebuttal-Fig1.C).

To allow the reviewers to judge directly if and how inhomogeneity in the distribution of GFP-NES signal in the cytoplasm could impact our measurements we plotted a representative example of a cumulative distribution of R^2 values, relative to more than 150 frames over 10 cells, for linear fits between GFP-NES cytoplasmic intensities and calibrated optical heights (Rebuttal-Fig1.B and ref to Figure 1.C). R^2 distribution is strongly biased towards 1, median = 0.93, sd = 0.14. Significant inhomogeneity in the distribution of GFP-NES signal, such as local accumulations or depauperation, would have perturbed the linear dependency between these two quantities and then the distribution of R^2 values.

All together these considerations suggest that GFP-NES background in the nucleus and as well as possible dishomogeneity in the distribution of GFP-NES in the cytoplasm and its impact on second calibration, are neglectable/minimal. This is probably due to the 'low' sensitivity and resolution of the imaging system settings as they are necessary for the N2FXm method to properly work, basically low numerical aperture and low magnification objective. Actually, it is important to notice that we challenged our method with two independent controls. First, we compared our nuclear volume measurements with 3D confocal reconstruction of nuclei, and we didn't record a significant shift in the distribution of nuclear volume estimates of the same cell population. Second, we measure in living cells the volume of artificial spherical objects (DAAMs particles). N2FXm measurements were in good agreement with theoretical volume computed by geometrical reconstruction. These two complementary validations well support the goodness of our method.

3. There are also concerns about the FRET-based nuclear envelope force sensor. This is a new sensor developed by this group, and its development is described only in this work and in a preprint. However, neither manuscript presents sufficient background characterization of this probe to demonstrate that it is acting as a force sensor at the nuclear envelope. Although the handful of data points are consistent, more quantitative and systematic controls would give more confidence that this sensor indeed measures force at the nuclear envelope.

We apologize for the lack of details relative to the NE tension FRET sensor we developed. The paper from Poli and al. is now accepted (DOI: 10.1038/s41467-023-37064-0). In the reply to reviewers section of that paper there are also additional controls. We hope now all the information relative to that valuable tool is publicly available.

4. One of the most surprising results is the linear growth of the nucleus vs. exponential growth of the cell volume. Other than pointing out the difference, this is not further investigated. More data are needed to confirm and develop this observation. One simple prediction is that NC ratios systematically fall as cells grow larger during interphase (however, this is not apparent in Figure 1F). Could this difference arise from some

systematic error in the N2FXm measurements? The reader is left without any context on how to think about this linear growth rate, or whether it carries much significance.

We agree with the reviewer: the observation that cell and nucleus volumes don't grow at the same pace is probably the most relevant result of our study. Most of our data focus on effects on time scales of minutes to less than an hour, and our explanations are fully coherent on these time scales. Cell-cycle time scales involve different processes and likely different explanations are necessary. Please see also reply to point ii of Reviewer#2. The fact that nuclear volume growth speed is constant (and independent of nuclear size) doesn't necessarily mean it is always slower than the cytoplasmic one. Indeed, while the cytoplasm doubles in volume during the cell cycle, the nucleus grows approximately 2.5 times (Fig2.E). To better clarify this point, we simulated time dependent volume growth curves for cyto and nucleus, and relative NC ratio (Rebuttal-Fig1.D). The cyto volume is assumed to grow exponentially while the nucleus linearly. The parameters used for the simulations, volume growth speed and minimal volume, are the ones obtained from the fit of experimental averages volume growth speed for RPE1 cell (Fig2.G). As the reviewer pointed out, at long time points, the NC ratio will collapse. However, for a time interval in the scale of a cell cycle and with the parameters obtained fitting the experimental data, the NC ratio is initially growing and lately starts to decay. For time points over the average span of a cell cycle, the NC ratio will certainly tend to zero. It would be for sure interesting to test our model in cells going to senescence and investigate the NC ratio dynamics in those conditions. This is however behind the scope of this work.

Moreover, we evaluated the time averaged values of cytoplasm and nucleus normalized volume growth speeds for all cell lines analyzed (Rebuttal-Fig1.D). Normalized average nucleus volume growth speeds are comparable, and typically higher (with the exception of the MCF7 cell line), than the one of the cytoplasm. This explains why the NC ratio doesn't systematically fall in the time scale of a cell cycle.

As suggested, in the new version of our manuscript we further discuss the linear versus exponential volume growth speed.

5. Another result is the apparent loss of nuclear volume after mitosis. Has it been confirmed by other methods or in other papers? Could the results be explained by the behavior of the NES marker at these cell cycle stages? If NC ratios are low after mitosis and then fall over interphase because of linear growth, then how does the nuclear volume ever catch up? How are NC ratios then maintained over multiple cell cycles? In Figure S2A, there is an example where nuclear growth suddenly speeds up after 50 frames (what is the frame rate?). Is this transition typical (as it is not seen in the averaged data)? Does it correspond to a known cell cycle transition? In general, these data are not discussed sufficiently in context to what is currently known about these processes. If the authors hope to describe thoroughly the NC ratio changes over the cell cycle, these issues should be clarified. Overall, the measurements of nuclear volume just before and after mitoses in Figure 2 represent a weak point in the manuscript and could even be deleted (or placed in another manuscript).

It is certainly true that our method can measure the volume of the nucleus only when this compartment is properly defined. In the few frames between nuclear envelope breakdown (NEB) and nucleus sealing after mitosis, centrally preceding nucleus postmitotic expansion (PME), the nucleus is not an isolated compartment anymore and can't be defined. During

this time interval, GFP-NES diffuses inside the nucleus and our method measures a residual volume that is still not accessible to it. We expressly clarify this point in the updated method session of our manuscript. For this reason, when we want to compare nuclear volume measurements before and after mitosis we compare them at NEB and PME, time points when the nucleus is still integer and when it is surely integer again, respectively. We apologize for the lack of explanation, nuclear post mitotic expansion (PME), the fast nuclear volume increase after mitosis, has been previously described, please see as example: Gerlich et al., NCB 2001; Baarlink et al., NCB 2017 and Krippner et al., EMBO Reports 2020. We now introduce PME with a relevant reference in the revised version of our manuscript.

Regarding how the nucleus can “catch up” its volume after mitosis, please see the reply to point 4.

The single-cell plot in Fig.S2A was intended to show the quality of the original data. Analyzing at the single-cell level the relationship between specific cell cycle stage or transition with nuclear and cellular volume growth is beyond the scope, and the reach, of this work. This goal can be potentially achieved with our method but will require a considerable amount of work in future studies.

Specific Comments

p.3 and Methods

More details of the calibration methods and evaluation of the accuracy of the approach are needed. First, can the authors provide an experimental version of Fig S1C (top right) to demonstrate linearity with the Texas red dextran marker they use?

We apologize for the lack of clarity, the plot in Fig1.C is indeed experimental. As previously mentioned, to better evaluate the robustness of fits along different frames and cells now in Rebuttal-Fig1.B we also show the cumulative distribution of R^2 relative to linear fits of GFP-NES intensity in function of cytoplasm optical heights measured with standard FXm. Please notice that the calibration is performed for each cell and at each time point.

Second, more detailed descriptions of how the NES-GFP intensity is calibrated with the cell volume data is needed. Can the authors show in a figure a real-life example (not just schematics)?

Please see the reply to the previous point. A real-life example was already provided in the original version of our manuscript.

For calibration, more details are needed on what cytoplasmic portions of the cell is used for comparison? How well do the red and green intensities inversely correlate; can the authors add plot showing correlation of green and red intensities per pixel? How is the noise in the intensity measurements dealt with for?

We apologize for the lack of clarity. For the GFP-NES calibration we use the intensity of all cell pixels excluding the ones in the nuclear area, as defined by the H2B signal. A real example of that is in Fig1.C. The linear dependency is robust, as can be appreciated by the cumulative distribution of R^2 for several single time frame linear fits from different cells (Rebuttal-Fig1.B).

Inhomogeneity in the cytoplasm could have a large impact on the interpretation of these measurements. Can the authors test whether inhomogeneities are significant enough to affect the calibration and subsequent measurements.? For example if there are many lysosomes in one part of the cytoplasm, is that portion of the cell not used for calibration?

We agree with the reviewer, that is an important point. Please see the response to your second major point. It is true, essentially, accumulation as well as depauperation of GFP-NES in the cytoplasm, as could be generated by vesicles excluding or concentrating the fluorescent protein, could perturb the linear dependency between the GFP-NES and the calibrated optical heights. Distribution of R^2 in Rebuttal-Fig1.B shows that is not the case.

L.7 p3 , L.6 p4 & L.27 p.5

Authors use the same nomenclature "NC ratio" for ratio of the nuclear to cell volume as well as nuclear to cytoplasmic volume. One definition should be used consistently. Also, the term "NC volume" might be reconsidered.

We apologize for this mistake and we amended it in the updated version of our manuscript.

L.12 p.4

"We also compared nuclear volume distribution of RPE1 cells measured with both N2FXm and 3D confocal reconstruction (see materials and methods) obtaining similar results"

This wording implies that the same cells were imaged with the two methods; is this true? The bar graph appears to show a sizable difference, but the authors say that it is not significant. Can they provide the mean values and the standard error? Can the authors include an example of the confocal z stacks with their markers. (This might address whether there is any detectable NES-GFP in the nucleus).

We apologize for the lack of clarity. The referee is correct, we did not measure the same cells, we refer to the same cell population. Now we clarified that sentence as: "We independently measured nuclear volume distribution of RPE1 cells with both N2FXm and 3D confocal reconstruction (see materials and methods). The difference between the distributions is not statistically significant."

In boxplot in Fig. S1.F, the boxes extend from the first to the third quartiles. The middle line represents the median. The whiskers extend up or down to $1.5 \times \text{IQR}$, where IQR is the inter-quartile range.

L.23 p.4

"DCIS.com "

Are the authors referring to the cell line "MCF10 DCIS.COM"? This should be clarified. Also, in the methods, the authors refer to DCIS.com medium; is this correct?

We apologize for the non appropriate naming. In the main text and in the method section of our revised manuscript we now specify that in text, figure and legend we used the name "DCIS.com" as an abbreviation of "MCF10 DCIS.COM".

Figure1

Can Panel D and E be plotted with the same y-axis to facilitate comparison.

Yes, in the new version of figure 1 now the two plots have the same scale.

Figure 2A

As the nucleus breaks down during mitosis, it is confusing what is being measured with this assay when there is no intact nucleus. This should also be clarified in the text. Consider removing these values after NEB on the graph, or make them a lighter orange color to denote that it is not really the nuclear volume.

This is a very important point and as suggested we clarified it in the text. We agree with the reviewer, indeed at the envelope breakdown there is no nucleus anymore and GFP-NES diffuse in the nucleus. Interestingly, with our method we measure a residual volume that is still not accessible to it. We could only speculate that this is the space occupied by the chromatin.

L.3 p5 & Figure 2C

"We found that NEB systematically precedes the onset of cellular roundup by ~10-20 min."

This statement is not backed well by the corresponding graph. The time points in the graph of 20 min apart do not allow for an accurate assessment of timing, and three of the five strains have similar percentages at -20 and 0 time points. These data should be discussed in relationship to previously published data on this time.

We agree with the reviewer and indeed in the text we specified: "However, the temporal resolution of our experiments (10 min) was too small to precisely distinguish these two events."

L10. P.5

"This implies a non-constancy of the NC volume ratio between NEB and PME (Fig. 2F)"

Can the authors present a statistical test between the paired Nuc and Cyto ratio results presented in Figure E and the NC ratio at PME and NEB in panel F.

We thank the reviewer for this suggestion and we added test results in the updated version of Fig2.

L12 p5

The results on growth rates need to be better described. Currently it is presented as the last graph in Figure 2, along the division data, and discussed in the same paragraph as the cell division, but describes behavior on an entirely different magnitude time scale. This is confusing to the reader. At the very least, the data on growth should be discussed in separate paragraphs for division in the text, and ideally presented in separate figures. As mentioned above, more context to this result should be given.

As suggested we describe this result in a separate paragraph.

L.17 p5

"Overall, these results indicate that in mammalian cells the nucleus-cytoplasm volumetric coupling could not be simply defined by a pure osmotic equilibrium, which would lead, instead, to a constant value of the NC ratio (16, 17)."

There are certainly circumstances in which NC ratios can vary even if they only use osmotic mechanism. Reference 16 shows that if the nuclear growth rate speed is proportional to the cell volume then a pure osmotic theory can explain N/C ratio maintenance. Can the authors show the Vol growth speed for nuclei as a function of Norm volume for cells? Is it flat or linear? If it is not linear than the pure osmotic model cannot explain these results.

As suggested by the reviewer we plotted nuclear volume grow speed as a function of cell volume (Rebuttal-Fig1.E). Nuclear volume grow speed doesn't correlate with cyto volume.

L.28 p5

“As expected from a purely osmotic model”

Can the authors clarify what they define as a "purely osmotic model". Can they refer to an equation? They cite this pure osmotic model with references 16, 17, but these papers use models that also take in account membrane tension.

We apologize, we correct that sentence with:“As expected from a model considering only osmotic equilibrium and membrane tension (eq.2 in SI appendix)” .

L.30 p.5

“Our model also shows that the same behavior is expected for a non-negligible constant surface tension, with a small correction on the slope, but the expected nuclear volume changes due to external forces are [...]”

Can they refer to an equation?

We now refer to eq.6 in the SI Appendix.

Figure 3 H & I : The plots look very similar, even in the SD. Please confirm that the NC data shown are correct.

We thank the reviewer and we confirm that data plotted are correct.

L.11 p.6

“was mostly unaffected along hyperosmotic shocks (Fig. 3E).”

Can the authors conduct a test for the Norm Mean Inv FRETindex at t=0 min and after the hypertonic shock t>10 minutes. It looks like there is a significant difference

We thank the reviewer for noticing it and indeed there is a statistical difference. Now we added statistical test analysis to the plots, please see updated version of figure 3.

Figure 3

Panel E,J,O. Can these be plotted with the same y axis to facilitate comparison?

Panel C,D,E. Can the authors label when the hyper osmotic shock occurs on the graph?

Figure 3 H & I.The plots look very similar, even in the SD. Please confirm that the NC data shown are correct.

Thank you for these suggestions. Following the reviewer's advice we updated Fig.3.

L.19 p.6

“nuclear and cytoplasmic volumes are strongly decoupled, with nuclear volume variations coherent with the changes of forces exerted on the NE and not with the changes of cytoplasmic volume (Fig. 4A).”

It would be important to test if NES-GFP localization is altered during these experiments.

Please find in Rebuttal-Fig1.C single plane confocal images of GFP-NES upon drug treatments. Moreover, the mean intensity ratio between cytoplasm and nucleus is not perturbed by drugs treatments.

L.21 p.7

"specific transcription factor such as YAP, key regulator of organ growth and regeneration as well as of mechanotransduction, also are affected by this mechanism."

Please cite a reference for this statement.

We apologize for this omission. We now add also here citation to ref.27 as it should be.

References

Update Lemiere reference #16

Fix Deveri reference #17

Fix Zimmerli reference #26

Thank you, we updated these references in the new version of our manuscript.

Rebuttal-Fig1. A. Left, widefield representative image of RPE1 cell expressing GFP-NES; highlighted in white the nucleus area; scale bar $20 \mu\text{m}$. Middle, average experimental radial profile of GFP-NES mean intensity in the nucleus area. Right, theoretical prediction of radial

profile. B. Cumulative distribution over several frames and different cells of R^2 relative to linear fits of pixels GFP-NES intensity in the cytoplasm in function of the corresponding pixel optical height computed with traditional FXm. C. Left, single plane confocal representative image of RPE1 cell expressing GFP-NES; highlighted in white the line used for the relative intensity scan; scale bar $20 \mu\text{m}$. Middle, line intensity profile relative to the image on the left. Right, cytoplasm to nucleus ratio of GFP-NES mean intensity for ctrl, Y-compound and latrunculin treated cells with relative representative images, scale bars $20 \mu\text{m}$. D. Left, simulations of time dependent cytoplasm and nucleus volume growth curves. Parameters used for the simulation were extrapolated from the fit of experimental grow speed of RPE1 cells (main Fig2.G). Middle, nucleus to cytoplasm volume ratio as obtained from the simulation. Right, average normalized cytoplasm and nucleus volume grow speeds. E. Conditional average of nuclear grow speed in function of cytoplasm volume.

REVIEWERS' COMMENTS

Reviewer #4 (Remarks to the Author):

The authors propose a new method to simultaneously measure cell and nuclear volumes in real-time. They also conduct experiment on different cell lines and propose a model on the relationship, thought the cell cycle, between cell and nuclear volume.

The method itself is an extension of a previous method used to measure only cellular volumes. The method is sound and validated. However, as a result, potential users may want to have a idea of the accuracy of the measured volumes, as an interval or confidence interval. Furthermore, this confidence interval may differ between cell and nuclear volumes, simply because nuclear volumes are smaller. Some explanations may be given for the slight difference between volumes obtained by 3D reconstruction and the proposed method. The p-value could also be mentioned.

I also have minor questions about the method:

- In M&M why a blue nuclear staining is mentioned since the method is based on nuclear exclusion measurement ?
- In a field of view, the total volumes of cells is measured or is it performed cell by cell ? How the "slightly enlarged cell area" is computed ?
- Which steps are automatic and which steps are manual in the measurements ?

The biophysical results seem sound, novel, and worth publishing, but I am not specialist of this domain.

Reviewer #4 (Remarks to the Author):

The authors propose a new method to simultaneously measure cell and nuclear volumes in real-time. They also conduct experiment on different cell lines and propose a model on the relationship, thought the cell cycle, between cell and nuclear volume.

The method itself is an extension of a previous method used to measure only cellular volumes. The method is sound and validated. However, as a result, potential users may want to have a idea of the accuracy of the measured volumes, as an interval or confidence interval. Furthermore, this confidence interval may differ between cell and nuclear volumes, simply because nuclear volumes are smaller.

As previously addressed in response to a reviewer's remark, we have validated the accuracy of our technique through two separate experiments. Initially, the distribution of nuclear volumes in Rpe1 cells was determined via 3D confocal reconstruction. This distribution is statistically in agreement with what was observed using N2FXm (refer to Supplementary Figure 1.g and the subsequent response for further details). Additionally, we used spherical particles with a refractive index similar to that of cells (this aspect is crucial), and specifically the "DAAM" particles, supplied by Daan Vorselen and described in their 2020 publication (Vorselen et al., <https://doi.org/10.1038/s41467-019-13804-z>). After cellular uptake of these particles, we assessed their volumes through geometric reconstruction (by measuring the diameter and calculating the volume) and compared them with those obtained using N2FXm. The results showed a strong correlation (detailed in Supplementary Figures 1.h-j). The ratio distribution of the volumes predicted geometrically versus those measured by N2FXm did not significantly deviate from a normal distribution with a mean of 1 (Supplementary Figure 1.h, p-value=0.39). Furthermore, a linear regression (using the `lm` function in R) comparing the volumes measured by N2FXm against the geometrically predicted volumes (Supplementary Figure 1.j) yielded a slope of 1.00 ± 0.06 .

We concur with the reviewer that our report does not include a confidence interval for our volume measurements. Providing such an interval is challenging due to the absence of a definitive volume standard, and the inherent uncertainties in any measurement technique (as discussed in the following point). The DAAM particles we used are, to the best of our knowledge, among the most suitable reference methods available. Based on our regression analysis of particle volume measurements, we estimate the methodological error to be approximately 6%. This figure is comparable to the roughly 10% error rate reported for standard Fxm as documented by Zlotek-Zlotkiewicz et al. in their 2015 study published in the *Journal of Cell Biology*.

Some explanations may be given for the slight difference between volumes obtained by 3D reconstruction and the proposed method. The p-value could also be mentioned.

Please note that we have independently measured the distribution of nuclear volumes in a population of RPE1 cells using both N2FXm and 3D reconstruction, as shown in Supplementary Figure 1g. It is important to underline that these two measurements cannot be performed concurrently due to the incompatibility of the N2FXm settings with 3D optical reconstruction. As a result, we used two different microscopes - a wide-field (10x objective) and a confocal (63x objective) for these measurements. There can be many reasons why there is no statistically significant difference between the two sets of measurements. The inherent uncertainty and intrinsic error in both methods could lead to the noted deviation when compared. However, given the lack of statistical significance, it is challenging (and

may be impossible) to discern if there are systematic effects at play. Our statistical test suggests there aren't. A p-value for a statistical test is reported in the legend of Supplementary Figure 1: "(g) A comparison of nuclear volume distributions considering volumes measured with the FXm (n=95) or with confocal 3D reconstruction (n=50). The Welch Two Sample t-test yielded a p-value of 0.30."

I also have minor questions about the method:

- In M&M why a blue nuclear staining is mentioned since the method is based on nuclear exclusion measurement ?

The reviewer is correct that the technique relies on preventing GFP-NES from entering the nucleus. Nevertheless, for the secondary calibration, it is necessary to label the cell pixels as either cytoplasmic or nuclear. To achieve this, we utilized GFP-NES and H2B-BFP as respective indicators of these compartments. Additionally, we verified that the expression of these fluorescent proteins does not alter the size of either the cell or the nucleus, as demonstrated in Supplementary Figure 1.k.

- In a field of view, the total volumes of cells is measured or is it performed cell by cell ? How the "slightly enlarged cell area" is computed ?

The measurement is performed for each cell individually. Additionally, it is important to note that the second calibration, which scales the GFP-NES intensity with respect to the measured optical thickness, is performed for each cell at each time point. This is done to correct for possible variations in GFP-NES expression between cells and over time. To improve the accuracy of the standard FXm, we automatically enlarged the identified cell contour by two pixels.

- Which steps are automatic and which steps are manual in the measurements ?

The analysis is semi-automatic. An ImageJ macro segments both cell and nucleus at each time point, and the user can check the accuracy of the procedure on line, and correct when needed.

The biophysical results seem sound, novel, and worth publishing, but I am not specialist of this domain.

We are grateful for the general positive judgment and for the positive recommendation.

Reviewer #1 (Remarks to the Author):

The authors have responded to almost all of my comments but may have misunderstood my first point where I emphasized the role of active transport. Their proposed revision discusses osmotic pressure and nuclear envelope tension and points to papers that review those. However, what I suggested was that in the introduction the authors specifically point to previous research on how the osmotic effects are actively regulated by the fact that transport through the nuclear pores is not diffusive and passive, but requires active regulation. Thus, some proteins are transported at a high rate from the nucleus to the cytoplasm but transported with lower rates from the cytoplasm to the nucleus. [Passive transport such as transport via channels results in equal rates in both directions.] For active transport, this results in steady-state, in a lower concentration of those proteins in the nucleus. The reverse situation can occur for other proteins. Thus, the osmotic pressure differences may be due to the different concentrations of the various proteins in the cytoplasm and in the nucleus, due to the difference in their active transport rates to and fro. I still feel that this should be explicated, contrasted with passive transport and discussed with references in the introduction.

We thank the reviewer for this valuable suggestion and we modified the introduction of our manuscript accordingly.

Reviewer #3 (Remarks to the Author):

The authors have not addressed sufficiently the major concerns of this work in this revision. While text revisions that soften and clarify the conclusions have improved the manuscript, there are still outstanding issues about the method as well as on the experiments on the distinction between linear and exponential growth. In this revision, there are no additional analyses or data, only a reciting of rebuttal arguments that have not changed our assessments.

We thank the reviewer for the time and effort dedicated to the accurate evaluation of our work. We are sorry for the negative outcome, but we have provided new analyses and comments that we believe should address the reviewer's concerns, which are partially due to insufficient clarity and depth in our previous analyses.

The concerns about the methods, its accuracy and possible GFP-NES leakage have not been addressed.

We believe we did address this concern in full, although our explanations might have been insufficiently clear. We summarize here our analyses and reasoning to address the reviewer's concerns.

The accuracy of our method has been tested with two independent experiments. First, we measured in Rpe1 cells the distribution of nuclear volumes by 3D confocal reconstruction. This distribution is not statistically different to the one obtained with N2FXm (Supp. Fig1.G). Second, we employed spherical particles whose refractive index is matched to the one of the cell (this aspect is crucial), the "DAAM" particles provided by Daan Vorselen and previously published in Vorselen et al. 2020 [doi: <https://doi.org/10.1038/s41467-019-13804-z>]. When these particles are engulfed by the cells, we measured their volume by geometrical reconstruction (essentially measuring the diameter and computing the corresponding volume) and, in parallel, by N2FXm. The two measurements are in good agreement (Supp. Fig1.H-J).

As stated in the first and second rebuttal letter, we cannot exclude leakage of GFP-NES in the nucleus and honestly it is not reasonable to expect that the nucleus is totally impermeable to the dye. However, what matters is that this is a minor effect, and if GFP-NES in the nucleus is the main source of signal in the nuclear area, the radial distribution of the signal should be maximum at the center and minimum at the periphery. With the settings used for N2FXm, wide-field microscopy with low magnification and low numerical aperture, we measured the contrary (Rebuttal Fig1.A). Thus, we believe that it is very reasonable to conclude that, although a non-zero background exists, the dominant component of the measured GFP-NES signal in the nuclear area is coming from outside the nucleus.

Finally, regarding how possible organelles in the cytoplasm could affect the second calibration, we show that the linear correlation between GFP-NES signal and cytoplasm optical height, as quantified with standard FXM, is very good (Supp. Fig1.F). Any strong inhomogeneity in the cytoplasm should have perturbed this correlation, which instead we find to be very robust.

The concerns about the distinction between linear and exponential growth of nuclear and cellular growth rates, which is a major novel take home of the paper, has also not been

adequately addressed. Figure 2G is the problematic figure used as the primary basis for this conclusion. We still think that apparent difference in the growth rate patterns between the nucleus and cell shown in Figure 2G are due to inappropriate scaling of the Y-axis used to plot the nuclear growth rate. As the nuclear growth rate values are many times smaller than the cellular rate, there is a problem of plotting the absolute values on the same graph. To illustrate this point for the editors, we have taken the points on the Figure 2G graph and rescaled it ourselves (See submitted figure). When nuclear growth rates are plotted with proper Y axis values, it clearly shows the rates are quite variable with different nuclear sizes: a pattern that is clearly non-linear and not so different from the cell growth rate pattern. Why one of these should be fitted to a linear function and the other to an exponential is not clear. Why these plots are binned differently also needs explanation. Thus, this data do not support the major claim that the nuclear growth rate is linear.

We admit that the representation provided with plots in Fig2.G and Supp. Fig2.B (we normalized the x-axis for cytoplasm and nuclear volume growth rate comparison) was not the most suited to support our conclusion that the nuclear volume is – on average - not growing as exponentially as the cytoplasm (as a side note, the binning difference is due to their basis in nuclear and cell volumes). We apologize and we regret to not have provided more promptly a better quantification. We also agree with the reviewer that there are further details beyond this main trend, which, however, we believe is robust.

To better explain this point and dispel any doubts we now computed the specific growth rate, $1/V*dV/dt$ by binned averages, independently for cell and nucleus, as previously done in Fig2.b of Cadart et al. [<https://doi.org/10.7554/eLife.70816>]. This alternative quantification clearly shows that (Rebuttal Fig2):

- I. the cytoplasm is compatible with exponential growth, but also with the cell-cycle trends observed previously (Cadart et al.) for HeLa cells;
- II. the nucleus (except for one outlier bin) shows a decreasing trend, meaning that its average growth must be sub-exponential;
- III. nucleus and cytoplasm do not grow at the same pace in terms of the average growth law followed by single cells.

It is important to notice that this last point is the core of our scientific message. Indeed, regardless of the details on the typical single-cell growth curve of the nucleus versus the cytoplasm, the novel and crucial finding is that the two compartments follow dramatically different growth rules, which we believe is beyond any debate (and is also very robust across cell lines in our data). This said, we agree with the referee that there are further details and sub-trends to be investigated regarding these single-cell growth laws.

In the new version of our manuscript we substitute the plots of volume growth speed with the ones of specific growth rate. Moreover, we toned down the conclusion that the nucleus grows on average at constant speed and we just propose it as a possible interpretation of the sub-exponential growth rate. This hypothesis is also supported by the fact that the residuals of the fit of the growth speed versus a constant value are normally distributed around zero.

Note: In Rebuttal figure 1E, similar non-linear nuclear growth rate patterns are also seen as a function of cellular volume; given the scaling between nucleus and cell size, this supports the significant variability in nuclear growth rates as a function of nuclear and cell size. (Note: in Figure 2G, the definition of the X axes used needs to be clarified in the figure legend).

We agree that there are cell-cycle sub-trends in the data, such as the cell-cycle dependent difference in exponential growth rate found by Cadart et al. However, as stated previously, the plots in Rebuttal Fig1.E clearly show that:

- I. for all the cell lines analyzed nuclear volume growth speed is not monotonically increasing with cell volume (thus its main trend is not compatible with exponential growth);
- II. the single-cell mean growth laws of nuclear and cytoplasm volume differ.

We stress once again that we agree with the reviewer that the shape of these curves is not trivial and there are further details that are interesting, but their interpretation, we believe, is beyond the scope of this paper. We have stated this point more clearly in the manuscript text. For a clarification about the X axes please see the response to the previous point.

Force sensor. We appreciate the rewording on the limitations of the nesprin force sensor. However, there is a question of how force from nesprin and cytoskeleton that is perpendicular to the membrane relates to tension parallel to the membrane which is presumably what regulates nuclear pore transport? The authors should be cautious in not equating these different forces.

We thank the reviewer for raising again this point. We are very conscious that the sensor we developed is measuring forces exerted by the cytoplasm to the nuclear envelope. Those are perpendicular rather than parallel to the membrane as we already expressly state in the description of the sensor.

Rebuttal Figure 2. Cell (top) and Nuclear (bottom) Volume Specific Growth Rates for Rpe1 cells.

Reviewer #1 (Remarks to the Author):

The authors have responded to the points we raised. We strongly suggest that they make the following final modifications related to these points before the paper is accepted.

We appreciate the recognition of our work during the revision process. We would like to express our gratitude to the reviewer for her/his helpful suggestions, which have significantly contributed to the improvement of our work.

1. Page 1 of remarks "We apologize...": No apology is needed, but the science should be explicated. I appreciate that that authors distinguish a "naive" tension model from one that accounts for active transport in the Discussion. However, I feel that since the active model (which also includes the "naive" model in the limit that the activity is set to zero) was previously published that it be contrasted with the model of the authors in the Introduction. I believe that this is a more accurate representation of the field and the contribution of the present paper that is a significant advance, and that we already said should, after revision, be published in Nat. Comm.

As suggested, we extended the introduction of the revised version of our manuscript. Now we added this sentence:

"For mammalian cells, the commonly accepted model considers cytoplasm and nucleus in mechanical equilibrium where the dominant forces are due to osmotic pressure, and nuclear envelope tension counteracting the growing tendency of the nucleus (9). Interestingly, a previous study on pig chondrocytes has already observed a differential behavior of the nucleus with respect to the cytoplasm in response to an osmotic perturbation, suggesting a possible decoupling of the two compartments, and described it by a model of a spherical gel with a membrane stress term subjected to osmotic loading (10)."

2. Major comments: Point 1: Similar to the spirit of the previous point, we think the work of Finan should be discussed in the Introduction. It does not detract from the very nice results of the present paper if the previous research is openly presented.

Please see the reply to the previous point.

3. Consistent with the spirit of points 1 and 2, we suggest that the authors explicate in the main text, the point they so nicely state in the response to Major comment: point 3, "We could say that nuclear.."

We thanks the reviewer for the suggestion and we added in the discussion of our revised manuscript this paragraph:

"More widely, nuclear envelope stress could have two possible independent effects, which can act on different timescales: one on nuclear import (and osmotic equilibrium), which we focus on in our study, which makes nuclear volume increase when the envelope is more stretched compared to a reference state; and one, that has a direct mechanical effect, and balances the difference of osmotic pressure between the nucleus and cytoplasm, by which

increasing tension leads to a decrease of nuclear volume for a given osmotic balance at steady state. In our study, due to cell spreading, we can hypothesize that we are in a regime of low effective constitutive tension, and the import effect dominates. A comprehensive understanding of how the combination of these complex opposite effects determine nucleus volume growth during the cell cycle would require further experiments and a clearer picture of the underlying mechanisms.”

4. We still do not understand why a compressive force on the nucleus (assuming, as do the authors, that it is transmitted mostly to the lamina) should result in an increase in the tension. Consider the opposite case of an extensile force -- for example, that the acto-myosin cytoskeleton is pulling on the nucleus: that pulling force would tend to expand the lamina and thus increase its tension. Thus, a compressive force on the nucleus might tend to buckle the lamina proteins and thus decrease the tension. The energy would then go into the buckling of the protein network, but the tension would be decreased; this is indeed the case for many biopolymers where stretching "costs" energy, but where compression results in buckling the network. We apologize if we are confused about this, but if the referee is confused, it might be that the readers will not follow the argument and the logic should be carefully explicated or modified as noted.

We apologize for the lack of clarity in the first rebuttal letter. For compressive cytoskeletal forces acting on the nucleus we intend forces that squeezing the nucleus as a whole, like compressing it from the top, would then generate an overall increase of NE tension. We agree with the reviewer that pointy acting forces, pinching inward the NE, are expected to buckle it and generate a local decrease of tension. It is difficult to figure out if these kinds of local forces are sufficient to then generate global effects. For the scope of our work, it is important to highlight and reiterate that direct forces, both positive and negative, are unlikely to deform the nucleus, the size of which is instead mainly set by mechano-regulated osmosis.

5. Minor points: point 4: We read what the authors added to the supplementary information and have a further comment. The fact that the fluctuations decrease may not be due to an decrease in the tension due to activity, but rather, to an increase in the "fluctuating forces" that act on the system due to activity, over and above the thermal fluctuating forces. In other words, the tension might remain the same, but since the fluctuations depend on the ratio of the tension and the magnitude of the fluctuating forces, the fluctuations might increase due to an effectively higher temperature, and not a lowered tension due to activity. The authors can see, for example, <https://journals.aps.org/prl/abstract/10.1103/PhysRevLett.106.238103>

The authors should consider and explicitly note this possibility.

This is exactly what we meant (see e.g. the introduction of Introini et al. 2013, where this point is explained in detail). We apologize for the lack of clarity, we modified the relative paragraph of the Appendix as follows:

“The apparent tension of the nucleus, estimating σ_N or σ_0 is found to be around 10^{-6} N/m in nuclear shape fluctuations experiments (Chu et al., 2017; Introini et al., 2023. However, this value is an underestimation of the actual tension, since it is affected by excess flickering due to transient deformations of active origin, since fluctuating forces due to activity decrease the effective tension with respect to the actual mechanical tension (see Introini et al., 2023 for a

detailed explanation). We can assume that this value is a lower bound for the real mechanical tension σ_0 ."

Reviewer #2 (Remarks to the Author):

I am mostly satisfied with the way the authors addressed my concerns. I believe that the novel methodology and its subsequent application in this study grant publication in Nature Communications. This methodology will certainly be applied by other research groups and stimulate investigations on nuclear volume regulation.

We appreciate the recognition of our work during the revision process. We would like to express our gratitude to the reviewer for her/his helpful suggestions, which have significantly contributed to the improvement of our work. Additionally, we are pleased by the reviewer's enthusiasm for our project and her/his recognition of the potential impact it could have in the community.

Reviewer #3 (Remarks to the Author):

In general, there are several major concerns of this work that have not been suitably addressed in this revision. Most of the concerns were discussed but not directly addressed. In fact, the additional data and information provided in this round further reveal the weaknesses in the data. Thus, there are still significant reservations about many aspects of this manuscript, including the validity of the N2FXm assay and the distinction of linear vs exponential growth rate. This work unfortunately is not ready for publication in any journal.

We would like to express our gratitude to the reviewer for her/his helpful suggestions, which have significantly contributed to the improvement of our work.

We are sorry but we disagree with the reviewer. The new quantifications and data provided in the first rebuttal letter, are not affecting the conclusions of our work and moreover they confirmed the observed decoupling between cellular and nuclear volume growth dynamics.

Major concerns

1. The title, summary and parts of the abstract are not appropriate as key statements are not sufficiently backed by the data in this paper.

Following editor and reviewer suggestion we propose to change the working title of our manuscript in: "Differential osmo-mechanical regulation of nuclear and cytoplasmic volume in mammalian cells".

We also toned down the abstract.

-The authors revised the text slightly to remove the implications that this work shows direct effects on nuclear pore size. However, the current title is still inappropriate as places emphasis on elements that have not been sufficiently tested in this work. As remarked in the previous round, the involvement of nuclear import (the conclusion of the title) is supported only by an effect of ivermectin, an inhibitor of α/β importin- based nuclear transport. Broader effects of actin inhibition are not considered. Thus, there is insufficient experimental data to make this the take-home message. To do so would require new experiments that directly document a change in nuclear import rate that is dependent on nesprin and its ability to bind to actin at the nuclear envelope.

As stated in the original rebuttal letter, it is not the scope of our work to demonstrate the existence of a tension-biased nuclear transport. This mechanism has been conceived and quantitatively described in two seminal papers from Roca-Cusachs lab (Elosegui-Artola et al., Cell 2017 and Andreu et al., NCB 2022). Moreover, two structural papers from different labs also suggest that cyto-skeletal tension could stretch nuclear pores (Zimmerli et al., Science 2021 and Schuller et al., Nature 2021). Although of course it is possible that other processes are in place, this evidence is well in line with Roca-Cusachs model. In our work we rely on this well established line of literature. In Figure 4C, we also show that GFP-NLS nuclear to cytoplasmic ratio, here used as a marker of nuclear "permeability", changes in function of nuclear envelope tension as measured by our sensor. Since the reviewer

suggested to not state a direct link from NE tension, dilatation of nuclear pores and nuclear permeability, we avoided that in the revised version of our manuscript. We still mention it as a possible speculative mechanism. However, the fact that NE tension impacts on nuclear volume, besides being related to past literature, is clearly supported by our data and is an essential element of the model we proposed. We also showed that, to our knowledge, forces required to directly extend the nucleus are out of scale for the cell (see reply to reviewer 1 minor point 3).

2. There are important concerns about the N2FXm method that need to be addressed. Additional data and more detailed descriptions of the calibration method are needed so that the accuracy of the method can be evaluated (see below).

-The authors have not provided sufficient additional data to address the multiple concerns of this method. There is still little data regarding its accuracy.

-Issue of organelle volume in the cytoplasm was not addressed.

As we show in the original rebuttal letter (see reply to reviewer 2 point 1-i and reviewer 3 major 2), the linear correlation between GFP-NES signal and cytoplasm optical height, as quantified with standard FXM, is very good (see rebuttal Figure 1 B). Any inhomogeneity in the distribution of GFP-NES signal in the cytoplasm, as could be generated by cytoplasmic organelles, would have perturbed this distribution.

-Is GFP-NES really excluded from the nucleus? Although the data in rebuttal Figure 1 is used for the authors to claim that this is not an issue, the data in rebuttal Figure 1 AC clearly shows measurable non-zero intensity in the nucleus in the confocal imaging in the image and the graph. The modeling in rebuttal Figure 1A does not rule out GFP-NES in the nucleus, and the graph in Figure 1A shows a non-zero intensity at the base of the graph. As the authors say, the leakage of GFP-NES into the nucleus would lead to inaccurate measurement of nuclear volume. For instance, it might explain the unexplained discrepancy between the measurements using confocal 3D reconstructions and the N2FXm.

In rebuttal Figure 1 A we show the radial distribution of the GFP-NES signal in the nucleus in wide-field images (the same used for N2FXm). It is clear that the signal is minimum at the center and increases toward the nuclear periphery. That is not compatible with a background of nuclear GFP-NES, that would be rather constant or maximum at the center and minimum at the periphery. On the contrary, this result is perfectly in line with what would be expected from a signal coming from the cytoplasm and then integrated over the cytoplasmic volume surrounding the nucleus. Clearly, the signal doesn't start from "zero" (as is not zero in confocal scan). This non zero line, could correspond to the portion of cell volume on top and below the nucleus center, or, as we clearly stated, could also be originated by nuclear GFP-NES. However, the impact of this background is minimal and does not affect our measurements. Indeed, as summarized in the rebuttal letter, we challenged our method with two independent controls. First, we compared population measurements of nuclear volumes performed with N2FXm and standard 3D confocal reconstructions, and we didn't record a significant shift in the distributions. Second, we measured in living cells the volume of artificial spherical objects (DAAMs particles). N2FXm measurements were in good agreement with theoretical volume computed by geometrical reconstruction.

-Do GFP-NES levels in the nuclear region change during drug or mechanical perturbations? The new data in rebuttal Figure 1C show that the distribution of intensity values certainly change (at least the variability increases) suggesting that at least for a good portion of cells that this might be a real issue. Changes in GFP-NES in the nucleus would further raise serious questions into the validity of the measurements.

We apologize for lack of clarity, we did not specify that we statistically tested the difference, and it is not significant (t-test vs Ctrl, p-values: Y-comp=0.140; Lat=0.0618).

3. There are also concerns about the FRET-based nuclear envelope force sensor.

-The authors should clarify the description of the force sensor. This description could be similar to what they explained to reviewer 1: "The FRET tension sensor we developed is designed on Nesprin1, one of the components of the LINC complex. It spans from the inner nuclear envelope, where it binds SUN domain containing proteins, to the cytoplasm, where it binds to actin. Therefore, it is sensitive only to forces exerted by the actin cytoskeleton to the NE." Importantly, it should be clarified that this FRET probe most likely does not sense membrane tension (parallel to the membrane) but rather forces by actin directly attached to the Nesprin probe (perpendicular to the membrane).

We thank the reviewer for this suggestion and we improved the description of the sensor in the revised version of our manuscript accordingly. We now also clearly state its possible limitations. As suggested we added in the main text: "This tension sensor spans from the inner NE, where it binds SUN domain containing proteins, to the cytoplasm, where it binds to actin. Therefore, it is sensitive only to forces exerted by the cytoskeleton to the NE and does not sense tension parallel to the membrane (23)."

4. One of the most surprising results is the linear growth of the nucleus vs. exponential growth of the cell volume. Other than pointing out the difference, this is not further investigated. More data are needed to confirm and develop this observation.

-New data on this point only further questions of the validity of this claim of linear growth. The graphs used to support linear growth of the nucleus (Figure 2G and rebuttal Figure 1E) are inconsistent with this conclusion. Rebuttal Figure 1E shows that the points for each cell type simply do not obviously fit a flat line (linear) or a line with positive slope (exponential). In Figure 2G, this deviation from a flat line is hidden by inappropriate choice of Y-axis values but is apparent by close examination (Note that this graph shows the same points as Rebuttal Figure 1E). Hence, the data do not support their conclusion that nuclear growth is linear, which is considered a major take-home of this work.

Rebuttal Figure 1E was produced to reply to a reviewer 3 minor point. Please, note that this plot does not show at all the same points of figure 2G. Here, conditional average is not performed over nuclear volume, but, as requested by the reviewer, over cell volume. Hence, the two plots are different, and they address two different questions. Nuclear Volume Growth Speed is indeed here averaged over bins of cell volume and not over bins of nuclear volume. Nothing is hidden by inappropriate scaling. Interestingly, the shape of the curve in Rebuttal Figure 1E is not trivial and we thank the reviewer for pointing our attention to this point. While it deserves more investigation, we can surely conclude that this curve is clearly more similar to a flat line than to an asymptotically growing one. These results are not affecting the

conclusions of Figure 2G. Moreover, they confirm a robust decoupling of nuclear and cell growth dynamics in single cells, a main point of our work. Regarding the flat line fits in Figure 2G and Supplementary Figure 2B, distributions of fit residuals are normal and their means are not statistically different from zero.

Other issues

Figure 1D. The Y-axis was changed in response to reviewers. However, the axis change was very crudely done, which in effect visually changes the values of the graph.

We apologize, our mistake. When we adjusted the size of text in the plot axis assembling the figure we shifted it by a few pixels. We thank the reviewer for spotting this mistake and we corrected it in the revised version of our manuscript.

The information on error bars, number of replicates, N must be provided for every experiment shown.

All this info is now in figure legends or in the method session.

Many details in the methods seem to be lacking. For instance, what were the concentration and time of treatment of ivermectin? How were the osmotic shocks performed.

This information was in the Supplementary Materials. In the revised version of our manuscript we renamed the relative section as: "Drug, adhesion and osmotic perturbations experiments". As stated, cells were treated with ivermectin (Sigma Aldrich) at 10 μ M for 24 h. For hyper-osmotic shock experiments, D-sucrose (Carlo Erba reagents) was resuspended in pure water at 500 mM and then added to cells.

Reviewer #1 (Remarks to the Author):

This paper experimentally explores the effects of various biochemical and mechanobiological perturbations of single cells on the ratio of their nuclear to cytoplasmic volumes. It is shown that changes in cytoskeletal forces exerted on the nucleus (via latrunculin or ROCK inhibition) changes the NC ratio in a manner that is not simply due to the direct effect of those forces on the osmotic pressure balance. In addition, it is shown that while the NC ratio in many cell types (under hyperosmotic conditions) is constant during most of interphase, this ratio varies considerably during cell division. Interestingly, it is found that right after cell division, the cytoplasmic volume increases exponentially, while the nuclear volume increases only linearly with time. The results will stimulate further experiments and theoretical models and should be published in Nature Communications, pending the following changes.

We are grateful for the positive remarks, and for the positive recommendation.

We believe that the authors overstate the impact of their results with respect to the concepts presented in Refs. 7, 16, and 17. In particular, the present manuscript uses the term “falsify” (once) “or “falsified” (twice) to relate their findings to the presumably naïve “osmotic” models in these references. This, besides being polemical, is not correct and does not reflect the subtlety of the ideas previously presented. While the NC volumetric ratio is indeed determined by osmotic pressure differences, the proteins involved are those whose active transport through the nuclear pores, results in differing concentrations in the cytoplasm and nucleus. The import rate and export rates can be different for systems with active transport, resulting in concentrations of proteins that are directly related to these rates. When various perturbations are applied to the cell, including changes in the cytoskeletal forces that act on the nucleus, those transport rates are modified, with – in general – different changes for the import and export rates. Thus while it is true that the naïve osmotic effect of changes in cytoskeletal force are not significant in the experiments described, there is an important indirect effect of those forces on the transport rates that determine the protein concentrations in the nucleus and cytoplasm, and hence the osmotic pressure difference and NC ratio. This was indeed a major point of these references which seems to have been missed in the present manuscript.

Nevertheless, the paper is interesting and presents several novel results about how the NC ratio changes during the cell cycle and under biochemical or mechanobiological changes, which justify publication in Nature Communications. However, before we can formally recommend publication, the interpretation of these changes with respect to such perturbations should be rewritten to accurately incorporate the previously discussed changes in osmotic pressures due to active transport and to qualitatively relate these transport changes to the perturbations applied in the present manuscript. In addition, we request the authors to address the following major and minor comments.

We apologize to the reviewer for our miscommunication. Our intent was not to go against previous models (which of course had different scopes and were not informed by our data), and we meant our statement referred exclusively to the ingredients of our own model, which is a naive version of the osmotic model, taking into account just few essential ingredients

and making some simplifications in order to keep a minimal number of parameters. We have rephrased the statements in order to avoid any misunderstanding, and also added a statement on the results of the cited refs. Specifically we added this sentence in the discussion section: “When perturbations are applied to the cell, including changes in the cytoskeletal forces that act on the nucleus, transport rates can be modified, with – in general – different changes for the import and export rates. Thus while it is true that the naïve osmotic effect of changes in cytoskeletal force are not significant in our experiments, there is an important indirect effect of those forces on the transport rates that determine the protein concentrations in the nucleus and cytoplasm, and hence the osmotic pressure difference and NC ratio. This was a major point of previous studies [REFS 7-16-17].”

Major comments:

1. At the bottom of page 5, the authors write “Our model also shows that the same behavior is expected for a non-negligible constant surface tension...”. Compared to what scale is the surface tension non negligible? The work by Finan et al. (Annals of Biomedical Engineering, 2009) presents the notion of NE that is stretched compared to one that is relaxed. The authors should relate the findings of their model with respect to the work of Finan et al.

We thank the reviewer for this comment and we refined the inaccurate statement in the current version of the manuscript. We have explored constitutive surface tensions in the range $1e^{-6}$ to $1e^{-2}$ N/m. The value around $2e^{-2}$ N/m found by Finan and coworkers by a fit of their model can be compared with the upper bound of the constitutive surface tension that we explored. Hence, our results should apply (see e.g. fig SM3 in the SI appendix). It is important to state that their richer model (based on the swelling of porous gels) contains other parameters, hence although the basic underlying physical picture of our simpler model is similar, it is possible that the values of the parameters do not map exactly between the two frameworks. Finally, we note that while we explore for simplicity our tension-biased transport model in a regime of negligible tension, this choice has no impact on our main statement, as the choice of a finite/large constitutive tension would make it equally difficult for osmotic and mechanical extensile forces to swell the nucleus.

2. The authors conducted three types of experiments to modulate the forces that are exerted on the cell and are transduced into the nucleus: hyperosmotic shock, cell detachment, and cell spreading. These types of experiments were thoroughly investigated in previous works such as Finan et al. (mentioned above), Guo et al. (ref. 19 in the manuscript), and others. However, discussion that relates the results of the manuscript to these previous works is lacking. The authors should present their experimental results in a wider context and relate them to these previous works.

The work of Finan and coworkers has the merit of showing that the volume increase of the nucleus upon hypo-osmotic shocks, saturates and does not follow a linear increase as the cell volume does (Ponder plot). The simplest interpretation of that is that part of the pressure is compensated by nuclear envelope tension, leading to less volume increase than expected when tension is negligible.

We could say that nuclear envelope stress can have two effects, which can act on different timescales: one on nuclear import, which we focus on in our study, which makes nuclear volume increase when the envelope is more stretched compared to a reference state, and one that has a direct mechanical effect, and balances the difference of osmotic pressure between the nucleus and cytoplasm, by which increasing tension leads to a decrease of nuclear volume for a given osmotic balance at steady state. In our study, due to cell spreading, we can hypothesize that we are in a regime of low effective constitutive tension, and the import effect dominates.

3. At the bottom of page 37, the authors relate the tension to external force by the equation $F_{ext} = 8\pi RN \sigma_{aext}$. The authors should explain this relation as its derivation is unclear. Perhaps a cartoon showing the force, tension, and the relation between them can make the derivation clearer.

We apologize for the missing elements in our argument. We have clarified the assumptions, and we have specified in the revised text that we intend this relation as a simple estimate relating a scenario of tension of external origin to the total magnitude of the radial forces that are needed to generate the equivalent mechanical pressure difference.

4. The authors model the effect of force on nuclear import using an exponential relation. The author should discuss the underlying biophysical reasoning of this dependence or state it is just a fitting function.

We agree that this assumption needs an explanation - which was pointed out also by reviewer 2. We have made it clear in the text that we have assumed an activated process. This assumption does not really affect the quantitative aspects, since other functional forms would be equivalent in the relevant range of values.

5. In their theoretical model, the authors treat the external force as a scalar although by definition force is not a scalar. Furthermore, the authors assume that the nucleus is spherically symmetric, which is not true in general, when deformed by non-isotropic external forces (that may have shear components).

We have clarified these aspects. We chose the spherically symmetric case to simplify the model from parameters that could not be fixed reliably based on our data. Given the assumption, the force field only has one relevant degree of freedom, hence the sloppy notation (now clarified). Additionally, we have clarified that we mean the model in this study more as a conceptual guide to understand the experiments than as a full-blown theoretical description. The purpose is to isolate and select scenarios, hence we systematically reverted to the simplest assumptions.

6. The authors state that they assume the forces that are exerted on the nucleus are compressive. In that case and when the forces are spherically symmetric, the contribution of the external forces to the tension should be negative. Is this the case? This should be clearly explicated since this affects the way the exponential relation between the surface tension and nucleocytoplasmic ratio is interpreted.

We have clarified this part in the revised SI text. In our notation the contribution of compressive external forces to the tension is positive (as they both contribute to forces pointing inwards). We analyzed both cases of positive and negative contribution of external forces to the tension, but given the previous experimental literature [e.g. <https://doi.org/10.1073%2Fpnas.0908686106>] we believe that the first scenario could be more likely.

Minor comments:

1. Can there be forces that are exerted on the nucleus which will not be detected by stretching of the LINC complex (e.g. squashing of the nucleus)?

This observation is correct. The FRET tension sensor we developed is designed on Nesprin1, one of the components of the LINC complex. It spans from the inner nuclear envelope, where it binds SUN domain containing proteins, to the cytoplasm, where it binds to actin. Therefore, it is sensitive only to forces exerted by the cytoskeleton to the NE. If external forces applied to the cell indirectly also affect the link between actin and SUN proteins at the NE, then the sensor can detect it. More detail about the FRET sensor can be found in Poli et al 2023 [<https://doi.org/10.1038/s41467-023-37064-0>]. Please consider also our response to point 3 below.

2. Some works (e.g. Jahed et al., Journal of Cell Science 2016) suggest that the cytoskeleton and LINC complex are indirectly involved in nucleocytoplasmic transport by modulating the Ran protein gradients. Is it possible that the nucleocytoplasmic transport in the latrunculin-treated or the ROCK inhibited cells is affected via this pathway rather than mechanical “expansion” of the NPCs? Can the author comment on this possibility.

We do not exclude that there may be additional/other mechanisms at play, as suggested by the reviewer. However, we have observed a strong correlation between nuclear envelope tension, as measured by our FRET sensor, and the mean intensity ratio of GFP-NLS between the nucleus and cytoplasm (used as a marker of protein nuclear incorporation efficiency) (Fig. 4C). This dependency is essential for our model, independently of whether this is due to nuclear pores expansion or to other possible mechanisms.

3. At the bottom of page 6 the authors write that the cytoskeletal forces are compressive. However, these can also be expansive if the nucleus is laterally stretched. Can the authors provide a reference to a work that distinguish between these two possible modes of nuclear deformation?

The reviewer’s observation is correct. It could be, in certain circumstances, that positive forces are exerted to the nucleus. With our model, indeed, we analyzed the effect on nuclear volume of positive as well as negative external forces (always with the assumption of a spherically symmetric nucleus). In both cases variations in nuclear volume induced by direct forces are negligible with respect to the ones arising from changes in osmolarity (less than 10% for σ_0 in the range 10^{-6} to 10^{-2} N/m).

We now modify that sentence in: “Based on the model, we reasoned that this was very unlikely to be a direct effect, as, independently of the direction, compressive or stretching, the force required to change nuclear size would be abnormally large (see SI appendix) [24, Kalukula et al, Nature Reviews Molecular Cell Biology, (2022), 583-602, 23(9), DOI: <https://doi.org/10.1038/s41580-022-00480-z>].”

4. The authors mention that the estimate of the NE tension (10^{-6} N/m) is a lower bound due to fluctuations of active origin of the NE shape. Are there evidence for such active fluctuations? While this does not change the conclusion of the authors, it is worthwhile to give a reference for such active processes that facilitate NE shape fluctuations.

We agree and we have added this statement: “The apparent tension of the nucleus, estimating σ_N or σ_0 is found to be around 10^{-6} N/m in nuclear shape fluctuations experiments (Chu et al., 2017; Introini et al., 2021).”

Reviewer #2 (Remarks to the Author):

Pennacchio et al extended the powerful fluorescence exclusion microscopy, FXm, to simultaneously measure cell volume and nuclear volume – a new method they coined “N2FXm”. This novel method is sound and allowed the authors to study the nuclear-to-cell volumetric (NC) ratio. First, they found that the NC ratio was not constant during the cell cycle, being particularly impacted by mitosis. Second, they showed that the cell and nucleus did not follow the same growth laws. Third, they discovered that the NC ratio could not be explained solely by an osmotic balance and that cytoskeletal forces exerted onto the nuclear envelope could bias nuclear import and nuclear osmotic pressure, thus also participating in setting the NC ratio.

The manuscript is clearly written, and the data are of high quality and interest, shading light on novel regulatory mechanisms of nuclear volume. I would greatly recommend publication should the following questions be addressed.

We thank the reviewer for the overall positive judgment of our work and for the enthusiastic recommendation.

1/ The N2FXm method

I found the N2FXm method extremely powerful and relevant. Although most of my concerns are addressed in the methods or in the paper, I have one additional question and one recommendation for the authors:

i) Have the authors checked that there is no GFP-NES in the nucleus, and if the signal is homogeneous in the cytosol? Depending on the cell state/strength of the NES, there may still be some GFP in the nucleus, which would affect the measurement – especially if this bias is cell cycle-dependent, through for instance NE tension impacting nuclear export. Even if non-significant, there seems to be a small difference in the nuclear volume measurement between the 3D reconstructed and the N2FXm in Fig. S1F. Moreover, this could also be important in the perturbation experiments, impacting the actual measurement.

We thank the referee for these relevant remarks, which were also raised by reviewer 3. GFP-NES background in the nucleus, indeed, could affect the precision of N2FXm. Similarly, dishomogeneity in the distribution of GFP-NES in the cytoplasm, could perturb the second calibration, then impacting on nuclear volume estimates. We now explicitly discuss these possible drawbacks in the revised version of our manuscript. Additionally, following the reviewer's advice, we evaluated the distribution of the GFP-NES signal in correspondence of the nucleus (Rebuttal-Fig1.A). The mean radial intensity profile is minimum at the center and increases toward the periphery. In low magnification widefield images, the signal background simply originated by the presence of GFP-NES in the nucleus is expected to be rather constant or proportional to nucleus depth, then maximum at the center and minimum at the periphery. The profile we measured (middle panel in Rebuttal-Fig1.A) is instead compatible with a signal coming from the cytoplasm and then integrated over the cytoplasmic volume surrounding the nucleus. In the third plot in Rebuttal-Fig1.A we show the expected theoretical radial profile for a spherical nucleus (of unitary radius) embedded in a labeled cytoplasm:

$$f(r) = \int_0^r (1 - \sqrt{1-x^2}) dx = r - \frac{1}{2} * \arcsin(r) - \frac{1}{2} * r * \sqrt{1-r^2}.$$

This is compatible with the experimental profile we measured.

Importantly, we cannot exclude that a small portion of the signal we record in correspondence to the nucleus comes from inside the nucleus. Indeed we recorded a background line on the radial profile that could have originated both by GFP-NES on top and bottom of the nucleus or by GFP-NES in the nucleus. However, when we evaluate in a confocal section (Rebuttal-Fig1.C) the proportion between cytoplasmic and nuclear GFP-NES signal, it becomes clear that the majority of the signal comes from the cytoplasm. In the mid-nucleus confocal sections, the mean intensity of GFP-NES in the cytoplasm is approximately 5 times higher than in the nucleus (Rebuttal-Fig1.C). Importantly, this ratio is not perturbed by drug treatments (Rebuttal-Fig1.C).

To allow the reviewers to judge directly if and how inhomogeneity in the distribution of GFP-NES signal in the cytoplasm could impact our measurements we plotted a representative example of a cumulative distribution of R^2 values, relative to more than 150 frames over 10 cells, for linear fits between GFP-NES cytoplasmic intensities and calibrated optical heights (Rebuttal-Fig1.B and ref to Figure 1.C). R^2 distribution is strongly biased towards 1, median = 0.93, sd = 0.14. Significant inhomogeneity in the distribution of GFP-NES signal, such as local accumulations or depauperation, would have perturbed the linear dependency between these two quantities and then the distribution of R^2 values.

All together these considerations suggest that GFP-NES background in the nucleus and as well as possible dishomogeneity in the distribution of GFP-NES in the cytoplasm and its impact on second calibration, are neglectable/minimal. This is probably due to the 'low' sensitivity and resolution of the imaging system settings as they are necessary for the N2FXm method to properly work, basically low numerical aperture and low magnification objective. Actually, it is important to notice that we challenged our method with two independent controls. First, we compared our nuclear volume measurements with 3D confocal reconstruction of nuclei, and we didn't record a significant shift in the distribution of nuclear volume estimates of the same cell population. Second, we measure in living cells the volume of artificial spherical objects (DAAMs particles). N2FXm measurements were in good agreement with theoretical volume computed by geometrical reconstruction. These two complementary validations well support the goodness of our method.

ii) I believe one important point is that the GFP calibration is done at every time point and for each cell, thus limiting temporal fluctuations or cell-cell heterogeneity in GFP concentration. As such, it may be better to directly state it in the main text, and not only in the methods.

We thank the reviewer for this comment and as suggested we now specified it in the main text.

2/ Nuclear growth and NC ratio

I have a few questions regarding nuclear volume growth and subsequent NC ratio values, as well as their interpretations based on the model proposed by the authors:

i) The NC ratio seems rather constant in each cell line from -500 min to 0 min. First, can the authors provide a y-axis zoom before and after mitosis, to have a better visualization (Figs. S1L & 1F)? Second, it seems constant even though nuclear volume does not increase exponentially. Can the authors comment on this point?

To enhance the difference in NC ratios we showed it at its maximum, between PME (post mitotic expansion, nucleus life start) and NEB (nuclear envelope breakdown, nucleus life end), please see plot in Fig2.F. NC ratios are not constant and typically increase by at least 20% along the nucleus lifetime. We added this quantification in the main text of the revised version of our manuscript. As suggested, we also added in Fig2.F a graphical representation of relative paired t-tests results.

ii) Can the authors speculate (and potentially expand the discussion on this point) as to why the nucleus would grow linearly in time? Per the authors' model, if it does not increase exponentially, this could for instance mean either that import/export is cell cycle-dependent such that nuclear osmotic pressure is not constant over the cell cycle. This could for instance be the case if NE tension changes during the cell cycle (see the question on tension in point 3/ below).

The average linear growth on cell-cycle time scale is an intriguing fact that we report experimentally, but whose explanation goes beyond our scopes. Most of our data focus on effects on time scales of minutes to less than an hour, and our explanations are fully coherent on these time scales. Cell-cycle time scales involve different processes and likely different explanations are necessary.

This said, we can reason on how the data relate to the prediction of different simple models.

Let us suppose first a scenario of perfect osmotic force balance. If the tension were always negligible, the volume of the nucleus would grow exponentially as the total volume. Our analyses suggest that the tension would have to be too large to have a relevant impact on nuclear size and explain the average linear growth.

If we consider our model coupling tension with transport, we can ask whether extrapolating it to cell-cycle time scales would support linear growth. In this model the observed tension with cell cycle would increase nuclear import, causing the nucleus to swell more than predicted by osmotic balance. Initially, it would grow faster than the cytoplasm, as it does in our data (please see also the reply to point 4 of Reviewer#3). However, without more specific information our model cannot be easily used to suggest that this growth would be linear, which will require a specific and gradual release of external tension.

Recently, Rollin and coworkers have proposed a mechanism that could partly explain this trend (<https://www.biorxiv.org/content/10.1101/2022.08.01.502021v1.full>). In this preprint the authors claim a role of counterion release by chromatin folding. According to their model, the NC ratio (formula 15) can be intermediate between an expression that is the ratio between nuclear proteins and cytoplasmic proteins (NC1, the prediction of pure osmotic force balance) and one (NC2, larger) that depends on DNA charge, which is constant during

G1 phase, while the number of proteins in the nucleus grows with time, so NC2 decreases with time. Hence, this prediction, valid only in G1, could generate the right qualitative trend.

More widely, nuclear envelope tension might have at least two effects (acting on different timescales): one on nuclear import, which we focus on here, which would make the volume increase when the envelope feels more tension, while one direct mechanical effect would lead to a decrease of nuclear volume for a given osmotic balance at steady state. For high enough nuclear envelope tension, the volume loss effect might dominate, while for low tension (as in our study, due to cell spreading), the import effect might dominate.

Altogether, these synergies of complex effects make it hard to predict the volume growth of the nucleus during the cell cycle: a complete model would be needed, plus many more experiments.

iii) Related to the previous point, it does seem that, after the spreading phase (phase III), there is no or little growth of the nuclear volume, followed by more rapid growth. This seems true for the cell lines in Fig. S1L. Has this been considered in the calculation of volumetric growth? Could it be that the binning hides different growth regimes of the nucleus during the cell cycle?

We agree with the reviewer that at faster time scales, there may be several intriguing cell-cycle dependent phenomena that we missed in this study. We hope to perform higher-resolution time-resolved studies in the near future to address this and other questions. In particular Venkova and coworkers (Venkova et al, eLife 2022: <https://doi.org/10.7554/eLife.72381>) have shown nontrivial osmotic behavior of the cytoplasm during the spreading phase, so the referee is correct that it is reasonable to expect interesting effects for nuclear volume during vs after post-mitotic cell spreading.

iv) Finally, can the authors speculate as to why the nucleus would decrease by roughly 2.5 times its volume at mitosis? This suggests that it grows “more than necessary”, and that this could be linked with cell cycle-dependent growth regimes.

In a pure dynamic osmotic coupling it is expected to follow the cell volume pace up to double. We speculate that tension coupling dynamic contributes to the extra accumulated nuclear volume. But other processes could also contribute to this unbalance.

3/ Force-biased import and nuclear envelope tension

I find the conclusions of the authors that the force-biased import plays a role in regulating nuclear volume very appealing. I have a few questions regarding the experiments and the model:

i) Since the measurement of NE tension is not a standard method (and even though Ref. 22 is in press, it is still unpublished), maybe the authors should give more details about this method in the main text.

We apologize for the lack of details relative to the NE tension FRET sensor we developed. The paper from Poli and al. is now accepted (<https://doi.org/10.1038/s41467-023-37064-0>).

In the reply to reviewers section of that paper there are also additional controls. We hope now all the information relative to that valuable tool is publicly available.

ii) Related to point 2/ above, have the authors checked how is NE tension regulated during the cell cycle? Could tension explain why nuclear volume increases linearly?

Nuclear envelope tension has already been reported to increase during the cell cycle [Introini et al., doi: <https://doi.org/10.1101/2021.11.25.469847>; Chu et al., doi: <https://doi.org/10.1073/pnas.170222611>; Lomakin et al., doi: [10.1126/science.aba2894](https://doi.org/10.1126/science.aba2894)]. Interestingly, also assuming that NE tension increases linearly with time, that is not sufficient to explain how nuclear volume grows. The relationship between NE tension and nuclear volume, as previously stated, is complex and it could be both positive and negative. Positive, since an increase in NE tension would facilitate diffusion of molecules into the nucleus, affecting the osmotic equilibrium between the nucleus and the cytoplasm. Negative, since NE membrane tension counterbalances nucleus inflation.

iii) Related to the hyperosmotic experiments: although at first I was satisfied with the result, I am now a bit lost. It is known that a hyperosmotic shock rapidly induces changes in the cytoskeleton (see for instance Thirone, 2009, PMC5047760). As such, there should be an effect on NE tension, and by the authors' model, an effect on the nucleus volume, but it does not seem to be the case. Can the authors comment on this point? Maybe the effect is only on the cortex and not translated to the NE envelop?

This is an excellent point and indeed, as the reviewer, we were puzzled by that result. Interestingly, looking carefully at the movies, we noticed that upon hyperosmotic shock cells shrink immediately but mainly by lowering their height and not by changing their spreading area. We can speculate that this kind of compression response is much faster and reversible than restructuring of adhesion and contraction of the cell base. This seems to imply that disassembly of focal adhesions might happen later as a slower response/adaptation.

iv) Per the authors' model, it is my understanding that growing cells on substrata of different rigidities should impact the NC ratio. I did not find anything on this point in the literature. Do the authors know something about this?

We agree with the reviewer, our model indeed predicts that substrate rigidity, impacting on cell contractility and NE tension, would affect NC ratio. While the situation is of course way more complex than our model can describe, this is a very interesting point that we would like to address more in a future study. Interestingly, recent work from Pundel, Blowes and Connelly (<https://doi.org/10.1002/adv.202105545>), shows that extracellular physical cues affect nuclear and nucleolar volumes. Their results may be related to our model.

v) Related to the mathematical model: I appreciated the model and its explanation/parametrization strategy. I had one question on the choice of the exponential dependence of the nuclear concentration of macromolecules with external stress: why this choice? Why not linear in the first place? Especially because it seems that it is linearized in the equation defining V_n at the bottom of p. 5.

We thank the reviewer for the positive remarks on the model. We chose the exponential dependence under the hypothesis that biased transport due to the stress felt by the nuclear pores could be an activated process. However, as the reviewer correctly points out, this choice is equivalent to a linear-response assumption, and of course the data do not allow us to formulate any claim regarding this assumption. In the revised text we provided an explanation for our assumption and explicitly stated that to our scopes the assumption of linear response would be equivalent.

Minor comments

i) Figures

- adding the legend (control vs. ivermectin) on Fig. S3D would help the reader
- which condition is ivermectin in Fig. 4C? in general, what are the conditions in this plot?
- maybe ivermectin could be its own subfigure on Fig. 4 to help the reader and avoid going back to Fig. 3
- Y-axis in Fig. S2B: unit should be $\mu\text{m}^3/\text{h}$ and not μm^3

We apologize for the lack of clarity and mistakes. We improved the plot and amended the relative legend.

ii) Although the manuscript is nicely written, I found several typos, in particular in the appendix. I may have missed some, but here is a list of those I found:

- in methods section "cell transfection", problem unit polybrene
- in methods section "N2FXm", GFP-nes instead of NES
- in methods section "DAAM particle measurement", typo on "technique"
- in appendix, "left" and "right" hand side are inverted I believe
- in appendix, "nucleus surface tension and e contribution" → a
- in appendix, page 2, $c^{\text{ions_out}}$ c_{out} instead of just $c^{\text{ions_out}}$ (I believe)
- in appendix, section "estimates", in order TO confirm, prOessure
- in appendix, section "force biased", there is a verb lacking in first sentence
- p. 6 repetition "the the"

We apologize for these mistakes and we thank the reviewer for spotting them.

Reviewer #3 (Remarks to the Author):

This manuscript describes experiments and modeling of the regulation of nuclear size by osmotic forces and mechanical properties of the nuclear envelope in cultured mammalian cells. The study of the NC ratio in mammalian cells has been hampered by a lack of accurate volume measurements. This manuscript introduces an elegant new method for estimating nuclear/ cytoplasmic volume using a fluorescence exclusion-based approach, as well as a new FRET sensor for measuring nuclear envelope tension in cells. The work investigates NC ratio changes through the cell cycle and upon perturbations such as osmotic shock and cellular detachment from the substrate. The data support a mechano-osmotic model that implicates osmotic pressure and nuclear transport as critical components for nuclear size control.

In general, this work has potential to be an important advance in the field, but it is currently too preliminary for publication. There are many significant concerns described below. These include insufficient characterization of the new methods and a general tendency for over interpretation leading to inappropriate conclusions. They present some interesting initial observations without establishing sufficient context or additional insights. Some limited experimental data and careful rewriting will be needed to improve this manuscript.

We thank the reviewer for seeing potential in our work.

Major comments

1. The title, summary and parts of the abstract are not appropriate as key statements are not sufficiently backed by the data in this paper. For example, the Summary statement is: "Cytoskeletal forces exerted on the nuclear envelope impact on nuclear volume through modulation of force-coupled nucleo-cytoplasmic transport. However, there is little in this paper on cytoskeletal forces or nuclear transport. The closest test is an effect with ivernectin. However this result does not directly show that nuclear transport rates have changed. There is also no clear data on cytoskeletal forces on nuclear import; small effects with LatA are shown, but these may be highly pleiotropic and cannot be just assumed to be actin effects on the nuclear envelope. Their statement in the last paragraph "forces exerted by the cytoskeleton, affecting NE tension, impact on nuclear pores size (26–28)" have not been conclusively demonstrated in these cited papers and have not been proven here. While the authors certainly may hypothesize upon force-effects on nuclear transport and pores, these statements should not be the central conclusion of this paper. I think that the authors should address these concerns by judicious rewriting the wording of their conclusions, rather than being asked to perform the large number of experiments needed to support claims.

We agree with the reviewer that it is not the scope of our work to prove that forces exerted by the cytoskeleton affect nuclear transport by impacting on nuclear pores size. The phenomenon of tension-biased transport that we leverage to explain our data appears to be well supported by the literature. Indeed, the idea of tension-biased transport was introduced, supported and quantified in a seminal paper from the Roca-Cusachs lab (Elosegui-Artola et al., Cell 2017, ref. 27). The Roca-Cusachs lab recently published an elegant and systematic report on how cytoskeletal forces affect nuclear cytoplasmic transport (Andreu et al., NCB 2022, ref. 25). The idea that this process could be coupled to pore size was supported by two later independent structural studies, Zimmerli et al. (Science 2021, ref.26 ) and Schuller

et al. (Nature 2021, ref. 28), both showing that nuclear pores size depends on the cell's tensional state. However, this is not essential for us. In our study, Fig4.C, shows a clear correlation between NE tension and GFP.nls NC ratio, here used as a marker of nuclear cytoplasmic transport efficiency. As the reviewer highlights, based on our data it is reasonable to conclude that cytoplasmic forces, affecting NE tension, impact on nuclear cytoplasmic transport and nuclear volume. We completely agree that our data does not add anything on the question of nuclear pores size changes. We rephrased our conclusions accordingly in the revised manuscript.

2. There are important concerns about the N2FXm method that need to be addressed. Additional data and more detailed descriptions of the calibration method are needed so that the accuracy of the method can be evaluated (see below). The authors statement that the GFP "localizes in the entire cytoplasm except the nucleus" is not true. For example, other organelles besides the nucleus may also exclude the cytoplasmic GFP; lysozymes, mitochondria, lipid droplets etc., might represent 2-10% of cellular volume. This issue and how it may affect NC ratio measurements using N2JXm certainly needs to be addressed. Can the authors determine if these organelle volumes significantly affect the NC ratio measurements? Another concern is whether NES-GFP is truly all excluded from the nucleoplasm. Can the authors include a control using high-resolution confocal sections to show that there is no significant nuclear signal? If there is some nuclear signal, could they determine how that would affect the accuracy measurements? They should also show whether the localization of the NES-GFP marker is affected by their perturbations. For instance, mechanical or drug perturbations could cause entry of the NES-GFP in the nucleus, which could potentially affect their measurements and conclusions.

We thank the referee for these relevant remarks, which were also raised by reviewer 2. GFP-NES background in the nucleus, indeed, could affect the precision of N2FXm. Similarly, dishomogeneity in the distribution of GFP-NES in the cytoplasm, could perturb the second calibration, then impacting on nuclear volume estimates. We now explicitly discuss these possible drawbacks in the revised version of our manuscript. Additionally, following the reviewer's advice, we evaluated the distribution of the GFP-NES signal in correspondence of the nucleus (Rebuttal-Fig1.A). The mean radial intensity profile is minimum at the center and increases toward the periphery. In low magnification widefield images, the signal background simply originated by the presence of GFP-NES in the nucleus is expected to be rather constant or proportional to nucleus depth, then maximum at the center and minimum at the periphery. The profile we measured (middle panel in Rebuttal-Fig1.A) is instead compatible with a signal coming from the cytoplasm and then integrated over the cytoplasmic volume surrounding the nucleus. In the third plot in Rebuttal-Fig1.A we show the expected theoretical radial profile for a spherical nucleus (of unitary radius) embedded in a labeled cytoplasm:

$$f(r) = \int_0^r (1 - \sqrt{1-x^2}) dx = r - \frac{1}{2} * \arcsin(r) - \frac{1}{2} * r * \sqrt{1-r^2}.$$

This is compatible with the experimental profile we measured.

Importantly, we cannot exclude that a small portion of the signal we record in correspondence to the nucleus comes from inside the nucleus. Indeed we recorded a

background line on the radial profile that could have originated both by GFP-NES on top and bottom of the nucleus or by GFP-NES in the nucleus. However, when we evaluate in a confocal section (Rebuttal-Fig1.C) the proportion between cytoplasmic and nuclear GFP-NES signal, it becomes clear that the majority of the signal comes from the cytoplasm. In the mid-nucleus confocal sections, the mean intensity of GFP-NES in the cytoplasm is approximately 5 times higher than in the nucleus (Rebuttal-Fig1.C). Importantly, this ratio is not perturbed by drug treatments (Rebuttal-Fig1.C).

To allow the reviewers to judge directly if and how inhomogeneity in the distribution of GFP-NES signal in the cytoplasm could impact our measurements we plotted a representative example of a cumulative distribution of R^2 values, relative to more than 150 frames over 10 cells, for linear fits between GFP-NES cytoplasmic intensities and calibrated optical heights (Rebuttal-Fig1.B and ref to Figure 1.C). R^2 distribution is strongly biased towards 1, median = 0.93, sd = 0.14. Significant inhomogeneity in the distribution of GFP-NES signal, such as local accumulations or depauperation, would have perturbed the linear dependency between these two quantities and then the distribution of R^2 values.

All together these considerations suggest that GFP-NES background in the nucleus and as well as possible dishomogeneity in the distribution of GFP-NES in the cytoplasm and its impact on second calibration, are neglectable/minimal. This is probably due to the 'low' sensitivity and resolution of the imaging system settings as they are necessary for the N2FXm method to properly work, basically low numerical aperture and low magnification objective. Actually, it is important to notice that we challenged our method with two independent controls. First, we compared our nuclear volume measurements with 3D confocal reconstruction of nuclei, and we didn't record a significant shift in the distribution of nuclear volume estimates of the same cell population. Second, we measure in living cells the volume of artificial spherical objects (DAAMs particles). N2FXm measurements were in good agreement with theoretical volume computed by geometrical reconstruction. These two complementary validations well support the goodness of our method.

3. There are also concerns about the FRET-based nuclear envelope force sensor. This is a new sensor developed by this group, and its development is described only in this work and in a preprint. However, neither manuscript presents sufficient background characterization of this probe to demonstrate that it is acting as a force sensor at the nuclear envelope. Although the handful of data points are consistent, more quantitative and systematic controls would give more confidence that this sensor indeed measures force at the nuclear envelope.

We apologize for the lack of details relative to the NE tension FRET sensor we developed. The paper from Poli and al. is now accepted (DOI: 10.1038/s41467-023-37064-0). In the reply to reviewers section of that paper there are also additional controls. We hope now all the information relative to that valuable tool is publicly available.

4. One of the most surprising results is the linear growth of the nucleus vs. exponential growth of the cell volume. Other than pointing out the difference, this is not further investigated. More data are needed to confirm and develop this observation. One simple prediction is that NC ratios systematically fall as cells grow larger during interphase (however, this is not apparent in Figure 1F). Could this difference arise from some

systematic error in the N2FXm measurements? The reader is left without any context on how to think about this linear growth rate, or whether it carries much significance.

We agree with the reviewer: the observation that cell and nucleus volumes don't grow at the same pace is probably the most relevant result of our study. Most of our data focus on effects on time scales of minutes to less than an hour, and our explanations are fully coherent on these time scales. Cell-cycle time scales involve different processes and likely different explanations are necessary. Please see also reply to point ii of Reviewer#2. The fact that nuclear volume growth speed is constant (and independent of nuclear size) doesn't necessarily mean it is always slower than the cytoplasmic one. Indeed, while the cytoplasm doubles in volume during the cell cycle, the nucleus grows approximately 2.5 times (Fig2.E). To better clarify this point, we simulated time dependent volume growth curves for cyto and nucleus, and relative NC ratio (Rebuttal-Fig1.D). The cyto volume is assumed to grow exponentially while the nucleus linearly. The parameters used for the simulations, volume growth speed and minimal volume, are the ones obtained from the fit of experimental averages volume growth speed for RPE1 cell (Fig2.G). As the reviewer pointed out, at long time points, the NC ratio will collapse. However, for a time interval in the scale of a cell cycle and with the parameters obtained fitting the experimental data, the NC ratio is initially growing and lately starts to decay. For time points over the average span of a cell cycle, the NC ratio will certainly tend to zero. It would be for sure interesting to test our model in cells going to senescence and investigate the NC ratio dynamics in those conditions. This is however behind the scope of this work.

Moreover, we evaluated the time averaged values of cytoplasm and nucleus normalized volume growth speeds for all cell lines analyzed (Rebuttal-Fig1.D). Normalized average nucleus volume growth speeds are comparable, and typically higher (with the exception of the MCF7 cell line), than the one of the cytoplasm. This explains why the NC ratio doesn't systematically fall in the time scale of a cell cycle.

As suggested, in the new version of our manuscript we further discuss the linear versus exponential volume growth speed.

5. Another result is the apparent loss of nuclear volume after mitosis. Has it been confirmed by other methods or in other papers? Could the results be explained by the behavior of the NES marker at these cell cycle stages? If NC ratios are low after mitosis and then fall over interphase because of linear growth, then how does the nuclear volume ever catch up? How are NC ratios then maintained over multiple cell cycles? In Figure S2A, there is an example where nuclear growth suddenly speeds up after 50 frames (what is the frame rate?). Is this transition typical (as it is not seen in the averaged data)? Does it correspond to a known cell cycle transition? In general, these data are not discussed sufficiently in context to what is currently known about these processes. If the authors hope to describe thoroughly the NC ratio changes over the cell cycle, these issues should be clarified. Overall, the measurements of nuclear volume just before and after mitoses in Figure 2 represent a weak point in the manuscript and could even be deleted (or placed in another manuscript).

It is certainly true that our method can measure the volume of the nucleus only when this compartment is properly defined. In the few frames between nuclear envelope breakdown (NEB) and nucleus sealing after mitosis, centrally preceding nucleus postmitotic expansion (PME), the nucleus is not an isolated compartment anymore and can't be defined. During

this time interval, GFP-NES diffuses inside the nucleus and our method measures a residual volume that is still not accessible to it. We expressly clarify this point in the updated method session of our manuscript. For this reason, when we want to compare nuclear volume measurements before and after mitosis we compare them at NEB and PME, time points when the nucleus is still integer and when it is surely integer again, respectively. We apologize for the lack of explanation, nuclear post mitotic expansion (PME), the fast nuclear volume increase after mitosis, has been previously described, please see as example: Gerlich et al., NCB 2001; Baarlink et al., NCB 2017 and Krippner et al., EMBO Reports 2020. We now introduce PME with a relevant reference in the revised version of our manuscript.

Regarding how the nucleus can “catch up” its volume after mitosis, please see the reply to point 4.

The single-cell plot in Fig.S2A was intended to show the quality of the original data. Analyzing at the single-cell level the relationship between specific cell cycle stage or transition with nuclear and cellular volume growth is beyond the scope, and the reach, of this work. This goal can be potentially achieved with our method but will require a considerable amount of work in future studies.

Specific Comments

p.3 and Methods

More details of the calibration methods and evaluation of the accuracy of the approach are needed. First, can the authors provide an experimental version of Fig S1C (top right) to demonstrate linearity with the Texas red dextran marker they use?

We apologize for the lack of clarity, the plot in Fig1.C is indeed experimental. As previously mentioned, to better evaluate the robustness of fits along different frames and cells now in Rebuttal-Fig1.B we also show the cumulative distribution of R^2 relative to linear fits of GFP-NES intensity in function of cytoplasm optical heights measured with standard FXm. Please notice that the calibration is performed for each cell and at each time point.

Second, more detailed descriptions of how the NES-GFP intensity is calibrated with the cell volume data is needed. Can the authors show in a figure a real-life example (not just schematics)?

Please see the reply to the previous point. A real-life example was already provided in the original version of our manuscript.

For calibration, more details are needed on what cytoplasmic portions of the cell is used for comparison? How well do the red and green intensities inversely correlate; can the authors add plot showing correlation of green and red intensities per pixel? How is the noise in the intensity measurements dealt with for?

We apologize for the lack of clarity. For the GFP-NES calibration we use the intensity of all cell pixels excluding the ones in the nuclear area, as defined by the H2B signal. A real example of that is in Fig1.C. The linear dependency is robust, as can be appreciated by the cumulative distribution of R^2 for several single time frame linear fits from different cells (Rebuttal-Fig1.B).

Inhomogeneity in the cytoplasm could have a large impact on the interpretation of these measurements. Can the authors test whether inhomogeneities are significant enough to affect the calibration and subsequent measurements.? For example if there are many lysosomes in one part of the cytoplasm, is that portion of the cell not used for calibration?

We agree with the reviewer, that is an important point. Please see the response to your second major point. It is true, essentially, accumulation as well as depauperation of GFP-NES in the cytoplasm, as could be generated by vesicles excluding or concentrating the fluorescent protein, could perturb the linear dependency between the GFP-NES and the calibrated optical heights. Distribution of R^2 in Rebuttal-Fig1.B shows that is not the case.

L.7 p3 , L.6 p4 & L.27 p.5

Authors use the same nomenclature "NC ratio" for ratio of the nuclear to cell volume as well as nuclear to cytoplasmic volume. One definition should be used consistently. Also, the term "NC volume" might be reconsidered.

We apologize for this mistake and we amended it in the updated version of our manuscript.

L.12 p.4

"We also compared nuclear volume distribution of RPE1 cells measured with both N2FXm and 3D confocal reconstruction (see materials and methods) obtaining similar results"

This wording implies that the same cells were imaged with the two methods; is this true? The bar graph appears to show a sizable difference, but the authors say that it is not significant. Can they provide the mean values and the standard error? Can the authors include an example of the confocal z stacks with their markers. (This might address whether there is any detectable NES-GFP in the nucleus).

We apologize for the lack of clarity. The referee is correct, we did not measure the same cells, we refer to the same cell population. Now we clarified that sentence as: "We independently measured nuclear volume distribution of RPE1 cells with both N2FXm and 3D confocal reconstruction (see materials and methods). The difference between the distributions is not statistically significant."

In boxplot in Fig. S1.F, the boxes extend from the first to the third quartiles. The middle line represents the median. The whiskers extend up or down to $1.5 \times \text{IQR}$, where IQR is the inter-quartile range.

L.23 p.4

"DCIS.com "

Are the authors referring to the cell line "MCF10 DCIS.COM"? This should be clarified. Also, in the methods, the authors refer to DCIS.com medium; is this correct?

We apologize for the non appropriate naming. In the main text and in the method section of our revised manuscript we now specify that in text, figure and legend we used the name "DCIS.com" as an abbreviation of "MCF10 DCIS.COM".

Figure1

Can Panel D and E be plotted with the same y-axis to facilitate comparison.

Yes, in the new version of figure 1 now the two plots have the same scale.

Figure 2A

As the nucleus breaks down during mitosis, it is confusing what is being measured with this assay when there is no intact nucleus. This should also be clarified in the text. Consider removing these values after NEB on the graph, or make them a lighter orange color to denote that it is not really the nuclear volume.

This is a very important point and as suggested we clarified it in the text. We agree with the reviewer, indeed at the envelope breakdown there is no nucleus anymore and GFP-NES diffuse in the nucleus. Interestingly, with our method we measure a residual volume that is still not accessible to it. We could only speculate that this is the space occupied by the chromatin.

L.3 p5 & Figure 2C

"We found that NEB systematically precedes the onset of cellular roundup by ~10-20 min."

This statement is not backed well by the corresponding graph. The time points in the graph of 20 min apart do not allow for an accurate assessment of timing, and three of the five strains have similar percentages at -20 and 0 time points. These data should be discussed in relationship to previously published data on this time.

We agree with the reviewer and indeed in the text we specified: "However, the temporal resolution of our experiments (10 min) was too small to precisely distinguish these two events."

L10. P.5

"This implies a non-constancy of the NC volume ratio between NEB and PME (Fig. 2F)"

Can the authors present a statistical test between the paired Nuc and Cyto ratio results presented in Figure E and the NC ratio at PME and NEB in panel F.

We thank the reviewer for this suggestion and we added test results in the updated version of Fig2.

L12 p5

The results on growth rates need to be better described. Currently it is presented as the last graph in Figure 2, along the division data, and discussed in the same paragraph as the cell division, but describes behavior on an entirely different magnitude time scale. This is confusing to the reader. At the very least, the data on growth should be discussed in separate paragraphs for division in the text, and ideally presented in separate figures. As mentioned above, more context to this result should be given.

As suggested we describe this result in a separate paragraph.

L.17 p5

"Overall, these results indicate that in mammalian cells the nucleus-cytoplasm volumetric coupling could not be simply defined by a pure osmotic equilibrium, which would lead, instead, to a constant value of the NC ratio (16, 17)."

There are certainly circumstances in which NC ratios can vary even if they only use osmotic mechanism. Reference 16 shows that if the nuclear growth rate speed is proportional to the cell volume then a pure osmotic theory can explain N/C ratio maintenance. Can the authors show the Vol growth speed for nuclei as a function of Norm volume for cells? Is it flat or linear? If it is not linear than the pure osmotic model cannot explain these results.

As suggested by the reviewer we plotted nuclear volume grow speed as a function of cell volume (Rebuttal-Fig1.E). Nuclear volume grow speed doesn't correlate with cyto volume.

L.28 p5

“As expected from a purely osmotic model”

Can the authors clarify what they define as a "purely osmotic model". Can they refer to an equation? They cite this pure osmotic model with references 16, 17, but these papers use models that also take in account membrane tension.

We apologize, we correct that sentence with:“As expected from a model considering only osmotic equilibrium and membrane tension (eq.2 in SI appendix)” .

L.30 p.5

“Our model also shows that the same behavior is expected for a non-negligible constant surface tension, with a small correction on the slope, but the expected nuclear volume changes due to external forces are [...]”

Can they refer to an equation?

We now refer to eq.6 in the SI Appendix.

Figure 3 H & I : The plots look very similar, even in the SD. Please confirm that the NC data shown are correct.

We thank the reviewer and we confirm that data plotted are correct.

L.11 p.6

“was mostly unaffected along hyperosmotic shocks (Fig. 3E).”

Can the authors conduct a test for the Norm Mean Inv FRETindex at t=0 min and after the hypertonic shock t>10 minutes. It looks like there is a significant difference

We thank the reviewer for noticing it and indeed there is a statistical difference. Now we added statistical test analysis to the plots, please see updated version of figure 3.

Figure 3

Panel E,J,O. Can these be plotted with the same y axis to facilitate comparison?

Panel C,D,E. Can the authors label when the hyper osmotic shock occurs on the graph?

Figure 3 H & I.The plots look very similar, even in the SD. Please confirm that the NC data shown are correct.

Thank you for these suggestions. Following the reviewer's advice we updated Fig.3.

L.19 p.6

“nuclear and cytoplasmic volumes are strongly decoupled, with nuclear volume variations coherent with the changes of forces exerted on the NE and not with the changes of cytoplasmic volume (Fig. 4A).”

It would be important to test if NES-GFP localization is altered during these experiments.

Please find in Rebuttal-Fig1.C single plane confocal images of GFP-NES upon drug treatments. Moreover, the mean intensity ratio between cytoplasm and nucleus is not perturbed by drugs treatments.

L.21 p.7

"specific transcription factor such as YAP, key regulator of organ growth and regeneration as well as of mechanotransduction, also are affected by this mechanism."

Please cite a reference for this statement.

We apologize for this omission. We now add also here citation to ref.27 as it should be.

References

Update Lemiere reference #16

Fix Deveri reference #17

Fix Zimmerli reference #26

Thank you, we updated these references in the new version of our manuscript.

Rebuttal-Fig1. A. Left, widefield representative image of RPE1 cell expressing GFP-NES; highlighted in white the nucleus area; scale bar $20 \mu\text{m}$. Middle, average experimental radial profile of GFP-NES mean intensity in the nucleus area. Right, theoretical prediction of radial

profile. B. Cumulative distribution over several frames and different cells of R^2 relative to linear fits of pixels GFP-NES intensity in the cytoplasm in function of the corresponding pixel optical height computed with traditional FXm. C. Left, single plane confocal representative image of RPE1 cell expressing GFP-NES; highlighted in white the line used for the relative intensity scan; scale bar $20 \mu\text{m}$. Middle, line intensity profile relative to the image on the left. Right, cytoplasm to nucleus ratio of GFP-NES mean intensity for ctrl, Y-compound and latrunculin treated cells with relative representative images, scale bars $20 \mu\text{m}$. D. Left, simulations of time dependent cytoplasm and nucleus volume growth curves. Parameters used for the simulation were extrapolated from the fit of experimental grow speed of RPE1 cells (main Fig2.G). Middle, nucleus to cytoplasm volume ratio as obtained from the simulation. Right, average normalized cytoplasm and nucleus volume grow speeds. E. Conditional average of nuclear grow speed in function of cytoplasm volume.